# Spectral Analysis of Diffusion Models with Application to Schedule Design

**Roi Benita**
Department of Electrical and Computer Engineering
Technion, Haifa, Israel
roibenita@campus.technion.ac.il

**Michael Elad**
Department of Computer Science
Technion, Haifa, Israel
elad@cs.technion.ac.il

**Joseph Keshet**
Department of Electrical and Computer Engineering
Technion, Haifa, Israel
jkeshet@technion.ac.il

## Abstract

Diffusion models (DMs) have emerged as powerful tools for modeling complex data distributions and generating realistic new samples. Over the years, advanced architectures and sampling methods have been developed to make these models practically usable. However, certain synthesis process decisions still rely on heuristics without a solid theoretical foundation.

In our work, we offer a novel analysis of the DM's inference process, introducing a comprehensive frequency response perspective. Specifically, by relying on Gaussianity assumption, we present the inference process as a closed-form spectral transfer function, capturing how the generated signal evolves in response to the initial noise. We demonstrate how the proposed analysis can be leveraged to design a noise schedule that aligns effectively with the characteristics of the data. The spectral perspective also provides insights into the underlying dynamics and sheds light on the relationship between spectral properties and noise schedule structure. Our results lead to scheduling curves that are dependent on the spectral content of the data, offering a theoretical justification for some of the heuristics taken by practitioners.

## 1 Introduction

Diffusion Models (DMs) have become powerful tools for generating high-quality, diverse signals, with applications in image, audio, and video synthesis. Alongside their practical success and the ability to handle complex distributions, some aspects of the diffusion processes still rely on heuristics rooted in empirical experimentation. A key example is choosing an appropriate noise schedule for the inference phase. Developing theoretical foundations for these heuristics may provide valuable insights into the diffusion process itself, and enable greater adaptation to different setups. Our work aims to provide such a theoretical backbone from a spectral perspective, as outlined below.

Our starting point lies in a well-established observation; while the continuous description of DMs via SDE or the associated flow ODE [28] are mathematically well-founded, their practice necessarily deviates from these theoretical foundations, introducing various errors [4, 24]. A major source of error is discretization, which replaces the analytical formulations by their discrete-time counterparts. Another source of error is the approximation error, originating from the gap between the ideal denoiser and its neural network realization.

39th Conference on Neural Information Processing Systems (NeurIPS 2025).

Significant efforts have been made to minimize these associated errors, with advanced numerical solvers [28, 12, 40, 20, 21, 42, 41] offering various methods for better treating the discretization of the differential equations. A fundamental aspect is the decision on the time point discretization, which directly affects the synthesis quality. We refer to these anchor points as the *noise schedule*, highlighting their role in setting the noise variance introduced at each diffusion step. Realizing their importance, researchers have recently shifted their focus from custom-tailored heuristics [10, 23, 13, 5] to the development of optimized noise schedules [26, 29, 34, 32, 37, 3, 38, 36]. More on these methods and their relation to this paper's contributions is detailed in Section 5.

In this work, we analyze the reverse diffusion process in the spectral domain, presenting the generated output signal as the outcome of a linear transfer function operating on the i.i.d. Gaussian input noise. This analysis is enabled by assuming that the destination distribution to sample from is Gaussian, providing an explicit spectral expression of the generated signal and insight on the evolving dynamics throughout the process. Our analysis, applied to the discretized inference procedure, encompasses both DDPM [10] and DDIM [27] frameworks, thereby revealing the effects of deviating from the continuous formulation. In addition, we extend our study for both variance preserving (VP) and variance exploding (VE) numerical schemes [28].

Posing the derived explicit expressions of the transfer systems as functions of the noise scheduling parameters, we can optimize a noise schedule tailored to a given dataset, its resolution, and the specified number of sampling steps required. We demonstrate how effectively solving these optimization problems numerically yields a noise schedule that accounts for these data characteristics, discuss the relation between the noise schedule structure and spectral properties, and validate our approach with existing works. Finally, we apply our method to publicly available datasets, including CIFAR-10 [18], AFHQv2 [6], MUSIC [22] and SC09 [33], examining the relation to heuristic choices in past work.

In summary, our contributions are the following: (i) Assuming the target signal is drawn from a Gaussian distribution, we present a novel spectral analysis on the discrete diffusion reverse process and derive a closed-form expression for its spectral transfer function. (ii) We formulate an optimization problem to find an optimal noise schedule that aligns with the data characteristics. Our approach provides an effective solution without relying on bounds or constraints on the number of diffusion steps. (iii) We analyze the evolving spectral properties of the signal throughout the diffusion process and at its output, demonstrating how handcrafted noise scheduling decisions and related diffusion phenomena are often well-predicted by our approach. (iv) Our spectral analysis examines various setups, including DDIM and DDPM procedures, VP and VE formulations, the selection of loss functions, and additional features such as expectancy drift.

## 2   Background

We introduce the notations and the framework of diffusion probabilistic models, which are designed to generate samples $\mathbf{x}_0 \in \mathbb{R}^d$ from an underlying, unknown probability distribution $p(\mathbf{x}_0)$. While the sampling procedure can be described as a Stochastic Differential Equation (SDE) or by its associated flow, an Ordinary Differential Equation (ODE), these formulations lack a general analytical solutions and are instead discretized and solved using numerical methods [28]. Accordingly, we turn to describe the discrete formulations – DDPM [10] and DDIM [27] – which stand as the basis for our work.

The diffusion process is a generative procedure constructed from two stochastic paths: a *forward* and a *reverse* trajectories in which data flows [10]. Each process is defined as a fixed Markovian chain composed of $T$ latent variables. During the forward process, a signal instance is gradually contaminated with white additive Gaussian noise as follows:[1]

$$\mathbf{x}_t = \sqrt{\alpha_t}\mathbf{x}_{t-1} + \sqrt{1 - \alpha_t}\boldsymbol{\epsilon}_t \ , \tag{1}$$

where $\alpha_t$ for $t \in [1, T]$ is referred to as the incremental noise schedule and $\boldsymbol{\epsilon}_t \sim \mathcal{N}(\mathbf{0}, \mathbf{I})$. Under the assumption that $\alpha_T$ is close to zero, we get that the final latent variable becomes $x_T \sim \mathcal{N}(\mathbf{0}, \mathbf{I})$. A direct consequence of the above equation is an alternative relation of the form:

$$\mathbf{x}_t = \sqrt{\bar{\alpha}_t}\mathbf{x}_0 + \sqrt{1 - \bar{\alpha}_t}\boldsymbol{\epsilon} \quad \boldsymbol{\epsilon} \sim \mathcal{N}(\mathbf{0}, \mathbf{I}) \ , \tag{2}$$

---

[1]Here we adopt the variance-preserving (VP) setting; see Appendix F for the variance-exploding (VE) form.

for $\bar{\alpha}_t = \prod_{i=1}^{t} \alpha_i$. Based on the above relationships, the reverse process aims to reconstruct $\mathbf{x}_0$ from the noise $\mathbf{x}_T$ by progressively denoising it. This can be written as

$$\mathbf{x}_{t-1} = \frac{1}{\sqrt{\alpha_t}} \left( \mathbf{x}_t - \frac{1 - \alpha_t}{\sqrt{1 - \bar{\alpha}_t}} \boldsymbol{\epsilon}_\theta(\mathbf{x}_t, t) \right) + \sigma_t \mathbf{z}_t \,, \tag{3}$$

where $\boldsymbol{\epsilon}_\theta$ is the estimator of $\boldsymbol{\epsilon}$ for a given $\mathbf{x}_t$ at time $t$ with a neural network parameterized by $\theta$, $\sigma_t = \sqrt{\frac{1 - \bar{\alpha}_{t-1}}{1 - \bar{\alpha}_t}(1 - \alpha_t)}$ and $\mathbf{z}_t \sim \mathcal{N}(\mathbf{0}, \mathbf{I})$. Alongside this stochastic formulation, The authors of [27] introduce DDIM, a deterministic framework for the inference process, which can be utilized to enable faster sampling. While DDIM relies on a non-Markovian forward process, it preserves the marginal distribution as in (2). As a result, the sampling trajectory can be expressed by

$$\mathbf{x}_{s-1} = \sqrt{\bar{\alpha}_{s-1}} \left( \frac{\mathbf{x}_s - \sqrt{1 - \bar{\alpha}_s} \cdot \boldsymbol{\epsilon}_\theta(\mathbf{x}_s, s)}{\sqrt{\bar{\alpha}_s}} \right) + \sqrt{1 - \bar{\alpha}_{s-1}} \cdot \boldsymbol{\epsilon}_\theta(\mathbf{x}_s, s), \tag{4}$$

where $s \in \{0, 1, \ldots, S\}$ is the time index in the DDIM formulation. Throughout the rest of the paper, we denote by $\bar{\boldsymbol{\alpha}}$ the set of noise schedule parameters, $\{\bar{\alpha}_t\}_{t=0}^{T}$ for DDPM and $\{\bar{\alpha}_s\}_{s=0}^{S}$ for DDIM.

## 3 Analysis of Diffusion processes

We consider the reverse process as a system that takes as input a noisy signal $\mathbf{x}_T$ and outputs $\mathbf{x}_0$. In this section, we develop the transfer function, which characterizes the relationship between these inputs and outputs in the spectral domain. To do so, we assume that the output signals are vectors drawn from a Gaussian distribution,

$$\mathbf{x}_0 \sim \mathcal{N}(\boldsymbol{\mu}_0, \boldsymbol{\Sigma}_0) \,, \tag{5}$$

where $\boldsymbol{\mu}_0 \in \mathbb{R}^d$ and $\boldsymbol{\Sigma}_0 \in \mathbb{R}^{d \times d}$. A similar assumption was used in previous work [24, 26, 31]. While this model simplifies the signal distribution, we will demonstrate that it facilitates spectral analysis throughout the diffusion process and allows us to design a noise scheduling mechanism for different objectives.

### 3.1 The optimal denoiser for a Gaussian input

The following theorem states a well-known fact: under the above Gaussianity assumption, the Minimum Mean-Squared Error (MMSE) denoiser operating on $\mathbf{x}_t$ to recover $\mathbf{x}_0$ is linear. It is the Wiener Filter [35] and can be expressed in a closed form.

**Theorem 3.1.** *Let* $\mathbf{x}_0 \sim \mathcal{N}(\boldsymbol{\mu}_0, \boldsymbol{\Sigma}_0)$ *and let* $\mathbf{x}_t$ *be defined by* (2). *Then, the denoised signal obtained from the MMSE (and the MAP) denoiser is given by:*

$$\mathbf{x}_0^* = (\bar{\alpha}_t \boldsymbol{\Sigma}_0 + (1 - \bar{\alpha}_t)\mathbf{I})^{-1} \left( \sqrt{\bar{\alpha}_t} \boldsymbol{\Sigma}_0 \mathbf{x}_t + (1 - \bar{\alpha}_t) \boldsymbol{\mu}_0 \right) . \tag{6}$$

A detailed proof is given in Appendix A. Here, we outline the main steps under the assumption of Gaussian distributions, where applying the Maximum A Posteriori (MAP) estimator is equivalent to minimizing the MSE. The MAP estimator seeks to maximize the posterior probability:

$$\max_{\mathbf{x}_0} \log p(\mathbf{x}_0 | \mathbf{x}_t) = \min_{\mathbf{x}_0} - \log p(\mathbf{x}_t | \mathbf{x}_0) - \log p(\mathbf{x}_0). \tag{7}$$

By substituting the explicit density functions $p(\mathbf{x}_0)$ and $p(\mathbf{x}_t | \mathbf{x}_0)$ according to (2) into the above, and differentiating with respect to $\mathbf{x}_0$ we obtain the desired result.

### 3.2 The reverse process in the time domain

We now turn to analyze the discrete sampling procedures, as introduced in [10, 27] and described in Section 2. We begin by focusing on the DDIM formulation presented in [27], as it highlights the fundamental principles more clearly and facilitates the analysis of faster sampling techniques. The lemma below describes the relationship between two adjacent time steps during the inference process.

**Lemma 3.2.** *Assume $\mathbf{x}_0 \sim \mathcal{N}(\boldsymbol{\mu}_0, \boldsymbol{\Sigma}_0)$ and let $\bar{\boldsymbol{\alpha}}$ be the noise schedule parameters, we have*

$$\mathbf{x}_{s-1} = \left( a_s \mathbf{I} + b_s \sqrt{\bar{\alpha}_s} \bar{\boldsymbol{\Sigma}}_{0,s}^{-1} \boldsymbol{\Sigma}_0 \right) \mathbf{x}_s + b_s (1 - \bar{\alpha}_s) \bar{\boldsymbol{\Sigma}}_{0,s}^{-1} \boldsymbol{\mu_0}, \tag{8}$$

*where $\bar{\boldsymbol{\Sigma}}_{0,s} = \bar{\alpha}_s \boldsymbol{\Sigma}_0 + (1 - \bar{\alpha}_s)\mathbf{I}$, and the coefficients are*

$$a_s = \frac{\sqrt{1 - \bar{\alpha}_{s-1}}}{\sqrt{1 - \bar{\alpha}_s}} \ , \ \ b_s = \sqrt{\bar{\alpha}_{s-1}} - \frac{\sqrt{\bar{\alpha}_s}\sqrt{1 - \bar{\alpha}_{s-1}}}{\sqrt{1 - \bar{\alpha}_s}} \ .$$

The above is obtained by plugging the optimal denoiser into the DDIM reverse process in (4). The derivation is given in Appendix B. This lemma establishes an explicit connection between adjacent diffusion steps, incorporating the characteristics of the destination signal density function as expressed by $\boldsymbol{\mu}_0$ and $\boldsymbol{\Sigma}_0$, along with the chosen noise schedule parameters $\bar{\boldsymbol{\alpha}}$.

### 3.3 Migrating to the spectral domain

Analyzing the diffusion process in the time domain can be mathematically and computationally challenging, particularly in high-dimensional spaces ($d \gg 1$). To address this, we leverage the spectral decomposition of the covariance matrix, simplifying the analysis and enabling the examination of the signal behavior along its eigen-directions.

Consider a destination signal $\mathbf{x}_0$ drawn from a multivariate Gaussian distribution with a given mean $\boldsymbol{\mu}_0$ and a covariance matrix $\boldsymbol{\Sigma}_0$, and let $\mathbf{U} \in \mathbb{R}^{d \times d}$ be a unitary matrix whose columns are the eigenvectors of $\boldsymbol{\Sigma}_0$. The projection of $\mathbf{x}_0$ onto this eigenbasis, denoted by $\mathbf{v}_0 = \mathbf{U}^\top \mathbf{x}_0$, also follows a Gaussian distribution. Specifically, $\mathbf{v}_0 \sim \mathcal{N}(\boldsymbol{\mu}_0^{\mathbf{u}}, \boldsymbol{\Lambda_0})$, where $\boldsymbol{\mu}_0^{\mathbf{u}} = \mathbf{U}^\top \boldsymbol{\mu}_0 \in \mathbb{R}^d$ is the transformed mean vector, and $\boldsymbol{\Lambda}_0 \in \mathbb{R}^{d \times d}$ is a positive semi-definite diagonal matrix, containing the eigenvalues of $\boldsymbol{\Sigma}_0$, denoted $\{\lambda_i\}_{i=1}^d$. Based on Lemma 3.2, we derive the following property:

**Lemma 3.3.** *Assume $\mathbf{x}_0 \sim \mathcal{N}(\boldsymbol{\mu}_0, \boldsymbol{\Sigma}_0)$. The covariance matrices associated with all intermediate steps $\{\mathbf{x}_s\}_{s=0}^S$ in the diffusion process are jointly diagonalizable using the eigenbasis $\mathbf{U}$ of $\boldsymbol{\Sigma}_0$.*

This lemma holds since all the diffusion steps in (8) involve only $\boldsymbol{\Sigma}_0$, its inverse, or the addition of the identity matrix, leading to covariance matrices that share the same eigenbasis. We now turn to describe the diffusion reverse process in the spectral domain. By projecting both sides of (8) onto the eigenbasis $\mathbf{U}$, we obtain the following result.

**Lemma 3.4.** *Assume $\mathbf{x}_0 \sim \mathcal{N}(\boldsymbol{\mu}_0, \boldsymbol{\Sigma}_0)$ and let $\bar{\boldsymbol{\alpha}}$ be the noise schedule parameters. The subsequent step in the reverse process can be expressed in the spectral domain via*

$$\mathbf{v}_{s-1} = \mathbf{G}(s)\mathbf{v}_s + \mathbf{M}(s)\boldsymbol{\mu}_0^{\mathbf{u}}, \tag{9}$$

*where $\mathbf{v}_s$ denotes the projection of the signal $\mathbf{x}_s$ onto the eigenbasis $\mathbf{U}$,*

$$\mathbf{G}(s) = \left[ a_s \mathbf{I} + b_s \sqrt{\bar{\alpha}_s} \left[ \bar{\alpha}_s \boldsymbol{\Lambda}_0 + (1 - \bar{\alpha}_s)\mathbf{I} \right]^{-1} \boldsymbol{\Lambda}_0 \right] \ and \ \mathbf{M}(s) = b_s(1 - \bar{\alpha}_s) \left[ \bar{\alpha}_s \boldsymbol{\Lambda}_0 + (1 - \bar{\alpha}_s)\mathbf{I} \right]^{-1} .$$

The lemma is proven in Appendix C. Equation (9) describes the relationship between two consecutive steps in the reverse process. Note that both matrices, $\mathbf{G}(s)$ and $\mathbf{M}(s)$, are diagonal, and thus, the reverse process in the spectral domain turns into a system of $d$ independent scalar equations. Based on the above relationship, we may derive an expression for the generated signal in the spectral domain, denoted as $\hat{\mathbf{v}}_0$. The complete derivation of the following result can be found in Appendix C as well.

**Theorem 3.5.** *Assume $\mathbf{x}_0 \sim \mathcal{N}(\boldsymbol{\mu}_0, \boldsymbol{\Sigma}_0)$ and let $\bar{\boldsymbol{\alpha}}$ be the noise schedule parameters. The generated signal in the frequency domain $\hat{\mathbf{v}}_0$ can be described as a function of $\mathbf{v}_S$ via*

$$\hat{\mathbf{v}}_0 = \mathbf{D}_1 \mathbf{v}_S + \mathbf{D}_2 \boldsymbol{\mu}_0^{\mathbf{u}}, \tag{10}$$

*where $\mathbf{v}_S \sim \mathcal{N}(\mathbf{0}, \mathbf{I})$ , $\mathbf{D}_1 = \prod_{k=1}^S \mathbf{G}(k)$ and $\mathbf{D}_2 = \sum_{i=1}^S \left( \prod_{j=1}^{i-1} \mathbf{G}(j) \right) \mathbf{M}(i)$ .*
*Moreover, $\hat{\mathbf{v}}_0$ follows Gaussian distribution:*

$$\hat{\mathbf{v}}_0 \sim \mathcal{N}(\mathbf{D}_2 \boldsymbol{\mu}_0^{\mathbf{u}}, \mathbf{D}_1^2), \quad \hat{\mathbf{v}}_0 \in \mathbb{R}^d. \tag{11}$$

Equation (10) provides a novel view of the generated signal in the spectral domain. Specifically, we can view (10) as a *transfer function* which models the relationship between the input signal $\mathbf{v}_S$ and

the output $\hat{\mathbf{v}}_0$. Furthermore, since the matrices $\mathbf{D}_1$ and $\mathbf{D}_2$ are diagonal, the expression simplifies to a set of $d$ scalar *transfer functions*, with the only parameters being the noise schedule, $\bar{\boldsymbol{\alpha}}$.

So far, we examined DDIM. Similarly, a closed-form expression for the stochastic DDPM method [10] is presented in the following Theorem and is proven in Appendix E.

**Theorem 3.6.** *Assume* $\mathbf{x}_0 \sim \mathcal{N}(\boldsymbol{\mu}_0, \boldsymbol{\Sigma}_0)$ *and let* $\bar{\boldsymbol{\alpha}}$ *be the noise schedule parameters. The signal* $\hat{\mathbf{v}}_0$ *generated by DDPM can be expressed as a function of* $\mathbf{v}_T$ *via*

$$\hat{\mathbf{v}}_0 = \mathbf{D}_1 \mathbf{v}_T + \sum_{i=1}^{T} \left( \prod_{j=1}^{i-1} \mathbf{G}(j) \right) c_i \mathbf{z}_i^{\boldsymbol{u}} + \mathbf{D}_2 \boldsymbol{\mu}_0^{\boldsymbol{u}} \tag{12}$$

*where* $\mathbf{v}_T \sim \mathcal{N}(\mathbf{0}, \mathbf{I})$, $c_i = \sqrt{\frac{1-\bar{\alpha}_{i-1}}{1-\bar{\alpha}_i}(1-\alpha_i)}$ . *The terms* $\mathbf{D}_1, \mathbf{D}_2$ *and the matrices* $\mathbf{G}(j)$ *are defined in Appendix E. Moreover,* $\hat{\mathbf{v}}_0$ *follows a Gaussian distribution:*

$$\hat{\mathbf{v}}_0 \sim \mathcal{N}\left( \mathbf{D}_2 \boldsymbol{\mu}_0^{\boldsymbol{u}} , \mathbf{D}_1^2 + \sum_{i=1}^{T} \left( \prod_{j=1}^{i-1} \mathbf{G}^2(j) \right) c_i^2 \mathbf{I} \right). \tag{13}$$

## 4 Optimal spectral schedules

With the closed-form expressions in (10) and (12), we can now explore different aspects of the diffusion process and examine how subtle changes in its design affect the output distribution. More specifically, a key aspect in this design is the choice of the noise schedule. In the discussion that follows we demonstrate how the proposed scheme enables optimal scheduler design.

We start by focusing on the direct dependence between the generated distribution and the noise schedule coefficients, $\bar{\boldsymbol{\alpha}}$. We define the spectral domain probability density function of the diffusion process output as $p(\hat{\mathbf{v}}_0; \bar{\boldsymbol{\alpha}})$. Our objective is to bring this distribution to become as close as possible to the original distribution, $p(\mathbf{v}_0)$. Specifically, given a dataset with a covariance matrix, $\boldsymbol{\Sigma}_0$, defined by the eigenvalues $\{\lambda_i\}_{i=1}^{d}$ and $S$ diffusion steps, our goal is to identify the coefficients $\bar{\boldsymbol{\alpha}}$ that minimize some distance $\mathcal{D}$ between these two distributions. This results in the following optimization problem with a set of specified constraints:

$$\bar{\boldsymbol{\alpha}}^* = \arg \min_{\bar{\boldsymbol{\alpha}}} \mathcal{D}\left(p(\hat{\mathbf{v}}_0; \bar{\boldsymbol{\alpha}}), p(\mathbf{v}_0)\right) \tag{14}$$
$$\text{subject to} \quad \bar{\alpha}_0 = 1 - \varepsilon_0, \quad \bar{\alpha}_S = \varepsilon_S,$$
$$\bar{\alpha}_{s-1} \geq \bar{\alpha}_s \text{ for } s = 1, \ldots, S.$$

The equality constraints ensure compatibility between the training and the synthesis processes. This involves ending the diffusion process with white Gaussian noise and starting it with very low noise to capture fine details in the objective distribution accurately [19]. The inequality constraints align with the core principles of diffusion models and their gradual denoising process [10]. The distance $\mathcal{D}$ between the probabilities can be chosen depending on the specific characteristics of the task. In this work, we consider the *Wasserstein-2* and *Kullback-Leibler divergence*, but other distances can also be used. The theorems presented below are detailed in Appendix D.

The *Wasserstein-2* distance (or Earth Mover's Distance) measures the minimal cost of transporting mass to transform one probability distribution into another. In the case of measuring a distance between two Gaussians, this has a closed-form expression.

**Theorem 4.1.** *The* Wasserstein-2 *distance* $\mathcal{D}_{W_2}$ *between* $p(\hat{\mathbf{v}}_0; \bar{\boldsymbol{\alpha}})$ *and* $p(\mathbf{v}_0)$ *is given by:*

$$\mathcal{D}_{W_2}^2\left(p(\hat{\mathbf{v}}_0; \bar{\boldsymbol{\alpha}}), p(\mathbf{v}_0)\right) = \sum_{i=1}^{d} \left( \sqrt{\lambda_i} - [\mathbf{D}_1]_i \right)^2 + \sum_{i=1}^{d} [\boldsymbol{\mu}_0^{\boldsymbol{u}}]_i^2 \left( [\mathbf{D}_2]_i - 1 \right)^2, \tag{15}$$

*where* $\{\lambda_i\}_{i=1}^{d}$ *denote the* $d$ *eigenvalues of* $\boldsymbol{\Sigma}_0$.

The KL divergence $D_{\mathrm{KL}}(P \| Q)$ assesses how much a model probability distribution $Q$ differs from a reference probability distribution $P$. Note that this divergence is not symmetric. As with the Wasserstein-2 case, here as well we obtain a closed-form expression for the two Gaussians considered.

**Theorem 4.2.** *The Kullback-Leibler divergence between the generated distribution $p(\hat{\mathbf{v}}_0; \bar{\boldsymbol{\alpha}})$ and the true distribution $p(\mathbf{v}_0)$ is given by*

$$D_{\mathrm{KL}}(p(\mathbf{v}_0) \| p(\hat{\mathbf{v}}_0; \bar{\boldsymbol{\alpha}})) = \sum_{i=1}^{d} \log [\mathbf{D}_1]_i - \frac{1}{2} \left( \sum_{i=1}^{d} \log \lambda_i - d + \sum_{i=1}^{d} \frac{\lambda_i + ([\mathbf{D}_2]_i - 1)^2 (\boldsymbol{\mu_0^u})_i^2}{[\mathbf{D}_1]_i^2} \right) \quad (16)$$

For solving the resulting optimization problems, we have employed the Sequential Least SQuares Programming (SLSQP) method [17], a well-suited method for minimization problems with boundary conditions, and equality and inequality constraints.

Before turning into the empirical evaluation of the proposed optimization, and the implications of the obtained noise scheduling on the various schemes, we pause to describe related work in this field.

## 5 Related work

Recent work has acknowledged the importance of the noise scheduling in diffusion models, and the need to shift the focus from custom-tailored heuristics [10, 23, 13, 5] to the development of optimized alternatives. For instance, the authors of [26] introduced the KL-divergence Upper Bound (KLUB), which minimizes the mismatch between the continuous reverse-time SDE and its linearized approximation. However, this approach, along with others [34, 32, 37, 29], aim to minimize the estimation error as well, hence requires retraining a denoiser or entailing substantial computation time and resources when solving the optimization problem.

While pursuing the same goal of optimizing the noise schedule, the work reported in [3, 38, 36] made notable strides in simplifying the induced optimization problem. Such efficiency is enabled in [38] by introducing an upper bound on the truncation error and assuming the data-dependent component to be negligible during optimization. Yet, despite this improved scalability, a direct relationship between data characteristics and the derived noise schedule still remains vague.

A potential approach to bridge this gap is spectral analysis, a fundamental tool in signal processing, which frequently serves to associate design choices and properties of diffusion models with data characteristics [25, 2, 8, 39, 7]. The application of frequency representation is diverse, ranging from introducing the *coarse-to-fine* behavior in the reverse process [25], to uncover unique architectural fingerprints and identify biases in modeling different frequency ranges [7, 39]. However, to the best of our knowledge, no existing approach has connected spectral analysis with noise scheduling design.

From a complementary perspective, several works have explored diffusion models by assuming a Gaussian target distribution [24, 31]. In [24], the authors leveraged a centered distribution and commutativity properties to derive an exact solution for both the reverse SDE and ODE. Similarly, a closed-form solution to the probability flow ODE provided in [31], which also demonstrated the relation between learned neural scores and their single Gaussian score approximation. Although these analytical solutions provide valuable insights into diffusion dynamics, they are derived in the continuous domain via integration, potentially neglecting discretization effects and missing an explicit connection to the full traversal of the noise schedule.

## 6 Experiments

We turn to empirically validate the schedules obtained by solving the optimization problem, referred to as the *spectral schedule* or *spectral recommendation*. In addition, we examine spectral phenomena arising from the diffusion process and their relation to the schedule structure.

### 6.1 Synthetic Gaussian distribution

In the first set of experiments, we assume a Gaussian data distribution, $\mathbf{x}_0 \sim \mathcal{N}(\boldsymbol{\mu}_0, \boldsymbol{\Sigma}_0)$, where $\mathbf{x}_0 \in \mathbb{R}^d$ and $\boldsymbol{\Sigma}_0$ is a circulant covariance matrix. Although circularity is not essential, it facilitates a direct examination of the frequency components, as $\boldsymbol{\Sigma}_0$ is diagonalized by the DFT, with eigenvalues $\{\lambda_i\}_{i=1}^{d}$ corresponding to its DFT coefficients [9]. Specifically, the covariance is chosen to satisfy $\boldsymbol{\Sigma}_0 = A^T A$ where $A$ is a circulant matrix whose first row is $a = [-l, -l+1/(d-1), \ldots, l-1/(d-1), l]$. The mean vector, $\boldsymbol{\mu}_0$, is chosen to be a constant-value vector, following the stationarity assumption.

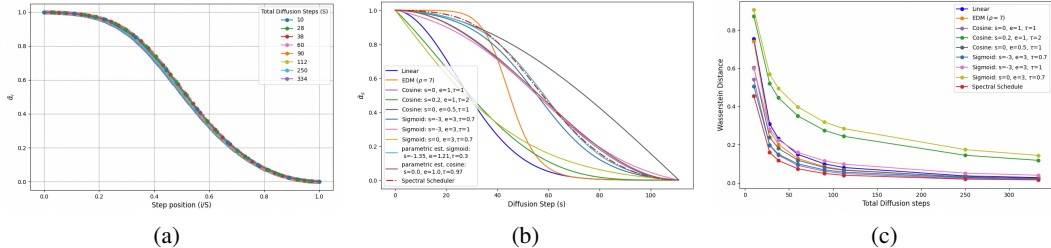

(a)                        (b)                        (c)

Figure 1: Figure 1a presents the optimized spectral schedules for $\mathbf{\Sigma}_0 = A^T A$ with $d = 50, l = 0.1$, and $\boldsymbol{\mu_0} = 0.05 \cdot \mathbf{1}_d$, obtained by minimizing the *Wasserstein-2* distance for various numbers of diffusion steps $S \in [10, 28, 38, 60, 90, 112, 250, 334]$. Figure 1b compares the spectral schedule (dotted red) for $S = 112$ with various heuristic alternatives, including linear, EDM $(\rho = 7)$, Cosine-based schedules such as *Cosine* ($s = 0, e = 1, \tau = 1$) as in [23, 5] and Sigmoid-based schedules like *Sigmoid* ($s = -3, e = 3, \tau = 1$) from [11, 5]. Parametric estimations for the Cosine and Sigmoid schedules appear in brown and cyan, respectively. Figure 1c compares the *Wasserstein-2* distance of the spectral recommendation with that of the baseline schedules across different step counts.

Finding the optimal noise-schedule scheme $\bar{\boldsymbol{\alpha}}^*$ depends on the target signal characteristics $\{\lambda_i\}_{i=1}^d$, the resolution $d$, and the number of diffusion steps applied $S$. Figure 1a shows the resulting noise schedules for $d = 50$, $l = 0.1$ and $\boldsymbol{\mu_0} = 0.05 \cdot \mathbf{1}_d$, obtained by minimizing the *Wasserstein-2* distance for different diffusion steps.[2] Further examples involving different forms of $\mathbf{\Sigma}_0$ and $\boldsymbol{\mu}_0$, as well as the use of the *KL divergence*, are provided in Appendix H.

At first glance, the optimization-based solution produces a noise schedule that aligns with the principles of diffusion models. Specifically, it exhibits a monotonically decreasing behaviour, with linear drop in $\bar{\boldsymbol{\alpha}}^*$ in the middle of the process and minimal variation near the extremes. Although each schedule was independently optimized for a specified number of diffusion steps, it can be observed that the overall structure remains the same.

Interestingly, solving the optimization problem in (14) while altering the initial conditions or removing the inequality constraints yields the same optimal solution. This suggests that these constraints are passive, and that known characteristics of noise schedules, such as monotonicity, naturally arise from the problem formulation, as demonstrated in Appendix G.3.

A key aspect is how the spectral schedule aligns with the existing heuristics. Figure 1b provides a comparison with the Cosine [23], the Sigmoid [11], the linear [10] and the EDM $(\rho = 7)$ [13] schedules, along with a parametric approximation of the spectral recommendation. To achieve this, Cosine and Sigmoid functions were fitted to the optimal solution by minimizing the $l_2$ loss, identifying the closest match. An interesting outcome from Figure 1b is that the spectral recommendation provides a partial retrospective justification for existing noise schedule heuristics, as the parametric estimation resembles Cosine and Sigmoid functions, when their parameters are properly tuned.

To validate the optimization procedure, Figure 1c compares the *Wasserstein-2* distance of various noise schedules with that of the spectral recommendation across different diffusion steps. While the spectral schedule consistently achieves the lowest *Wasserstein-2* distance, the optimization is most effective with fewer diffusion steps, where discretization errors are higher. As the number of steps increases, the gap between the different noise schedules narrows.

We also compare our derived schedules with those proposed in prior work. Specifically, the authors in [26] present a closed-form solution for the optimal noise schedule in a *simplified setting*, assuming an initial isotropic Gaussian distribution with standard deviation $C$, i.e $\mathbf{x}_0 \sim \mathcal{N}(\mathbf{0}, C^2\mathbf{I})$. To enable a fair comparison, we frame our optimization problem using the $D_{\mathrm{KL}}$ loss (4.2) and adopt the Variance-Exploding (VE) framework , following the setup in [26]. Appendix G.2 demonstrates the equivalence between the solutions and highlight their relation to commonly used heuristics.

**Frequencies components and schedule structure:** We turn to explore the relationship between the eigenvalue magnitudes and the structure of the derived schedule. To do so, we solved the optimization problem for each eigenvalue individually, with the contributions from the other eigenvalues set to

---

[2]This follows the principles outlined in [19], ensuring a fair comparison with other noise schedules.

zero. It can be observed from Figure 2a that the solution becomes more concave as the magnitude of the eigenvalue decreases (yellow curve) and more convex as the magnitude increases (blue curve).

Notably, under the shift-invariance conditions, the eigenvalues directly correspond to the system's frequency components. In scenarios where this components follow a monotonically decreasing distribution (e.g., the $1/f$ trend in speech [30]), the first eigenvalues correspond to the low frequencies, having larger amplitudes, while the last correspond to high frequencies and smaller amplitudes. This pattern, along with the previous observation, aligns with the well-known coarse-to-fine signal construction behavior of diffusion models. farther details are provided in Appendix K.1

**Relative error evolution:** While the optimization problem primarily focuses on the output signal $\hat{\mathbf{v}}_0$, each diffusion step $\mathbf{v}_s$ can be also expressed as a function of the initial noise as described in (29) in Appendix C. This enables the analysis of spectral properties across all diffusion steps, rather than only at the output signal. Figure 2b presents the relative error of the 10 largest eigenvalues (sorted, largest on the right) over the final 20 steps of a 60-step diffusion process using the *Cosine* (0,1,1) [23, 5] schedule. While the error consistently decreases for high-amplitude eigenvalues, it increases near the final diffusion steps for those with smaller magnitudes. Additionally, the final relative error (yellow) tends to be smaller for large eigenvalues (i.e., low frequencies). This behavior aligns with the observed bias in mid-to-high frequencies reported in prior works [39, 7]. Further discussion, including an illustration of the eigenvalue and *Wasserstein-2* error dynamics, appears in App. K.2.

**Mean drift:** The explicit expressions in (10) and (12) offer a further insight into the diffusion process. A notable consideration is whether this process introduces a bias, i.e. drifting the mean component during synthesis. To study this, we analyze the mean bias expression for DDIM, $(\mathbf{D}_2 - \mathbf{I})\boldsymbol{\mu}_0^{\boldsymbol{u}}$ derived from the difference between $\mathbb{E}[\mathbf{v}_0]$ and $\mathbb{E}[\hat{\mathbf{v}}_0]$. In Appendix L, we further explore the relationship between the target signal characteristics $\{\lambda_i\}_{i=1}^{d}$, the noise schedule $\bar{\boldsymbol{\alpha}}$, and the expression $|\mathbf{D}_2 - \mathbf{I}|$. It appears that different choices of the noise schedule influence the bias value, with some choices effectively mitigat-

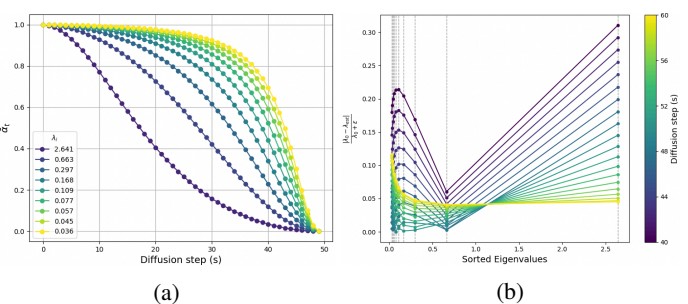

(a)  (b)

Figure 2: Figure 2a shows the spectral schedules obtained by solving the optimization problem individually for each eigenvalue, with other contributions set to zero (note that the reverse process proceeds from right to left). Figure 2b illustrates the relative error of the 10 largest eigenvalues over the final 20 steps of a 60-step diffusion process using the Cosine $(0, 1, 1)$ schedule.

ing it. Additionally, as the depth of the diffusion process increases, the bias value tends to grow, regardless of the selected noise schedule.

**DDPM vs DDIM:** The explicit formulations of DDPM and DDIM in (10) and (12) enable a comparison of their losses across varying diffusion depths and noise schedules. In Appendix M, we present such an evaluation using the *Wasserstein-2* distance. The results clearly show that DDIM sampling is faster and yields lower loss values, aligning with the empirical observations in [27].

## 6.2 Empirical Gaussian distribution

We now shift towards a more practical scenario in which we refer to real data, while still maintaining the Gaussianity assumption. To assess the generality of our approach, we consider different signal types. Specifically, we use CIFAR-10, and the MUSIC dataset [22], which comprises recordings of various musical instruments down-sampled to 16kHz. We fit a Gaussian distribution for each dataset by estimating their mean and covariance matrix, while for the MUSIC dataset, we extracted piano-only recordings. These datasets are referred to hereafter as the *Gaussian MUSIC piano dataset* and the *Gaussian CIFAR-10 dataset*

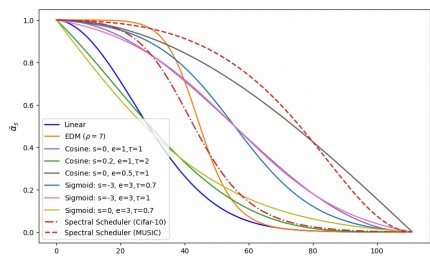

Figure 3: Comparison of the spectral schedules for CIFAR-10 and MUSIC Datasets with various heuristic noise schedules, using 112 diffusion steps.

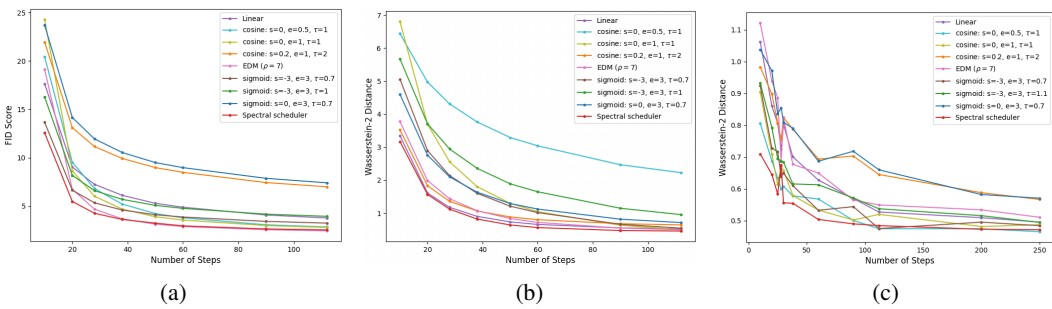

(a)  (b)  (c)

Figure 4: Figures 4b and 4c show the *Wasserstein-2 distance* of the proposed spectral noise schedule (in red) compared to existing heuristics, evaluated across various diffusion step sizes, for CIFAR-10 and MUSIC. Figure 4a presents the corresponding FID scores for CIFAR-10. Across all comparisons, the spectral schedule generally outperforms the heuristics, with a wider gap at lower step counts. For CIFAR-10, the approximation error is less pronounced, with results showing greater stability.

[24]. The estimation procedure for temporal signals, including the role of the window size ($d$) and silence threshold ($th$), is detailed in Appendix I.4.

Figure 3 presents the spectral recommendations obtained by minimizing the *Wasserstein-2* distance for each estimated covariance matrix, alongside several heuristic noise schedules. While both spectral schedules retain some resemblance to the hand-crafted approaches, the one derived for the MUSIC dataset, introduces a somewhat different design of slower decay, adapting to the unique data properties. Consequently, adopting a spectral analysis perspective enables the design of noise schedules tailored to specific needs and data characteristics. The spectral recommendations for the SC09 [33] dataset, with the *Wasserstein-2* evaluations, are provided in Appendix I.

### 6.3 Empirical distribution

We now aim to evaluate whether the optimized noise schedule from Sec. 6.2 remains effective when the Gaussianity assumption is removed and a trained neural denoiser is employed within the diffusion process. Notably, In this setting, approximation error arises due to suboptimality of the denoiser [24].

Our approach towards performance assessment of different noise schedules is the following: We run the diffusion process using a trained denoiser, and the DDIM inference procedure (4) to generate a large corpus of samples. We then compare the statistics of the synthesized signals with those of real data by measuring the distance between their second moments, applying the *Wasserstein-2* distance to the empirical covariance matrices. For CIFAR-10, we also use the standard FID score.

Figure 4 performs such a comparison with various heuristic noise schedules. For each schedule and number of diffusion steps tested, 50,000 signals are synthesized using a trained model. For CIFAR-10, we used a pretrained denoiser from [13] which is based on a continuous noise schedule. For the MUSIC dataset, we trained a model based on the architecture presented in [16, 1], which employs a linear noise schedule with $T = 1000$ diffusion steps during training.

Figure 4 demonstrates that the spectral schedules consistently outperform most heuristic alternatives across both metrics, with the performance gap narrowing as the number of diffusion steps increases and discretization error diminishes. Notably, the Cosine $(0,0.5,1)$ and EDM heuristics, whose structures closely resemble the spectral schedules derived for the CIFAR-10 and MUSIC datasets in Figure 3, also perform competitively (Figures 4a, 4c). These suggests that the proposed spectral recommendation effectively preserves the core data properties and may support the broader design of scheduling strategies. A more detailed comparison, with results for SC09, appears in Appendix J.

## 7 Discussion

By recognizing how the eigenvalues shape the noise schedule, we can refine the loss term to better align with our needs. In Appendix K.3, we introduce a *weighted-l1* loss, which promotes low frequencies while sacrificing high ones, ultimately yielding a solution consistent with the well-known Cosine $(0,1,1)$ heuristic. This relationship can be further explored and may help in mitigating

phenomena such as bias in certain frequency ranges [7, 39]. Another potential research direction is leveraging the evolution of the stepwise distribution for enabling accelerated sampling (29).

We analyze the optimization times of our method. For small step counts, where discretization error is more pronounced, the solutions are typically obtained within a few seconds. However, for higher step counts or high-resolution signals, the computation time may vary. By prioritizing the most significant eigenvalue components during schedule design or initializing the optimization with solutions from smaller step counts, the runtime can be reduced to a few tens of seconds, as detailed in Appendix N.

**Limitations:** While our approach demonstrates the benefits of a principled scheduler design and outperforms heuristic methods in FID across multiple datasets, a natural next step is to extend the framework to settings closer to real data distributions, such as Gaussian mixture models.

This paper presents a spectral perspective on the inference process in diffusion models. Under the assumption of Gaussianity, we establish a direct link between the input white noise and the output signal. Our approach enables noise schedule design based on data characteristics, diffusion steps count, and sampling methods. Effective across synthetic and realistic settings, the optimized schedules resemble existing heuristics, offering insights on handcrafted design choices. By leveraging our method to explore various aspects of the reverse process, we hope this work encourages further exploration of diffusion models and their dynamics from a spectral perspective.

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

# A The Optimal Denoiser for a Gaussian Input

This appendix provides the derivation and explanation of Theorem 3.1.

Let $\mathbf{x}_0 \sim \mathcal{N}(\boldsymbol{\mu}_0, \boldsymbol{\Sigma}_0)$ represent the distribution of the original dataset, where $\mathbf{x}_0 \in \mathbb{R}^d$ . The probability density function $f(\mathbf{x}_0)$ can be written as:

$$f(\mathbf{x}_0) = \frac{1}{\sqrt{(2\pi)^d |\boldsymbol{\Sigma}_0|}} \cdot \exp\left\{ -\frac{1}{2} \left( \bar{\mathbf{x}}_0 - \bar{\boldsymbol{\mu}}_0 \right)^T \boldsymbol{\Sigma}_0^{-1} \left( \bar{\mathbf{x}}_0 - \bar{\boldsymbol{\mu}}_0 \right) \right\}$$

Through the diffusion process, the signal undergoes noise contamination, leading to the following marginal expression for $\mathbf{x}_t$:

$$\mathbf{x}_t = \sqrt{\bar{\alpha}_t} \mathbf{x}_0 + \sqrt{1 - \bar{\alpha}_t} \boldsymbol{\epsilon} \quad \boldsymbol{\epsilon} \sim \mathcal{N}(\mathbf{0}, \mathbf{I}) \tag{17}$$

For the Maximum A Posteriori (MAP) estimation, we seek to maximize the posterior distribution:

$$\max_{x_0} \log p(\mathbf{x}_0 | \mathbf{x}_t)$$

Using Bayes' rule, this can be written as:

$$\min_{x_0} - \log \left[ \frac{p(\mathbf{x}_t | \mathbf{x}_0) p(\mathbf{x}_0)}{p(\mathbf{x}_t)} \right] = \min_{x_0} - \log p(\mathbf{x}_t | \mathbf{x}_0) - \log p(\mathbf{x}_0) \tag{18}$$

The conditional log-likelihood $\log p(\mathbf{x}_t | \mathbf{x}_0)$ is given by:

$$p(\mathbf{x}_t | \mathbf{x}_0) = \frac{1}{\sqrt{(2\pi)^d |\Sigma_1|}} \exp\left\{ -\frac{1}{2} \left( \mathbf{x}_t - \sqrt{\bar{\alpha}_t} \mathbf{x}_0 \right)^T ((1 - \bar{\alpha}_t)\mathbf{I})^{-1} \left( \mathbf{x}_t - \sqrt{\bar{\alpha}_t} \mathbf{x}_0 \right) \right\}$$

$$\log p(\mathbf{x}_t | \mathbf{x}_0) = -\frac{1}{2} \log (2\pi)^d |\Sigma_1| - \frac{1}{2(1 - \bar{\alpha}_t)} \left( \mathbf{x}_t - \sqrt{\bar{\alpha}_t} \mathbf{x}_0 \right)^T \left( \mathbf{x}_t - \sqrt{\bar{\alpha}_t} \mathbf{x}_0 \right)$$

The log-likelihood $\log p(\mathbf{x}_0)$ is given by:

$$p(\mathbf{x}_0) = \frac{1}{\sqrt{(2\pi)^d |\boldsymbol{\Sigma}_0|}} \exp\left\{ -\frac{1}{2} \left( \mathbf{x}_0 - \boldsymbol{\mu}_0 \right)^T \boldsymbol{\Sigma}_0^{-1} \left( \mathbf{x}_0 - \boldsymbol{\mu}_0 \right) \right\}$$

$$\log p(\mathbf{x}_0) = -\frac{1}{2} \log (2\pi)^d |\boldsymbol{\Sigma}_0| - \frac{1}{2} \left( \mathbf{x}_0 - \boldsymbol{\mu}_0 \right)^T \boldsymbol{\Sigma}_0^{-1} \left( \mathbf{x}_0 - \boldsymbol{\mu}_0 \right)$$

We will differentiate the given expression in (18) with respect to $x_0$ and equate it to zero:

$$\frac{d \log p(\mathbf{x}_t | \mathbf{x}_0)}{d \mathbf{x}_0} = \frac{2\sqrt{\bar{\alpha}_t} \left( \mathbf{x}_t - \sqrt{\bar{\alpha}_t} \mathbf{x}_0 \right)}{2(1 - \bar{\alpha}_t)}$$

$$\frac{d \log p(\mathbf{x}_0)}{d \mathbf{x}_0} = -\frac{2\boldsymbol{\Sigma}_0^{-1} \left( \mathbf{x}_0 - \boldsymbol{\mu}_0 \right)}{2}$$

$$\frac{-2\sqrt{\bar{\alpha}_t} \left( \mathbf{x}_t - \sqrt{\bar{\alpha}_t} \mathbf{x}_0 \right)}{2(1 - \bar{\alpha}_t)} + \frac{2\boldsymbol{\Sigma}_0^{-1} \left( \mathbf{x}_0 - \boldsymbol{\mu}_0 \right)}{2} = 0$$

This simplifies to:

$$\frac{-\sqrt{\bar{\alpha}_t} \left( \mathbf{x}_t - \sqrt{\bar{\alpha}_t} \mathbf{x}_0 \right)}{(1 - \bar{\alpha}_t)} + \boldsymbol{\Sigma}_0^{-1} \left( \bar{\mathbf{x}}_0 - \boldsymbol{\mu}_0 \right) = 0$$

Resulting in:

$$-\sqrt{\bar{\alpha}_t} \boldsymbol{\Sigma}_0 \mathbf{x}_t + \bar{\alpha}_t \boldsymbol{\Sigma}_0 \mathbf{x}_0 + (1 - \bar{\alpha}_t) \mathbf{x}_0 - (1 - \bar{\alpha}_t) \boldsymbol{\mu}_0 = 0$$

Thus:

$$\left( \bar{\alpha}_t \boldsymbol{\Sigma}_0 + (1 - \bar{\alpha}_t)\mathbf{I} \right) \mathbf{x}_0 = \sqrt{\bar{\alpha}_t} \boldsymbol{\Sigma}_0 \mathbf{x}_t + (1 - \bar{\alpha}_t) \boldsymbol{\mu}_0$$

Finally:

$$\mathbf{x}_0^* = \left( \bar{\alpha}_t \boldsymbol{\Sigma}_0 + (1 - \bar{\alpha}_t)\mathbf{I} \right)^{-1} \left( \sqrt{\bar{\alpha}_t} \boldsymbol{\Sigma}_0 \mathbf{x}_t + (1 - \bar{\alpha}_t) \boldsymbol{\mu}_0 \right) \tag{19}$$

# B The Reverse Process in the Time Domain

Here, we present the reverse process in the time domain for the DDIM [27], as outlined in Lemma 3.2.

Let $\mathbf{x}_0$ follow the distribution:

$$\mathbf{x}_0 \sim \mathcal{N}(\boldsymbol{\mu}_0, \boldsymbol{\Sigma}_0), \quad \mathbf{x}_0 \in \mathbb{R}^d$$

Using the procedure outline in [27], the diffusion process begins with $\mathbf{x}_S \sim \mathcal{N}(\mathbf{0}, \mathbf{I})$, where $\mathbf{x}_S \in \mathbb{R}^d$ and progresses through an iterative denoising process described as follows:[3]

$$\mathbf{x}_{s-1}(\eta) = \sqrt{\bar{\alpha}_{s-1}} \left( \frac{\mathbf{x}_s - \sqrt{1 - \bar{\alpha}_s} \cdot \boldsymbol{\epsilon}_\theta(\mathbf{x}_s, s)}{\sqrt{\bar{\alpha}_s}} \right) + \sqrt{1 - \bar{\alpha}_{s-1} - \sigma_s^2(\eta)} \cdot \boldsymbol{\epsilon}_\theta(\mathbf{x}_s, s) + \sigma_s(\eta)\mathbf{z}_s \quad (20)$$

where

$$\sigma_s(\eta) = \eta \sqrt{\frac{1 - \bar{\alpha}_{s-1}}{1 - \bar{\alpha}_s}} \sqrt{1 - \frac{\bar{\alpha}_s}{\bar{\alpha}_{s-1}}} \quad (21)$$

Substituting the marginal property from (2):

$$\boldsymbol{\epsilon}_\theta(\mathbf{x}_s, s) = \frac{\mathbf{x}_s - \sqrt{\bar{\alpha}_s}\hat{\mathbf{x}}_0}{\sqrt{1 - \bar{\alpha}_s}} \qquad \hat{\mathbf{x}}_0 = \frac{\mathbf{x}_s - \sqrt{1 - \bar{\alpha}_s} \cdot \boldsymbol{\epsilon}_\theta(\mathbf{x}_s, s)}{\sqrt{\bar{\alpha}_s}} \quad (22)$$

$$\mathbf{x}_{s-1}(\eta) = \sqrt{\bar{\alpha}_{s-1}}\hat{\mathbf{x}}_0 + \sqrt{1 - \bar{\alpha}_{s-1} - \sigma_s^2(\eta)} \left( \frac{\mathbf{x}_s - \sqrt{\bar{\alpha}_s}\hat{\mathbf{x}}_0}{\sqrt{1 - \bar{\alpha}_s}} \right) + \sigma_s(\eta)\mathbf{z}_s \quad (23)$$

For the deterministic scenario, we choose $\eta = 0$ in (21) and obtain $\sigma_s(\eta = 0) = 0$. Therefore:

$$\begin{aligned} \mathbf{x}_{s-1}(\eta = 0) &= \sqrt{\bar{\alpha}_{s-1}}\hat{\mathbf{x}}_0 + \sqrt{1 - \bar{\alpha}_{s-1}} \left( \frac{\mathbf{x}_s - \sqrt{\bar{\alpha}_s}\hat{\mathbf{x}}_0}{\sqrt{1 - \bar{\alpha}_s}} \right) \\ &= \frac{\sqrt{1 - \bar{\alpha}_{s-1}}}{\sqrt{1 - \bar{\alpha}_s}}\mathbf{x}_s + \left[ \sqrt{\bar{\alpha}_{s-1}} - \frac{\sqrt{\bar{\alpha}_s}\sqrt{1 - \bar{\alpha}_{s-1}}}{\sqrt{1 - \bar{\alpha}_s}} \right] \hat{\mathbf{x}}_0 \end{aligned} \quad (24)$$

We denote the following:

$$a_s = \frac{\sqrt{1 - \bar{\alpha}_{s-1}}}{\sqrt{1 - \bar{\alpha}_s}} \qquad b_s = \sqrt{\bar{\alpha}_{s-1}} - \frac{\sqrt{\bar{\alpha}_s}\sqrt{1 - \bar{\alpha}_{s-1}}}{\sqrt{1 - \bar{\alpha}_s}}$$

Therefore we get the following equation:

$$\mathbf{x}_{s-1} = a_s\mathbf{x}_s + b_s\mathbf{x}_0^* .$$

Using the result from the MAP estimator:

$$\mathbf{x}_0^* = (\bar{\alpha}_s\boldsymbol{\Sigma}_0 + (1 - \bar{\alpha}_s)\mathbf{I})^{-1} \left( \sqrt{\bar{\alpha}_s}\boldsymbol{\Sigma}_0\mathbf{x}_t + (1 - \bar{\alpha}_s)\boldsymbol{\mu}_0 \right)$$

we get:

$$\mathbf{x}_{s-1} = a_s\mathbf{x}_s + b_s (\bar{\alpha}_s\boldsymbol{\Sigma}_0 + (1 - \bar{\alpha}_s)\mathbf{I})^{-1} \left( \sqrt{\bar{\alpha}_s}\boldsymbol{\Sigma}_0\mathbf{x}_s + (1 - \bar{\alpha}_s)\boldsymbol{\mu}_0 \right)$$

$$\mathbf{x}_{s-1} = \left( a_s + b_s (\bar{\alpha}_s\boldsymbol{\Sigma}_0 + (1 - \bar{\alpha}_s)\mathbf{I})^{-1} \sqrt{\bar{\alpha}_s}\boldsymbol{\Sigma}_0 \right) \mathbf{x}_s + \left( b_s (\bar{\alpha}_s\boldsymbol{\Sigma}_0 + (1 - \bar{\alpha}_s)\mathbf{I})^{-1} (1 - \bar{\alpha}_s)\boldsymbol{\mu}_0 \right) \quad (25)$$

Introduce the notation:

$$\bar{\boldsymbol{\Sigma}}_{0,s} = \bar{\alpha}_s\boldsymbol{\Sigma}_0 + (1 - \bar{\alpha}_s)\mathbf{I}$$

We can rewrite the equation as:

$$\mathbf{x}_{s-1} = \left( a_s\mathbf{I} + b_s\sqrt{\bar{\alpha}_s} \left( \bar{\boldsymbol{\Sigma}}_{0,s} \right)^{-1} \boldsymbol{\Sigma}_0 \right) \mathbf{x}_s + b_s (1 - \bar{\alpha}_s) \left( \bar{\boldsymbol{\Sigma}}_{0,s} \right)^{-1} \boldsymbol{\mu_0} \quad (26)$$

---

[3]We follow here the DDIM notations that replaces $t$ with $s$, where the steps $[1, \ldots, S]$ form a subsequence of $[1, \ldots, T]$ and $S = T$.

## C Migrating to the Spectral Domain

Here, we demonstrate the application of projecting both sides of Eq. (8) onto the eigenbasis of $\Sigma_0$ as outlined in Lemma 3.4. Since $\Sigma_0$ is symmetric, its eigenvalue decomposition can written as $\Sigma_0 = U\Lambda_0 U^\mathsf{T}$, where the columns of $U$ represent the eigenvectors, and $\Lambda_0$ is the diagonal matrix containing the corresponding eigenvalues.[4]

$$U^\mathsf{T} x_{s-1} = U^\mathsf{T} \left[ \left( a_s I + b_s \sqrt{\bar{\alpha}_s} \bar{\Sigma}_{0,s}^{-1} \Sigma_0 \right) x_s + b_s (1 - \bar{\alpha}_s) \bar{\Sigma}_{0,s}^{-1} \mu_0 \right] \tag{27}$$

$$U^\mathsf{T} x_{s-1} = a_s U^\mathsf{T} x_s + U^\mathsf{T} b_s \sqrt{\bar{\alpha}_s} \bar{\Sigma}_{0,s}^{-1} U U^\mathsf{T} \Sigma_0 U U^\mathsf{T} x_s + U^\mathsf{T} b_s (1 - \bar{\alpha}_s) \bar{\Sigma}_{0,s}^{-1} U U^\mathsf{T} \mu_0$$

By introducing the notations $v = U^\mathsf{T} x$ and $\mu_0^{\mathbf{u}} = U^\mathsf{T} \mu_0$, we obtain:

$$v_{s-1} = a_s v_s + b_s \sqrt{\bar{\alpha}_s} U^\mathsf{T} \bar{\Sigma}_{0,s}^{-1} U U^\mathsf{T} \Sigma_0 U v_s + b_s (1 - \bar{\alpha}_s) U^\mathsf{T} \bar{\Sigma}_{0,s}^{-1} U \mu_0^{\mathbf{u}}$$

By applying eigenvalue decomposition, the symmetric matrix $\Sigma_0$ can be diagonalized using its eigenbasis $U$, where $U^{-1} = U^\mathsf{T}$ Specifically:

- $\Sigma_0 = U\Lambda_0 U^\mathsf{T}, \quad \Lambda_0 = U^\mathsf{T} \Sigma_0 U$
- $a\Sigma_0 + bI = aU\Lambda_0 U^\mathsf{T} + bUIU^\mathsf{T} = U(a\Lambda_0 + bI)U^\mathsf{T}$
- $\Sigma_0^{-1} = U\Lambda_0^{-1} U^\mathsf{T}, \quad \Lambda_0^{-1} = U^\mathsf{T} \Sigma_0^{-1} U$

Therefore, we obtain:

$$\bar{\Sigma}_{0,s}^{-1} = [\bar{\alpha}_s \Sigma_0 + (1 - \bar{\alpha}_s)I]^{-1} = [\bar{\alpha}_s U\Lambda_0 U^\mathsf{T} + (1 - \bar{\alpha}_s)UIU^\mathsf{T}]^{-1} = [U(\bar{\alpha}_s \Lambda_0 + (1 - \bar{\alpha}_s)I)U^\mathsf{T}]^{-1}$$

$$\bar{\Sigma}_{0,s}^{-1} = U[\bar{\alpha}_s \Lambda_0 + (1 - \bar{\alpha}_s)I]^{-1} U^\mathsf{T}$$

Incorporating these elements into the main equation, we obtain:

$$v_{s-1} = a_s v_s + b_s \sqrt{\bar{\alpha}_s} [\bar{\alpha}_s \Lambda_0 + (1 - \bar{\alpha}_s)I]^{-1} \Lambda_0 v_s + b_s (1 - \bar{\alpha}_s) [\bar{\alpha}_s \Lambda_0 + (1 - \bar{\alpha}_s)I]^{-1} \mu_0^{\mathbf{u}}$$

$$v_{s-1} = \left[ a_s I + b_s \sqrt{\bar{\alpha}_s} [\bar{\alpha}_s \Lambda_0 + (1 - \bar{\alpha}_s)I]^{-1} \Lambda_0 \right] v_s + b_s (1 - \bar{\alpha}_s) [\bar{\alpha}_s \Lambda_0 + (1 - \bar{\alpha}_s)I]^{-1} \mu_0^{\mathbf{u}}$$

We will denote the following:

$$G(s) = \left[ a_s + b_s \sqrt{\bar{\alpha}_s} [\bar{\alpha}_s \Lambda_0 + (1 - \bar{\alpha}_s)I]^{-1} \Lambda_0 \right]$$

$$M(s) = b_s (1 - \bar{\alpha}_s) [\bar{\alpha}_s \Lambda_0 + (1 - \bar{\alpha}_s)I]^{-1}$$

and get:

$$v_{s-1} = G(s)v_s + M(s)\mu_0^{\mathbf{u}} \tag{28}$$

We can then recursively obtain $v_l$ for a general $l$:

$$v_l = \left[ \prod_{s'=l+1}^{S} G(s') \right] v_S + \left[ \sum_{i=l+1}^{S} \left( \prod_{j=l+1}^{i-1} G(j) \right) M(i) \right] \mu_0^{\mathbf{u}} \tag{29}$$

---

[4]In case the covariance matrix $\Sigma_0$ is circulant, it can be diagonalized using the Discrete Fourier Transform (DFT) denoted by $\mathcal{F}\{\cdot\}$. In this case, the projection exposes the frequency components, enabling their explicit analysis.

specifically for $l = 0$:

$$\hat{\mathbf{v}}_0 = \left[\prod_{s'=1}^{S} \mathbf{G}(s')\right] \mathbf{v}_S + \left[\sum_{i=1}^{S} \left(\prod_{j=1}^{i-1} \mathbf{G}(j)\right) \mathbf{M}(i)\right] \boldsymbol{\mu}_0^{\mathbf{u}} \tag{30}$$

We will denote the following:

$$\mathbf{D}_1 = \prod_{s=1}^{S} \mathbf{G}(s) = \prod_{s=1}^{S} \left[a_s + b_s \sqrt{\bar{\alpha}_s} \left[\bar{\alpha}_s \boldsymbol{\Lambda}_0 + \mathbf{I}(1 - \bar{\alpha}_s)\right]^{-1} \boldsymbol{\Lambda}_0\right] \tag{31}$$

$$\mathbf{D}_2 = \sum_{i=1}^{S} \left(\prod_{j=1}^{i-1} \mathbf{G}(j)\right) \mathbf{M}(i) \tag{32}$$

$$= \sum_{i=1}^{S} \left[\left(\prod_{j=1}^{i-1} \left[a_j + b_j \sqrt{\bar{\alpha}_j} \left[\bar{\alpha}_j \boldsymbol{\Lambda}_0 + \mathbf{I}(1 - \bar{\alpha}_j)\right]^{-1} \boldsymbol{\Lambda}_0\right]\right) b_i (1 - \bar{\alpha}_i) \left[\bar{\alpha}_i \boldsymbol{\Lambda}_0 + \mathbf{I}(1 - \bar{\alpha}_i)\right]^{-1}\right]$$

Substitute $D_1$ and $D_2$ into the last equation, we get:

$$\hat{\mathbf{v}}_0 = \mathbf{D}_1 \mathbf{v}_S + \mathbf{D}_2 \boldsymbol{\mu}_0^{\mathbf{u}} \tag{33}$$

The resulting vector from Equation 33 is a linear combination of Gaussian signals, therefore it also follows a Gaussian distribution. We now aim to determine the mean and covariance of that distribution.

$$\hat{\mathbf{v}}_0 \sim \mathcal{N}(\mathbb{E}\left[\hat{\mathbf{v}}_0\right], \boldsymbol{\Sigma}_{\hat{\mathbf{v}}_0}), \quad \hat{\mathbf{v}}_0 \in \mathbb{R}^d$$

**Mean:**
$$\mathbb{E}\left[\hat{\mathbf{v}}_0\right] = \mathbb{E}\left[\mathbf{D}_1 \mathbf{v}_S + \mathbf{D}_2 \boldsymbol{\mu}_0^{\mathbf{u}}\right] = \mathbf{D}_1 \mathbf{F}\mathbb{E}\left[\mathbf{v}_S\right] + \mathbb{E}\left[\mathbf{D}_2 \boldsymbol{\mu}_0^{\mathbf{u}}\right] = \mathbf{D}_2 \boldsymbol{\mu}_0^{\mathbf{u}}$$
$$\mathbf{v}_S \sim \mathcal{N}(\mathbf{0}, \mathbf{I})$$
$$\mathbb{E}\left[\hat{\mathbf{v}}_0\right] = \mathbf{D}_2 \boldsymbol{\mu}_0^{\mathbf{u}}$$

**Covariance:**
$$\boldsymbol{\Sigma}_{\hat{\mathbf{v}}_0} = \mathbb{E}\left[\left(\mathbf{D}_1 \mathbf{v}_S + \mathbf{D}_2 \boldsymbol{\mu}_0^{\mathbf{u}} - \mathbb{E}\left[\mathbf{D}_1 \mathbf{v}_S + \mathbf{D}_2 \boldsymbol{\mu}_0^{\mathbf{u}}\right]\right)\left(\mathbf{D}_1 \mathbf{v}_S + \mathbf{D}_2 \boldsymbol{\mu}_0^{\mathbf{u}} - \mathbb{E}\left[\mathbf{D}_1 \mathbf{v}_S + \mathbf{D}_2 \boldsymbol{\mu}_0^{\mathbf{u}}\right]\right)^T\right]$$

$$= \mathbb{E}\left[\left(\mathbf{D}_1 \mathbf{v}_S + \mathbf{D}_2 \boldsymbol{\mu}_0^{\mathbf{u}} - \mathbf{D}_2 \boldsymbol{\mu}_0^{\mathbf{u}}\right)\left(\mathbf{D}_1 \mathbf{v}_S + \mathbf{D}_2 \boldsymbol{\mu}_0^{\mathbf{u}} - \mathbf{D}_2 \boldsymbol{\mu}_0^{\mathbf{u}}\right)^T\right]$$

$$= \mathbb{E}\left[\left(\mathbf{D}_1 \mathbf{v}_S\right)\left(\mathbf{D}_1 \mathbf{v}_S\right)^T\right] = \mathbf{D}_1 \mathbb{E}\left[\mathbf{v}_S \left(\mathbf{v}_S\right)^T\right] \mathbf{D}_1^T$$

$$\boldsymbol{\Sigma}_{\hat{\mathbf{v}}_0} = \mathbf{D}_1 \mathbf{D}_1^T = \mathbf{D}_1^2$$

$$\hat{\mathbf{v}}_0 \sim \mathcal{N}(\mathbf{D}_2 \boldsymbol{\mu}_0^{\mathbf{u}}, \mathbf{D}_1^2), \quad \hat{\mathbf{v}}_0 \in \mathbb{R}^d. \tag{34}$$

The same procedure used to derive the distribution of $\hat{\mathbf{v}}_0$ in (34) from (30) can similarly be applied to any intermediate step $\mathbf{v}_l$ in (29), enabling exploration of the spectral dynamics throughout the inference process.

For the data distribution $\mathbf{x}_0 \sim \mathcal{N}(\boldsymbol{\mu}_0, \boldsymbol{\Sigma}_0)$, where $\mathbf{x}_0 \in \mathbb{R}^d$, its first and second moments in the frequency domain are given by:

$$\mathbb{E}\left[\mathbf{v}_0\right] = \mathbf{U}^T \mathbb{E}\left[\mathbf{x}_0\right] = \boldsymbol{\mu}_0^{\mathbf{u}}$$
$$\Sigma_{\mathbf{v}_0} = \boldsymbol{\Lambda}_0$$

$$\mathbf{v}_0 \sim \mathcal{N}(\boldsymbol{\mu}_0^{\mathbf{u}}, \boldsymbol{\Lambda}_0) \tag{35}$$

## D    Evaluating loss functions expressions

Here, we present selected loss functions based on the derivations provided in Section 3.

### D.1    Wasserstein-2 Distance:

The Wasserstein-2 distance between two Gaussian distributions with means $\mu_1$ and $\mu_2$, and covariance matrices $\Sigma_1$ and $\Sigma_2$, and the corresponding eigenvalues $\{\lambda_1^{(i)}\}_{i=1}^d$ and $\{\lambda_2^{(i)}\}_{i=1}^d$ is given by:

$$W_2(\mathcal{N}_1, \mathcal{N}_2) = \sqrt{(\boldsymbol{\mu}_1 - \boldsymbol{\mu}_2)^T(\boldsymbol{\mu}_1 - \boldsymbol{\mu}_2) + \sum_i \left(\sqrt{\lambda_1^{(i)}} - \sqrt{\lambda_2^{(i)}}\right)^2} \tag{36}$$

Since    $\hat{\mathbf{v}}_0 \sim \mathcal{N}(\mathbf{D}_2\boldsymbol{\mu}_0^{\mathbf{u}}, \mathbf{D}_1^2), \quad \hat{\mathbf{v}}_0 \in \mathbb{R}^d$    and    $\mathbf{v}_0 \sim \mathcal{N}(\boldsymbol{\mu}_0^{\mathbf{u}}, \boldsymbol{\Lambda}_0), \quad \mathbf{v}_0 \in \mathbb{R}^d$

we obtain:

$$(\boldsymbol{\mu}_1 - \boldsymbol{\mu}_2)^T (\boldsymbol{\mu}_1 - \boldsymbol{\mu}_2) = (\mathbf{D}_2\boldsymbol{\mu}_0^{\mathbf{u}} - \boldsymbol{\mu}_0^{\mathbf{u}})^T (\mathbf{D}_2\boldsymbol{\mu}_0^{\mathbf{u}} - \boldsymbol{\mu}_0^{\mathbf{u}}) = \left((\boldsymbol{\mu}_0^{\mathbf{u}})^T\mathbf{D}_2^T - (\boldsymbol{\mu}_0^{\mathbf{u}})^T\right)(\mathbf{D}_2\boldsymbol{\mu}_0^{\mathbf{u}} - \boldsymbol{\mu}_0^{\mathbf{u}})$$

$$= (\boldsymbol{\mu}_0^{\mathbf{u}})^T\boldsymbol{\mu}_0^{\mathbf{u}} - 2(\boldsymbol{\mu}_0^{\mathbf{u}})^T\mathbf{D}_2\boldsymbol{\mu}_0^{\mathbf{u}} + (\boldsymbol{\mu}_0^{\mathbf{u}})^T\mathbf{D}_2^T\mathbf{D}_2\boldsymbol{\mu}_0^{\mathcal{F}}$$

$$= \sum_{i=1}^d (\boldsymbol{\mu}_0^{\mathbf{u}})_i^2 - 2\sum_{i=1}^d (\boldsymbol{\mu}_0^{\mathbf{u}})_i^2 \mathbf{D}_2^{(i)} + \sum_{i=1}^d (\boldsymbol{\mu}_0^{\mathbf{u}})_i^2 \left(\mathbf{D}_2^{(i)}\right)^2$$

$$= \sum_{i=1}^d \left(\mathbf{D}_2^{(i)} - 1\right)^2 (\boldsymbol{\mu}_0^{\mathbf{u}})_i^2$$

$$(\boldsymbol{\mu}_1 - \boldsymbol{\mu}_2)^T (\boldsymbol{\mu}_1 - \boldsymbol{\mu}_2) = \sum_{i=1}^d \left(\mathbf{D}_2^{(i)} - 1\right)^2 (\boldsymbol{\mu}_0^{\mathbf{u}})_i^2$$

$$\sum_i \left(\sqrt{\lambda_1^{(i)}} - \sqrt{\lambda_2^{(i)}}\right)^2 = \sum_i \left(\sqrt{\lambda_0^{(i)}} - \sqrt{\left(\mathbf{D}_1^{(i)}\right)^2}\right)^2$$

$$W_2(\mathbf{v}_0, \hat{\mathbf{v}}_0) = \sqrt{\sum_{i=1}^d \left(d_2^{(i)} - 1\right)^2 (\boldsymbol{\mu}_0^{\mathbf{u}})_i^2 + \sum_i \left(\sqrt{\lambda_0^{(i)}} - \sqrt{\left(\mathbf{D}_1^{(i)}\right)^2}\right)^2} \tag{37}$$

### D.2    Kullback-Leibler divergence:

The Kullback-Leibler (KL) divergence between two Gaussian distributions with means $\mu_1$ and $\mu_2$, and covariance matrices $\Sigma_1$ and $\Sigma_2$, and the corresponding eigenvalues $\{\lambda_1^{(i)}\}_{i=1}^d$ and $\{\lambda_2^{(i)}\}_{i=1}^d$ is given by:

$$D_{\text{KL}}\left(\mathcal{N}(\boldsymbol{\mu}_1, \boldsymbol{\Sigma}_1) \,\|\, \mathcal{N}(\boldsymbol{\mu}_2, \boldsymbol{\Sigma}_2)\right) = \frac{1}{2}\left(\log\frac{|\boldsymbol{\Sigma}_2|}{|\boldsymbol{\Sigma}_1|} - d + \text{tr}\left(\boldsymbol{\Sigma}_2^{-1}\boldsymbol{\Sigma}_1\right) + (\boldsymbol{\mu}_2 - \boldsymbol{\mu}_1)^T\boldsymbol{\Sigma}_2^{-1}(\boldsymbol{\mu}_2 - \boldsymbol{\mu}_1)\right)$$

Given:

$$\hat{\mathbf{v}}_0 \sim \mathcal{N}(\mathbf{D}_2\boldsymbol{\mu}_0^{\mathbf{u}}, \mathbf{D}_1^2), \quad \hat{\mathbf{v}}_0 \in \mathbb{R}^d \quad \text{and} \quad \mathbf{v}_0 \sim \mathcal{N}(\boldsymbol{\mu}_0^{\mathbf{u}}, \boldsymbol{\Lambda}_0), \quad \mathbf{v}_0 \in \mathbb{R}^d$$

The *KL divergence* is given by:

$$D_{\text{KL}}\left(\mathbf{v}_0 \parallel \hat{\mathbf{v}}_0\right) = D_{\text{KL}}\left(\mathcal{N}\left(\mathbb{E}\left[\mathbf{v}_0\right], \mathbf{\Sigma}_{\mathbf{v}_0}\right), \mathcal{N}\left(\mathbb{E}\left[\hat{\mathbf{v}}_0\right], \mathbf{\Sigma}_{\hat{\mathbf{v}}_0}\right)\right)$$

$$= D_{\text{KL}}\left(\mathcal{N}\left(\boldsymbol{\mu}_0^{\mathbf{u}}, \mathbf{\Lambda}_0\right), \mathcal{N}\left(\mathbf{D}_2, \boldsymbol{\mu}_0^{\mathbf{u}}\mathbf{D}_1^2\right)\right)$$

By decomposing the KL divergence elements, we obtain the following terms:

- $|\mathbf{\Sigma}_2| = |\mathbf{D}_1^T\mathbf{D}_1| = |\mathbf{D}_1^2| = \prod_{i=1}^d \mathbf{D}_1^{(i)^2}$
- $|\mathbf{\Sigma}_1| = \prod_{i=1}^d \lambda_0^{(i)}$
- $\text{tr}\left(\mathbf{\Sigma}_2^{-1}\mathbf{\Sigma}_1\right) = \sum_{i=1}^d \frac{\lambda_0^{(i)}}{\mathbf{D}_1^{(i)^2}}$
- $(\boldsymbol{\mu}_2 - \boldsymbol{\mu}_1)^T\mathbf{\Sigma}_2^{-1}(\boldsymbol{\mu}_2 - \boldsymbol{\mu}_1) = (\mathbf{D}_2\boldsymbol{\mu}_0^{\mathbf{u}} - \boldsymbol{\mu}_0^{\mathbf{u}})^T\left(\mathbf{D}_1^2\right)^{-1}(\mathbf{D}_2\boldsymbol{\mu}_0^{\mathbf{u}} - \boldsymbol{\mu}_0^{\mathbf{u}}) =$

$$= (\boldsymbol{\mu}_0^{\mathbf{u}})^T\left(\mathbf{D}_2^T - I\right)\left(\mathbf{D}_1^2\right)^{-1}(\mathbf{D}_2 - I)\boldsymbol{\mu}_0^{\mathbf{u}} = \sum_{i=1}^d \frac{\left(\mathbf{D}_2^{(i)} - 1\right)^2}{\mathbf{D}_1^{(i)^2}}(\boldsymbol{\mu}_{0\,i}^{\mathbf{u}})^2$$

Applying the substitution, the term results in:

$$D_{\text{KL}}\left(\mathcal{N}\left(\boldsymbol{\mu}_0^{\mathbf{u}}, \mathbf{\Lambda}_0\right), \mathcal{N}\left(\mathbf{D}_2\boldsymbol{\mu}_0^{\mathbf{u}}, \mathbf{D}_1^2\right)\right) =$$

$$= \frac{1}{2}\left[\sum_{i=1}^d \log \mathbf{D}_1^{(i)^2} - \sum_{i=1}^d \log \lambda_0^{(i)} - d + \sum_{i=1}^d \frac{\lambda_0^{(i)}}{\mathbf{D}_1^{(i)^2}} + \sum_{i=1}^d \frac{\left(\mathbf{D}_2^{(i)} - 1\right)^2}{\mathbf{D}_1^{(i)^2}}(\boldsymbol{\mu}_{0\,i}^{\mathbf{u}})^2\right]$$

$$D_{\text{KL}}\left(\mathbf{v}_0 \parallel \hat{\mathbf{v}}_0\right) = \frac{1}{2}\left[\sum_{i=1}^d 2\log \mathbf{D}_1^{(i)} - \sum_{i=1}^d \log \lambda_0^{(i)} - d + \sum_{i=1}^d \frac{\lambda_0^{(i)} + \left(\mathbf{D}_2^{(i)} - 1\right)^2(\boldsymbol{\mu}_0^{\mathbf{u}})_i^2}{\mathbf{D}_1^{(i)^2}}\right] \quad (38)$$

## E  DDPM Formulation:

Here, we apply an equivalent procedure to the DDPM scenario, as we did for the DDIM, as outlined in Theorem 3.6.

### E.1  The Reverse Process in the Time Domain

Using the procedure outline in [10], the diffusion process begins with $\mathbf{x}_T \sim \mathcal{N}(\mathbf{0}, \mathbf{I})$, where $\mathbf{x}_T \in \mathbb{R}^d$, and progresses through an iterative denoising process described as follows:

$$\mathbf{x}_{t-1} = \frac{1}{\sqrt{\alpha_t}}\left(\mathbf{x}_t - \frac{1 - \alpha_t}{\sqrt{1 - \bar{\alpha}_t}}\boldsymbol{\epsilon}_\theta(\mathbf{x}_t, t)\right) + \sigma_t \mathbf{z}_t \quad \mathbf{z}_t \sim \mathcal{N}(0, I) \quad (39)$$

Where $\sigma_t = \sqrt{\frac{1 - \bar{\alpha}_{t-1}}{1 - \bar{\alpha}_t}(1 - \alpha_t)}$.

Given the marginal property from (2):

$$\boldsymbol{\epsilon}_\theta(\mathbf{x}_t, t) = \frac{\mathbf{x}_t - \sqrt{\bar{\alpha}_t}\hat{\mathbf{x}}_0}{\sqrt{1 - \bar{\alpha}_t}}$$

we can incorporate it into (39):

$$\mathbf{x}_{t-1} = \frac{1}{\sqrt{\alpha_t}} \left( \mathbf{x}_t - \frac{1 - \alpha_t}{\sqrt{1 - \bar{\alpha}_t}} \left( \frac{\mathbf{x}_t - \sqrt{\bar{\alpha}_t}\hat{\mathbf{x}}_0}{\sqrt{1 - \bar{\alpha}_t}} \right) \right) + \sigma_t \mathbf{z}_t$$

$$\mathbf{x}_{t-1} = \frac{1}{\sqrt{\alpha_t}} \left( \mathbf{x}_t \left( 1 - \frac{1 - \alpha_t}{1 - \bar{\alpha}_t} \right) + \frac{(1 - \alpha_t)\sqrt{\bar{\alpha}_t}}{1 - \bar{\alpha}_t} \hat{\mathbf{x}}_0 \right) + \sigma_t \mathbf{z}_t$$

$$\mathbf{x}_{t-1} = \frac{1}{\sqrt{\alpha_t}} \left( \frac{\alpha_t - \bar{\alpha}_t}{1 - \bar{\alpha}_t} \right) \mathbf{x}_t + \sqrt{\frac{\bar{\alpha}_t}{\alpha_t} \frac{(1 - \alpha_t)}{1 - \bar{\alpha}_t}} \hat{\mathbf{x}}_0 + \sigma_t \mathbf{z}_t \tag{40}$$

We denote the following, where the final term in each equation is represented by $\bar{\alpha}_t$ and $\bar{\alpha}_{t-1}$:

$$a_t = \frac{1}{\sqrt{\alpha_t}} \left( \frac{\alpha_t - \bar{\alpha}_t}{1 - \bar{\alpha}_t} \right) = \frac{\sqrt{\bar{\alpha}_t}}{1 - \bar{\alpha}_t} \left[ \frac{1}{\sqrt{\bar{\alpha}_{t-1}}} - \sqrt{\bar{\alpha}_{t-1}} \right]$$

$$b_t = \sqrt{\frac{\bar{\alpha}_t}{\alpha_t} \frac{(1 - \alpha_t)}{1 - \bar{\alpha}_t}} = \sqrt{\bar{\alpha}_{t-1}} \left( \frac{1 - \frac{\bar{\alpha}_t}{\bar{\alpha}_{t-1}}}{1 - \bar{\alpha}_t} \right)$$

$$c_t = \sigma_t = \sqrt{\frac{1 - \bar{\alpha}_{t-1}}{1 - \bar{\alpha}_t}(1 - \alpha_t)} = \sqrt{\frac{1 - \bar{\alpha}_{t-1}}{1 - \bar{\alpha}_t} \left( 1 - \frac{\bar{\alpha}_t}{\bar{\alpha}_{t-1}} \right)}$$

Therefore we get the following equation:

$$\mathbf{x}_{t-1} = a_t \mathbf{x}_t + b_t \hat{\mathbf{x}}_0 + c_t \mathbf{z}_t$$

Using the result for the MAP estimator from (6):

$$\mathbf{x}_0^* = (\bar{\alpha}_t \boldsymbol{\Sigma}_0 + (1 - \bar{\alpha}_t)\mathbf{I})^{-1} \left( \sqrt{\bar{\alpha}_t} \boldsymbol{\Sigma}_0 \mathbf{x}_t + (1 - \bar{\alpha}_t)\boldsymbol{\mu}_0 \right)$$

we get:

$$\mathbf{x}_{t-1} = a_t \mathbf{x}_t + b_t (\bar{\alpha}_t \boldsymbol{\Sigma}_0 + (1 - \bar{\alpha}_t)\mathbf{I})^{-1} \left( \sqrt{\bar{\alpha}_t} \boldsymbol{\Sigma}_0 \mathbf{x}_t + (1 - \bar{\alpha}_t)\boldsymbol{\mu}_0 \right) + c_t \mathbf{z}_t$$

$$\mathbf{x}_{t-1} = \left( a_t + b_t (\bar{\alpha}_t \boldsymbol{\Sigma}_0 + (1 - \bar{\alpha}_t)\mathbf{I})^{-1} \sqrt{\bar{\alpha}_t} \boldsymbol{\Sigma}_0 \right) \mathbf{x}_t + \left( b_t (\bar{\alpha}_t \boldsymbol{\Sigma}_0 + (1 - \bar{\alpha}_t)\mathbf{I})^{-1} (1 - \bar{\alpha}_t)\boldsymbol{\mu}_0 \right) + c_t \mathbf{z}_t$$

Using the notation from Appendix B:

$$\bar{\boldsymbol{\Sigma}}_{0,t} = \bar{\alpha}_t \boldsymbol{\Sigma}_0 + (1 - \bar{\alpha}_t)\mathbf{I}$$

Thus, we can rewrite the equation as:

$$\mathbf{x}_{t-1} = \left( a_t + b_t \sqrt{\bar{\alpha}_t} \left( \bar{\boldsymbol{\Sigma}}_{0,t} \right)^{-1} \boldsymbol{\Sigma}_0 \right) \mathbf{x}_t + b_t (1 - \bar{\alpha}_t) \left( \bar{\boldsymbol{\Sigma}}_{0,t} \right)^{-1} \boldsymbol{\mu}_0 + c_t \mathbf{z}_t \tag{41}$$

### E.2 Migrating to the Spectral Domain

Next, we project both sides of the Eq. (41) onto the spectral domain. As previously described in Appendix C, this step relies on the eigenvalue decomposition of the covariance matrix $\boldsymbol{\Sigma}_0 = \mathbf{U}\boldsymbol{\Lambda}_0\mathbf{U}^{\mathsf{T}}$ where the columns of $\mathbf{U}$ represent the eigenvectors, and $\boldsymbol{\Lambda}_0$ is the diagonal matrix containing the corresponding eigenvalues.

$$\mathbf{U}^{\mathsf{T}}\mathbf{x}_{t-1} = \mathbf{U}^{\mathsf{T}} \left[ \left( a_t + b_t \sqrt{\bar{\alpha}_t} \left( \bar{\boldsymbol{\Sigma}}_{0,t} \right)^{-1} \boldsymbol{\Sigma}_0 \right) \mathbf{x}_t + b_t (1 - \bar{\alpha}_t) \left( \bar{\boldsymbol{\Sigma}}_{0,t} \right)^{-1} \boldsymbol{\mu}_0 + c_t \mathbf{z}_t \right] \tag{42}$$

$$\mathbf{U}^{\mathsf{T}}\mathbf{x}_{t-1} = a_t \mathbf{U}^{\mathsf{T}}\mathbf{x}_t + \mathbf{U}^{\mathsf{T}} b_t \sqrt{\bar{\alpha}_t} \left( \bar{\boldsymbol{\Sigma}}_{0,t} \right)^{-1} \mathbf{U}\mathbf{U}^{\mathsf{T}}\boldsymbol{\Sigma}_0\mathbf{U}\mathbf{U}^{\mathsf{T}}\mathbf{x}_t + \mathbf{U}^{\mathsf{T}} b_s (1 - \bar{\alpha}_t) \left( \bar{\boldsymbol{\Sigma}}_{0,s} \right)^{-1} \mathbf{U}\mathbf{U}^{\mathsf{T}}\boldsymbol{\mu}_0 + c_t \mathbf{U}^{\mathsf{T}}\mathbf{z}_t$$

By introducing the notations $\mathbf{v} = \mathbf{U}^{\mathsf{T}}\mathbf{x}$ and $\boldsymbol{\mu}_0^{\mathbf{u}} = \mathbf{U}^{\mathsf{T}}\boldsymbol{\mu}_0$, we obtain:

$$\mathbf{v}_{t-1} = a_t\mathbf{v}_t + b_t\sqrt{\bar{\alpha}_t}\mathbf{U}^\mathsf{T}\left(\bar{\boldsymbol{\Sigma}}_{0,t}\right)^{-1}\mathbf{U}\mathbf{U}^\mathsf{T}\boldsymbol{\Sigma}_0\mathbf{U}\mathbf{v}_t + b_t(1-\bar{\alpha}_t)\mathbf{U}^\mathsf{T}\left(\bar{\boldsymbol{\Sigma}}_{0,t}\right)^{-1}\mathbf{U}\boldsymbol{\mu}_0^\mathbf{u} + c_t\mathbf{z}_t^\mathbf{u}$$

By applying eigenvalue decomposition, the symmetric matrix $\boldsymbol{\Sigma}_0$ can be diagonalized using its eigenbasis $\mathbf{U}$, where $\mathbf{U}^{-1} = \mathbf{U}^\mathsf{T}$. Therefore, we obtain:

$$\bar{\boldsymbol{\Sigma}}_{0,s}^{-1} = [\bar{\alpha}_s\boldsymbol{\Sigma}_0 + (1-\bar{\alpha}_s)\mathbf{I}]^{-1} = \left[\bar{\alpha}_s\mathbf{U}\boldsymbol{\Lambda}_0\mathbf{U}^\mathsf{T} + (1-\bar{\alpha}_s)\mathbf{U}\mathbf{I}\mathbf{U}^\mathsf{T}\right]^{-1} = \left[\mathbf{U}\left(\bar{\alpha}_s\boldsymbol{\Lambda}_0 + (1-\bar{\alpha}_s)\mathbf{I}\right)\mathbf{U}^\mathsf{T}\right]^{-1}$$

$$\bar{\boldsymbol{\Sigma}}_{0,s}^{-1} = \mathbf{U}\left[\bar{\alpha}_s\boldsymbol{\Lambda}_0 + (1-\bar{\alpha}_s)\mathbf{I}\right]^{-1}\mathbf{U}^\mathsf{T}$$

Incorporating these elements into the main equation, we obtain:

$$\mathbf{v}_{t-1} = a_t\mathbf{v}_t + b_t\sqrt{\bar{\alpha}_t}\left[\bar{\alpha}_t\boldsymbol{\Lambda}_0 + (1-\bar{\alpha}_t)\mathbf{I}\right]^{-1}\boldsymbol{\Lambda}_0\mathbf{v}_t + b_t(1-\bar{\alpha}_t)\left[\bar{\alpha}_t\boldsymbol{\Lambda}_0 + (1-\bar{\alpha}_t)\mathbf{I}\right]^{-1}\boldsymbol{\mu}_0^\mathbf{u} + c_t\mathbf{z}_t^\mathbf{u}$$

$$\mathbf{v}_{t-1} = \left[a_t\mathbf{I} + b_t\sqrt{\bar{\alpha}_t}\left[\bar{\alpha}_t\boldsymbol{\Lambda}_0 + (1-\bar{\alpha}_t)\mathbf{I}\right]^{-1}\boldsymbol{\Lambda}_0\right]\mathbf{v}_t + b_t(1-\bar{\alpha}_t)\left[\bar{\alpha}_t\boldsymbol{\Lambda}_0 + (1-\bar{\alpha}_t)\mathbf{I}\right]^{-1}\boldsymbol{\mu}_0^\mathbf{u} + c_t\mathbf{z}_t^\mathbf{u}$$

Including those elements in the main equation:

$$\mathbf{v}_{t-1} = a_t\mathbf{v}_t + b_t\sqrt{\bar{\alpha}_t}\left[\bar{\alpha}_t\boldsymbol{\Lambda}_0 + (1-\bar{\alpha}_t)\mathbf{I}\right]^{-1}\boldsymbol{\Lambda}_0\mathbf{v}_t + b_t(1-\bar{\alpha}_t)\left[\bar{\alpha}_t\boldsymbol{\Lambda}_0 + (1-\bar{\alpha}_t)\mathbf{I}\right]^{-1}\boldsymbol{\mu}_0^\mathbf{u} + c_t\mathbf{z}_t^\mathbf{u}$$

$$\mathbf{v}_{t-1} = \left[a_t\mathbf{I} + b_t\sqrt{\bar{\alpha}_t}\left[\bar{\alpha}_t\boldsymbol{\Lambda}_0 + (1-\bar{\alpha}_t)\mathbf{I}\right]^{-1}\boldsymbol{\Lambda}_0\right]\mathbf{v}_t + b_t(1-\bar{\alpha}_t)\left[\bar{\alpha}_t\boldsymbol{\Lambda}_0 + (1-\bar{\alpha}_t)\mathbf{I}\right]^{-1}\boldsymbol{\mu}_0^\mathbf{u} + c_t\mathbf{z}_t^\mathbf{u}$$

We will denote the following:

$$\mathbf{G}(t) = \left[a_t + b_t\sqrt{\bar{\alpha}_t}\left[\bar{\alpha}_t\boldsymbol{\Lambda}_0 + (1-\bar{\alpha}_t)\mathbf{I}\right]^{-1}\boldsymbol{\Lambda}_0\right]$$

$$\mathbf{M}(t) = b_t(1-\bar{\alpha}_t)\left[\bar{\alpha}_t\boldsymbol{\Lambda}_0 + (1-\bar{\alpha}_t)\mathbf{I}\right]^{-1}$$

and get:

$$\mathbf{v}_{t-1} = \mathbf{G}(t)\mathbf{v}_t + \mathbf{M}(t)\boldsymbol{\mu}_0^\mathbf{u} + c_t\mathbf{z}_t^\mathbf{u}$$

We can then recursively obtain $\mathbf{v}_l$ for a general $l$:

$$\mathbf{v}_l = \left[\prod_{t'=l+1}^{T}\mathbf{G}(t')\right]\mathbf{v}_t + \left[\sum_{i=l+1}^{T}\left(\prod_{j=l+1}^{i-1}\mathbf{G}(j)\right)\mathbf{M}(i)\right]\boldsymbol{\mu}_0^\mathbf{u} + \left[\sum_{i=l+1}^{T}\left(\prod_{j=l+1}^{i-1}\mathbf{G}(j)\right)c_i\mathbf{z}_i^\mathbf{u}\right]$$

where the process iterates over all the steps: $[1,\ldots,T]$.

$$\hat{\mathbf{v}}_0 = \left[\prod_{t'=1}^{T}\mathbf{G}(t')\right]\mathbf{v}_t + \left[\sum_{i=1}^{T}\left(\prod_{j=1}^{i-1}\mathbf{G}(j)\right)c_i\mathbf{z}_i^\mathbf{u}\right] + \left[\sum_{i=1}^{T}\left(\prod_{j=1}^{i-1}\mathbf{G}(j)\right)\mathbf{M}(i)\right]\boldsymbol{\mu}_0^\mathbf{u}$$

We will denote the following:

$$\mathbf{D}_1 = \prod_{s=1}^{S}\mathbf{G}(s)$$

$$\mathbf{D}_2 = \sum_{i=1}^{S}\left(\prod_{j=1}^{i-1}\mathbf{G}(j)\right)\mathbf{M}(i)$$

Substitute $\mathbf{D}_1$ and $\mathbf{D}_2$ into the last equation, we get:

$$\hat{\mathbf{v}}_0 = \mathbf{D}_1\mathbf{v}_t + \left[\sum_{i=1}^{T}\left(\prod_{j=1}^{i-1}\mathbf{G}(j)\right)c_i\mathbf{z}_i^{\mathbf{u}}\right] + \mathbf{D}_2\boldsymbol{\mu}_0^{\mathbf{u}} \tag{43}$$

The resulting vector from Equation 43 is a linear combination of Gaussian signals, therefore it also follows a Gaussian distribution. We now aim to determine the mean vector and the covariance matrix of that distribution.

$$\hat{\mathbf{v}}_0 \sim \mathcal{N}(\mathbb{E}\left[\hat{\mathbf{v}}_0\right], \boldsymbol{\Sigma}_{\hat{\mathbf{v}}_0}), \quad \hat{\mathbf{v}}_0 \in \mathbb{R}^d$$

$$\mathbf{x}_T \sim \mathcal{N}(\mathbf{0},\mathbf{I}) \quad \text{and} \quad \mathbf{z}_i \sim \mathcal{N}(\mathbf{0},\mathbf{I}), \quad \forall i$$

**Mean:**

$$\mathbb{E}\left[\hat{\mathbf{v}}_0\right] = \mathbb{E}\left[\mathbf{D}_1\mathbf{v}_t + \left[\sum_{i=1}^{T}\left(\prod_{j=1}^{i-1}\mathbf{G}(j)\right)c_i\mathbf{z}_i^{\mathbf{u}}\mathbf{I}\right] + \mathbf{D}_2\boldsymbol{\mu}_0^{\mathbf{u}}\right] = \mathbf{D}_2\boldsymbol{\mu}_0^{\mathbf{u}}$$

$$\mathbb{E}\left[\hat{\mathbf{v}}_0\right] = \mathbf{D}_2\boldsymbol{\mu}_0^{\mathbf{u}}$$

**Covariance:**

$$\boldsymbol{\Sigma}_{\hat{\mathbf{v}}_0} = \mathbb{E}[\left(\mathbf{D}_1\mathbf{v}_t + \mathbf{D}_2\boldsymbol{\mu}_0^{\mathbf{u}} + \left[\sum_{i=1}^{T}\left(\prod_{j=1}^{i-1}\mathbf{G}(j)\right)c_i\mathbf{z}_i^{\mathbf{u}}\right] - \mathbb{E}\left[\mathbf{D}_1\mathbf{v}_t + \mathbf{D}_2\boldsymbol{\mu}_0^{\mathbf{u}} + \left[\sum_{i=1}^{T}\left(\prod_{j=1}^{i-1}\mathbf{G}(j)\right)c_i\mathbf{z}_i^{\mathbf{u}}\right]\right]\right)$$

$$\left(\mathbf{D}_1\mathbf{v}_t + \mathbf{D}_2\boldsymbol{\mu}_0^{\mathbf{u}} + \left[\sum_{i=1}^{T}\left(\prod_{j=1}^{i-1}\mathbf{G}(j)\right)c_i\mathbf{z}_i^{\mathbf{u}}\right] - \mathbb{E}\left[\mathbf{D}_1\mathbf{v}_t + \mathbf{D}_2\boldsymbol{\mu}_0^{\mathbf{u}} + \left[\sum_{i=1}^{T}\left(\prod_{j=1}^{i-1}\mathbf{G}(j)\right)c_i\mathbf{z}_i^{\mathbf{u}}\right]\right]\right)^T \ ]$$

$$\boldsymbol{\Sigma}_{\hat{\mathbf{v}}_0} = \mathbb{E}\left[\left(\mathbf{D}_1\mathbf{v}_t + \left[\sum_{i=1}^{T}\left(\prod_{j=1}^{i-1}\mathbf{G}(j)\right)c_i\mathbf{z}_i^{\mathbf{u}}\right]\right)\left(\mathbf{D}_1\mathbf{v}_t + \left[\sum_{i=1}^{T}\left(\prod_{j=1}^{i-1}\mathbf{G}(j)\right)c_i\mathbf{z}_i^{\mathbf{u}}\right]\right)^T\right]$$

**Lemma E.1.** *Let* $\mathbf{x}_1, \mathbf{x}_2, \ldots, \mathbf{x}_n$ *be* $n$ *independent Gaussian random vectors with mean* $\mathbb{E}[\mathbf{x}_i] = \mathbf{0}$ *and covariance matrices* $Cov(\mathbf{x}_i) = \boldsymbol{\Sigma}_i$*, for* $i = 1, 2, \ldots, n$*. Let the linear combination be defined as:*

$$\mathbf{y} = \sum_{i=1}^{n} a_i\mathbf{x}_i,$$

*where* $a_1, a_2, \ldots, a_n$ *are constants. The covariance matrix of* $\mathbf{y}$*, denoted as* $Cov(\mathbf{y})$*, is given by:*

$$Cov(\mathbf{y}) = \sum_{i=1}^{n} a_i^2\boldsymbol{\Sigma}_i.$$

Applying the result of Lemma E.1 to the expression in Equation 44, where $\mathbf{x}_T \sim \mathcal{N}(\mathbf{0},\mathbf{I})$ and $\mathbf{z}_i \sim \mathcal{N}(\mathbf{0},\mathbf{I})$ are independent Gaussian noises for all $i$, we have:

$$\mathbb{E}\left[\mathbf{v}_t\mathbf{v}_t^T\right] = \mathbb{E}\left[\mathbf{U}^\mathsf{T}\mathbf{x}_T\mathbf{x}_T^T\mathbf{U}\right] = \mathbf{U}^\mathsf{T}\mathbb{E}\left[\mathbf{x}_T\mathbf{x}_T^T\right]\mathbf{U} = \mathbf{I}$$

$$\mathbb{E}\left[\mathbf{z}_i^{\mathbf{u}}\mathbf{z}_i^{\mathbf{u}T}\right] = \mathbf{I}$$

Thus, the covariance is given by:

$$\boldsymbol{\Sigma}_{\hat{\mathbf{v}}_0} = \mathbb{E}\left[\hat{\mathbf{v}}_0\hat{\mathbf{v}}_0^T\right] = [\mathbf{D}_1]^2 + \sum_{i=1}^{T}\left[\left(\prod_{j=1}^{i-1}\mathbf{G}(j)\right)c_i\mathbf{I}\right]^2$$

$$\Sigma_{\hat{\mathbf{v}}_0} = [\mathbf{D}_1]^2 + \sum_{i=1}^{T} \left[ \left( \prod_{j=1}^{i-1} \mathbf{G}(j) \right) c_i \mathbf{I} \right]^2$$

$$\hat{\mathbf{v}}_0 \sim \mathcal{N}\left( \mathbf{D}_2 \boldsymbol{\mu}_0^{\mathbf{u}} \,,\, \mathbf{D}_1^2 + \sum_{i=1}^{T} \left( \prod_{j=1}^{i-1} \mathbf{G}^2(j) \right) c_i^2 \mathbf{I} \right) \tag{44}$$

As discussed in Appendix C, for a data distribution $\mathbf{x}_0 \sim \mathcal{N}(\boldsymbol{\mu}_0, \boldsymbol{\Sigma}_0)$, where $\mathbf{x}_0 \in \mathbb{R}^d$, the first and second moments in the frequency domain are given by:

$$\mathbb{E}\left[\mathbf{v}_0\right] = \mathbf{F}\mathbb{E}\left[\mathbf{x}_0\right] = \boldsymbol{\mu}_0^{\mathbf{u}}$$
$$\boldsymbol{\Sigma}_{\mathbf{x}_0^{\mathcal{F}}} = \boldsymbol{\Lambda}_0$$

$$\mathbf{v}_0 \sim \mathcal{N}(\boldsymbol{\mu}_0^{\mathbf{u}}, \boldsymbol{\Lambda}_0) \tag{45}$$

## F    Variance preserving and Variance exploding theoretical analysis

The paper [28] distinguishes between two sampling methods: *Variance Preserving* (VP) and *Variance Exploding* (VE). The primary difference lies in how variance evolves during the process. while VP maintains a fixed variance, VE results in an exploding variance as $t \to T$. Here, we focus on comparing these approaches within the context of our spectral noise schedule derivation for the DDIM procedure [10]. Throughout this paper, we described our methods based on the *Variance Preserving* (VP) formulation, given by:

$$p(\mathbf{x}_t|\mathbf{x}_0) = \mathcal{N} \sim \left( \sqrt{\bar{\alpha}_t}\mathbf{x}_0, \sqrt{1 - \bar{\alpha}_t}\mathbf{I} \right) \tag{46}$$

where the only hyperparameters are the noise schedule parameters: $\{\bar{\alpha}_s\}_{s=0}^{S}$ where $\bar{\alpha}_s \in (0, 1]$.

In contrast, under the Variance Exploding (VE) method, the hyperparameters are given by $\sigma_t$ where $\sigma_t \in [0, \infty)$, and the marginal distribution takes the form:

$$p(\bar{\mathbf{x}}_t|\mathbf{x}_0) = \mathcal{N} \sim (\bar{\mathbf{x}}_0, \sigma_t^2\mathbf{I}) \tag{47}$$

We used the notation $\bar{\mathbf{x}}_t$ to distinguish it from $\mathbf{x}_t$, except in the special case where $\mathbf{x}_0 = \bar{\mathbf{x}}_0$. Applying the reparameterization trick, we obtain:

$$\text{VP: } \mathbf{x}_t = \sqrt{\bar{\alpha}_t}\mathbf{x}_0 + \sqrt{1 - \bar{\alpha}_t}\boldsymbol{\epsilon} \quad \boldsymbol{\epsilon} \sim \mathcal{N}(\mathbf{0}, \boldsymbol{I}) \tag{48}$$

$$\text{VE: } \bar{\mathbf{x}}_t = \bar{\mathbf{x}}_0 + \sigma_t\boldsymbol{\epsilon} \quad \boldsymbol{\epsilon} \sim \mathcal{N}(\mathbf{0}, \boldsymbol{I}) \tag{49}$$

A key relationship between the VP and VE formulations, as derived in [14], is given by:

$$\bar{\mathbf{x}}_t = \frac{\mathbf{x}_t}{\sqrt{1 + \sigma_t^2}} \qquad \bar{\alpha}_t = \frac{1}{(1 + \sigma_t^2)} \tag{50}$$

### F.1    Determining the Optimal Denoiser:

Following the derivation in A, we obtained the expression for the optimal denoiser in the Gaussian case under the Variance Preserving (VP) formulation:

$$\hat{\mathbf{x}}_0^{\text{wiener,VP}} = \left( \bar{\alpha}_t \boldsymbol{\Sigma}_0 + I \left( 1 - \bar{\alpha}_t \right) \right)^{-1} \left( \sqrt{\bar{\alpha}_t} \boldsymbol{\Sigma}_0 \mathbf{x}_t + \left( 1 - \bar{\alpha}_t \right) \boldsymbol{\mu}_0 \right).$$

Leveraging a similar approach, we derive the corresponding expression for the Variance Exploding (VE) scenario:

$$\hat{\mathbf{x}}_0^{\text{wiener,VE}} = \left(\boldsymbol{\Sigma}_0 + I\sigma_t^2\right)^{-1}\left(\boldsymbol{\Sigma}_0\bar{\mathbf{x}}_t + \sigma_t^2\boldsymbol{\mu}_0\right). \tag{51}$$

## F.2 Evaluating the Inference Process in the time Domain:

This part can be performed using two equivalent methods:

**Method 1:**

The ODE for the VE scenario in DDIM, as outlined in [27], is given by:

$$d\bar{x} = -\frac{1}{2}g(t)^2\nabla_{\bar{x}}\log p_t(\bar{x})dt \quad g(t) = \sqrt{\frac{d\sigma^2(t)}{dt}} \tag{52}$$

Additionally, the score expression and the marginal equation are also derived in [27] as follows:

$$\nabla_{\bar{x}}\log p_t(\bar{x}) = -\frac{\epsilon_\theta^{(t)}}{\sigma(t)} \tag{53}$$

$$\bar{x}_t = \bar{x}_0 + \sigma_t\epsilon \quad where \quad \epsilon \sim \mathcal{N}(\mathbf{0}, \mathbf{I}) \tag{54}$$

Substituting Equation 53 into Equation 52:

$$d\bar{x} = \frac{1}{2}\frac{d\sigma^2(t)}{dt}\frac{\epsilon_\theta^{(t)}}{\sigma(t)}dt$$

$$\frac{d\bar{x}}{dt} = \frac{d\sigma(t)}{dt}\epsilon_\theta^{(t)} \tag{55}$$

Substituting Equation 54 into Equation 55, we obtain:

$$\bar{x}_t - \bar{x}_{t-1} = (\sigma_t - \sigma_{t-1})\frac{\bar{x}_t - \bar{x}_0}{\sigma_t}$$

$$\bar{x}_{t-1} = \bar{x}_t + \left(\frac{\sigma_{t-1}}{\sigma_t} - 1\right)\bar{x}_t - \bar{x}_0$$

$$\bar{x}_{t-1} = \frac{\sigma_{t-1}}{\sigma_t}\bar{x}_t + \left(1 - \frac{\sigma_{t-1}}{\sigma_t}\right)\bar{x}_0 \tag{56}$$

**Method 2:**

given the inference process in the VP formulation [27]:[5]

$$\mathbf{x}_{s-1}(\eta = 0) = \frac{\sqrt{1 - \bar{\alpha}_{s-1}}}{\sqrt{1 - \bar{\alpha}_s}}\mathbf{x}_s + \left[\sqrt{\bar{\alpha}_{s-1}} - \frac{\sqrt{\bar{\alpha}_s}\sqrt{1 - \bar{\alpha}_{s-1}}}{\sqrt{1 - \bar{\alpha}_s}}\right]\hat{\mathbf{x}}_0$$

By leveraging the connections in Equation 50 we can derive the following relationship between the two successive steps, $\mathbf{x}_{s-1}$ and $\mathbf{x}_s$, in the inference process:

$$\bar{x}_{s-1} = \sqrt{\frac{\sigma_{s-1}^2}{\sigma_s^2}}\bar{x}_s + \left(1 - \sqrt{\frac{\sigma_{s-1}^2}{\sigma_s^2}}\right)x_0 \tag{57}$$

The Resulted expressions in (57) and (56) are identical.

By defining the following terms:

$$\bar{a}_s = \frac{\sigma_{s-1}}{\sigma_s}$$

---

[5]We follow here the DDIM notations that replaces $t$ with $s$, where the steps $[1, \ldots, S]$ form a subsequence of $[1, \ldots, T]$ and $S = T$.

$$\bar{b}_s = 1 - \frac{\sigma_{s-1}}{\sigma_s} = 1 - \bar{a}_s$$

we can express the relationship between $\mathbf{x}_{s-1}$ and $\mathbf{x}_s$ as:

$$\mathbf{x}_{s-1} = a_s \mathbf{x}_s + b_s \hat{\mathbf{x}}_0 \tag{58}$$

## F.3 Evaluating the Inference Process in the Spectral Domain

Since a similar expression to Equation 58 has already been discussed in Appendix C, we can now describe the inference process in the spectral domain as follows:

$$\mathbf{v}_{s-1} = \mathbf{G}(s)\mathbf{v}_s + \mathbf{M}(s)\boldsymbol{\mu}_0^{\mathbf{u}} \tag{59}$$

where:

$$\mathbf{G}(s) = \left[ \bar{a}_s + \bar{b}_s \left[ \boldsymbol{\Lambda}_0 + \mathbf{I}\sigma_s^2 \right]^{-1} \boldsymbol{\Lambda}_0 \right]$$

$$\mathbf{M}(s) = \bar{b}_s \left[ \boldsymbol{\Lambda}_0 + \mathbf{I}\sigma_s^2 \right]^{-1} \sigma_s^2$$

Following this and in alignment with the same methodology described in Appendix C we obtain:

$$\hat{\mathbf{v}}_0 = \mathbf{D}_1 \mathbf{v}_s + \mathbf{D}_2 \boldsymbol{\mu}_0^{\mathbf{u}} \tag{60}$$

$$\hat{\mathbf{v}}_0 \sim \mathcal{N}(\mathbb{E}\left[\hat{\mathbf{v}}_0\right], \boldsymbol{\Sigma}_{\hat{\mathbf{v}}_0}), \quad \hat{\mathbf{v}}_0 \in \mathbb{R}^d$$

$$\mathbb{E}\left[\hat{\mathbf{v}}_0\right] = \mathbf{D}_2 \boldsymbol{\mu}_0^{\mathbf{u}}, \quad \boldsymbol{\Sigma}_{\hat{\mathbf{v}}_0} = \mathbf{D}_1 \mathbf{D}_1^T = \mathbf{D}_1^2$$

$$\mathbf{D}_1 = \prod_{s=1}^{S} \mathbf{G}(s) = \prod_{s=1}^{S} \left[ a_s + b_s \sqrt{\bar{\alpha}_s} \left[ \bar{\alpha}_s \boldsymbol{\Lambda}_0 + \mathbf{I}(1 - \bar{\alpha}_s) \right]^{-1} \boldsymbol{\Lambda}_0 \right] \tag{61}$$

$$\mathbf{D}_2 = \sum_{i=1}^{S} \left( \prod_{j=1}^{i-1} \mathbf{G}(j) \right) \mathbf{M}(i) \tag{62}$$

$$= \sum_{i=1}^{S} \left[ \left( \prod_{j=1}^{i-1} \left[ a_j + b_j \sqrt{\bar{\alpha}_j} \left[ \bar{\alpha}_j \boldsymbol{\Lambda}_0 + \mathbf{I}(1 - \bar{\alpha}_j) \right]^{-1} \boldsymbol{\Lambda}_0 \right] \right) b_i (1 - \bar{\alpha}_i) \left[ \bar{\alpha}_i \boldsymbol{\Lambda}_0 + \mathbf{I}(1 - \bar{\alpha}_i) \right]^{-1} \right] \tag{63}$$

# G   Clarifications and Validations:

## G.1   Method Evaluations: Temporal and Spectral Domains

We evaluated the compatibility between the diffusion process in the time domain, using the DDIM method [10], and its counterpart derived from Equation 10 in the spectral domain. Using an artificial covariance matrix, $\Sigma_0$, with parameters $l = 0.1$ and $d = 50$ from 6.1, we estimated the covariance of $50,000$ signals that were denoised according to Equation 4, using the optimal denoiser from Equation 6, and computed their eigenvalues, denoted as $\{\lambda_i^{time}\}_{i=1}^d$. In the spectral domain, we applied the formulation from Equation 10 for deriving $D_1^2$ and extracted $\{\lambda_i^{spectral}\}_{i=1}^d$ from its diagonal elements. The results are illustrated in Figure 5.

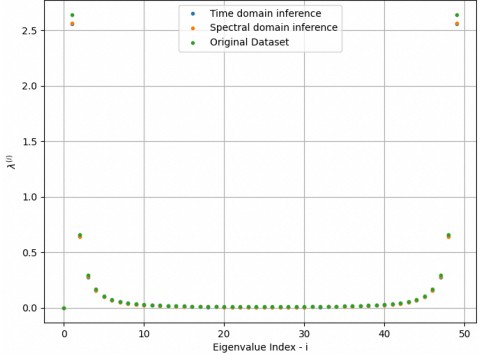
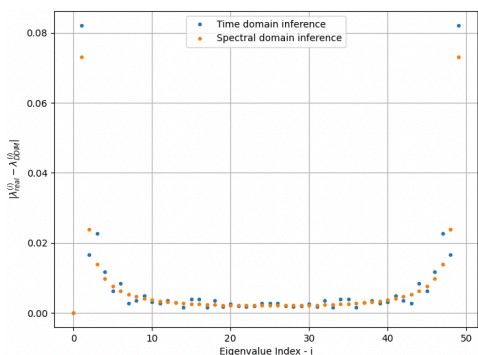

(a) A comparison of the eigenvalues obtained from each method

(b) The absolute relative error between the estimated eigenvalues and the data eigenvalues.

Figure 5: Figure 5a compares the eigenvalues derived from the spectral and time domain formulations of the DDIM method [10]. The dataset, described in 6.1 with $l = 0.1$ and d $= 50$, is used for both approaches, involving 112 diffusion steps and following the linear noise schedule proposed in [27]. Furthermore, Figure 5b illustrates the absolute error between the estimated and original eigenvalues for both methods.

Figure 5a shows that the derived eigenvalues from both procedures align with each other, thus verifying the transition from the time to spectral domain. However, they are not necessarily identical to the properties of the original dataset. Notably, as the number of steps increases, both processes converge toward the original dataset values. Figure 5b allows for an examination of the absolute error in each process relative to the characteristics of the original dataset. It is evident that while the spectral equation exhibits stable behavior, the time-domain equation displays fluctuations that depend on the number of sampled signals. As the number of samples increases, these fluctuations diminish.

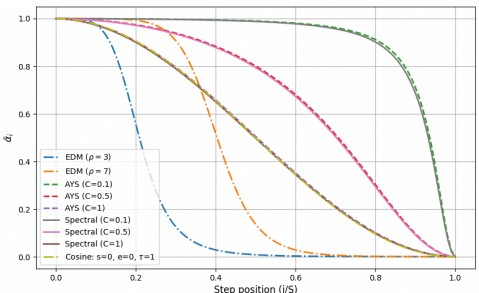

Figure 6: Comparison between the closed-form solution from AYS [26] and our numerical solution for the simplified case where $\mathbf{x}_0 \sim \mathcal{N}(\mathbf{0}, C^2\mathbf{I})$ with $C = [0.1, 0.5, 1]$.

## G.2 Method Evaluation: Comparison with Prior Works

We also compare our optimal solution with those from previous works. Specifically, The authors in [26] derive a closed-form expression for the optimal noise schedule under a *simplified case*, where the initial distribution is an isotropic Gaussian with a standard deviation of $C$, *i.e* $\mathbf{x}_0 \sim \mathcal{N}(\mathbf{0}, C^2\mathbf{I})$. To enable a proper comparison, we frame our optimization problem using the *Kullback-Leibler divergence* $D_{\mathrm{KL}}$ loss (4.2) as done in [26].

Figure 6 compares our optimal solution, obtained by numerically solving Equation (14), with the closed-form solution from [26]. [6] It can be observed that both methods align for arbitrary values of $C$. Notably, for $C = 1$, both noise schedules converge exactly to the *Cosine* $(0, 1, 1)$ noise schedule, which was originally *chosen heuristically* [23].

## G.3 Constraint Omission and Different Types of Initializations

We explore the influence of the initializations and the inequality constraints in the optimization problem. Figures 7 and 8 illustrate the evolution of the noise schedule parameter, $\{\bar{\alpha}_s\}_{s=0}^S$, during the optimization process for two different initializations: *uniformly random* and *linearly decreasing* schedules, respectively. In both cases, the diffusion process consists of $28$ steps, and the *Wasserstein-2* distance is used. Each scenario was conducted twice: once with the inequality constraints from 4 and once without. The results are plotted at $15$ evenly spaced intervals throughout the process to avoid presenting each individual optimization step.

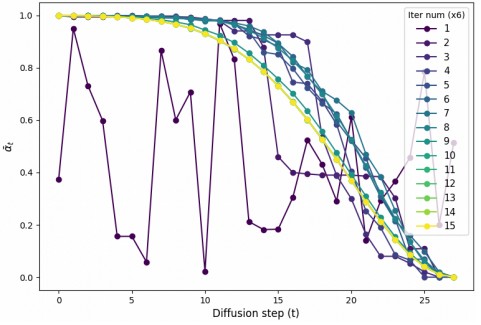

(a) Optimization with inequality constraints.

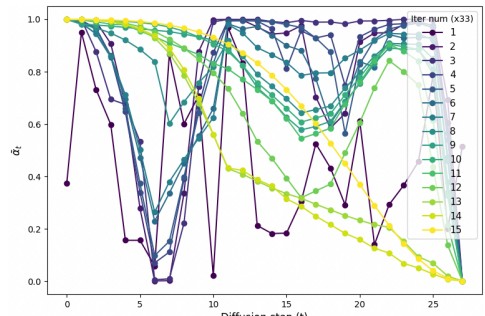

(b) Optimization without inequality constraints.

Figure 7: A Comparison of the noise schedule parameters, $\{\bar{\alpha}_s\}_{s=0}^S$, during the optimization process. The optimization was conducted over $28$ diffusion steps, with a uniformly random initialization. Figure 7a shows the results with inequality constraints, and Figure 7b presents those without.

---

[6]Since the variance-exploding (VE) formulation of the diffusion process was employed in [26], we used the corresponding relationship $\bar{\alpha}_t = \frac{1}{1+\sigma_t^2}$ to transition the resulting noise schedule to the *variance-preserving (VP)* formulation, as derived in Appendix F.

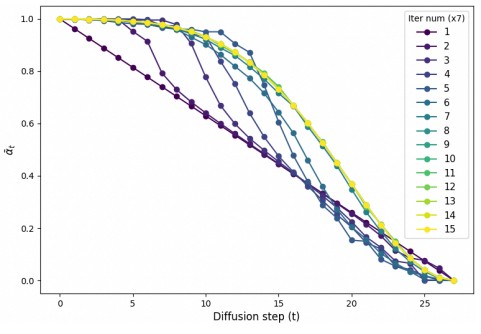
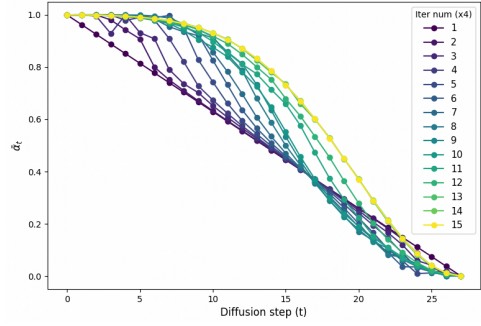

(a) Optimization with inequality constraints.   (b) Optimization without inequality constraints.

Figure 8: A Comparison of the noise schedule parameters, $\{\bar{\alpha}_s\}_{s=0}^{S}$, during the optimization process. The optimization was conducted over 28 diffusion steps, with a Linearly decreasing initialization. Figure 8a shows the results with inequality constraints, and Figure 8b presents those without.

The results reveal that the optimized schedule is consistent across both initializations and independent of the inequality constraints. This suggests that known characteristics of noise schedules, such as monotonicity, naturally emerge from the problem's formulation itself, even without an explicit demand for inequality constraints. Similar consistency is observed for other initializations, including linear and cosine schedules, demonstrating the stability of the optimization procedure.

## H  Supplementary Experiments for Synthetic Gaussian Distribution:

In the following sections, we demonstrate the received spectral recommendations for various alternative selections, applied to the matrix $\mathbf{\Sigma}_0$ and the vector $\boldsymbol{\mu}_0$, differing from those presented in Section 6.1. Additionally, we present the solutions obtained for defining the *Wasserstein-2* distance and the *KL-divergence*. Through this, we aim to provide a broader perspective on the behavior and applicability of the proposed approach.

Figure 9 visualizes the covariance matrices $\mathbf{\Sigma}_0$ and $\mathbf{\Lambda}_0$ as discussed in Sec. 6.1.

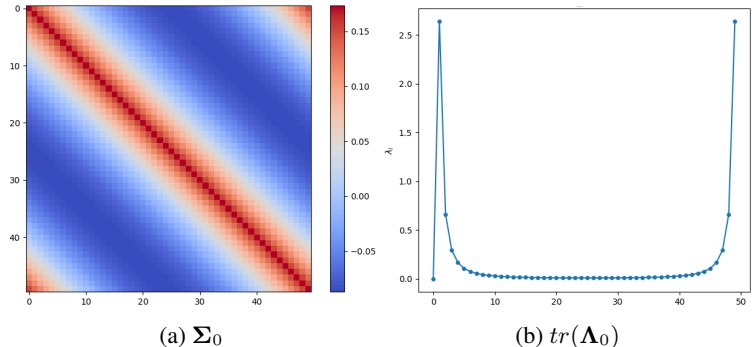

(a) $\mathbf{\Sigma}_0$          (b) $tr(\mathbf{\Lambda}_0)$

Figure 9: Visualization of $\mathbf{\Sigma}_0$ and the trace of $\mathbf{\Lambda}_0$ for $d = 50$ and $l = 0.1$. The covariance matrix $\mathbf{\Sigma}_0$ is circulant (9a), while $\mathbf{\Lambda}_0$ is diagonal with symmetric diagonal elements (9b).

### H.1  Wasserstein-2 distance

Figure 10 presents the resulting noise schedule based on the *Wasserstein-2* distance.

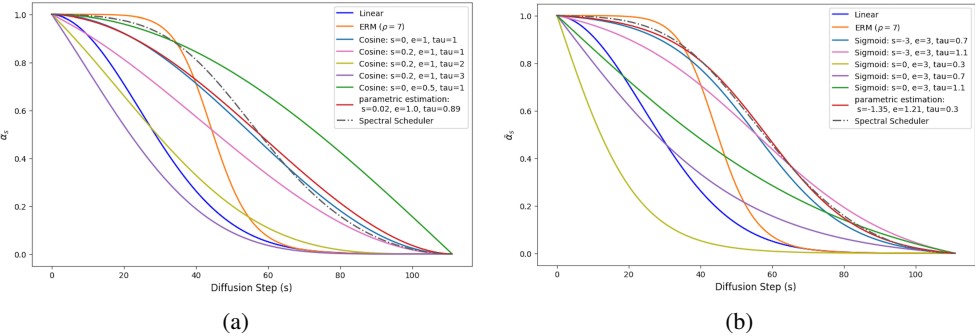

(a)          (b)

Figure 10: Figures 10a and 10b compare the optimized noise schedules from Sec. 6.1, using $d = 50$, $l = 0.1$, and $\boldsymbol{\mu}_0 = 0.05$, with heuristic schedules for 112 diffusion steps, where the optimization is based on the *Wasserstein-2* distance. Figure 10a examines the spectral schedule alongside the Linear [10], EDM [13] and Cosine-based schedules, including *Cosine* ($s = 0$, $e = 1$, $\tau = 1$) from [23, 5]. Likewise, Figure 10b compares it to Sigmoid-based schedules [11]. The parametric estimations for the Cosine and Sigmoid functions are highlighted in red.

## H.2 KL-Divergence

Figure 11 presents the resulting noise schedule based on the *KL-Divergence*.

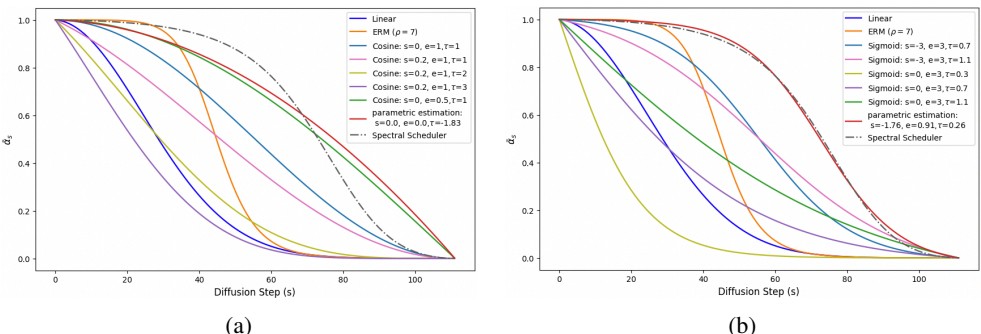

(a)            (b)

Figure 11: Figures 11a and 11b compare the optimized noise schedules from Sec. 6.1, using $d = 50$, $l = 0.1$, and $\boldsymbol{\mu}_0 = 0.05$, with heuristic schedules for 112 diffusion steps, where the optimization is based on the *KL divergence*. Figure 11a examines the spectral schedule alongside the Linear [10], EDM [13] and Cosine-based schedules, including *Cosine* ($s = 0$, $e = 1$, $\tau = 1$) from [23, 5]. Likewise, Figure 10b compares it to Sigmoid-based schedules [11]. The parametric estimations for the Cosine and Sigmoid functions are highlighted in red.

Notably, under the same conditions, the *KL divergence* results in a more *concave* spectral recommendation compared to the *Wasserstein-2* distance.

## H.3 Variations in Covariance Matrices and Mean Configurations

In 6.1, we designed a specific covariance matrix $\boldsymbol{\Sigma}_0$ and a mean vector $\boldsymbol{\mu}_0$ with the intention of resembling characteristics observed in real signals, such as a centered signal with $\boldsymbol{\mu}_0 \approx \mathbf{0}$. However, the optimization process is not restricted to these particular choices and can be generalized to accommodate various alternative decisions. Figure 12 displays different covariance matrices along with their corresponding $\boldsymbol{\mu}_0$ vectors, followed by the resulting spectral schedules computed using the *Wasserstein-2* distance for $60$ diffusion steps.

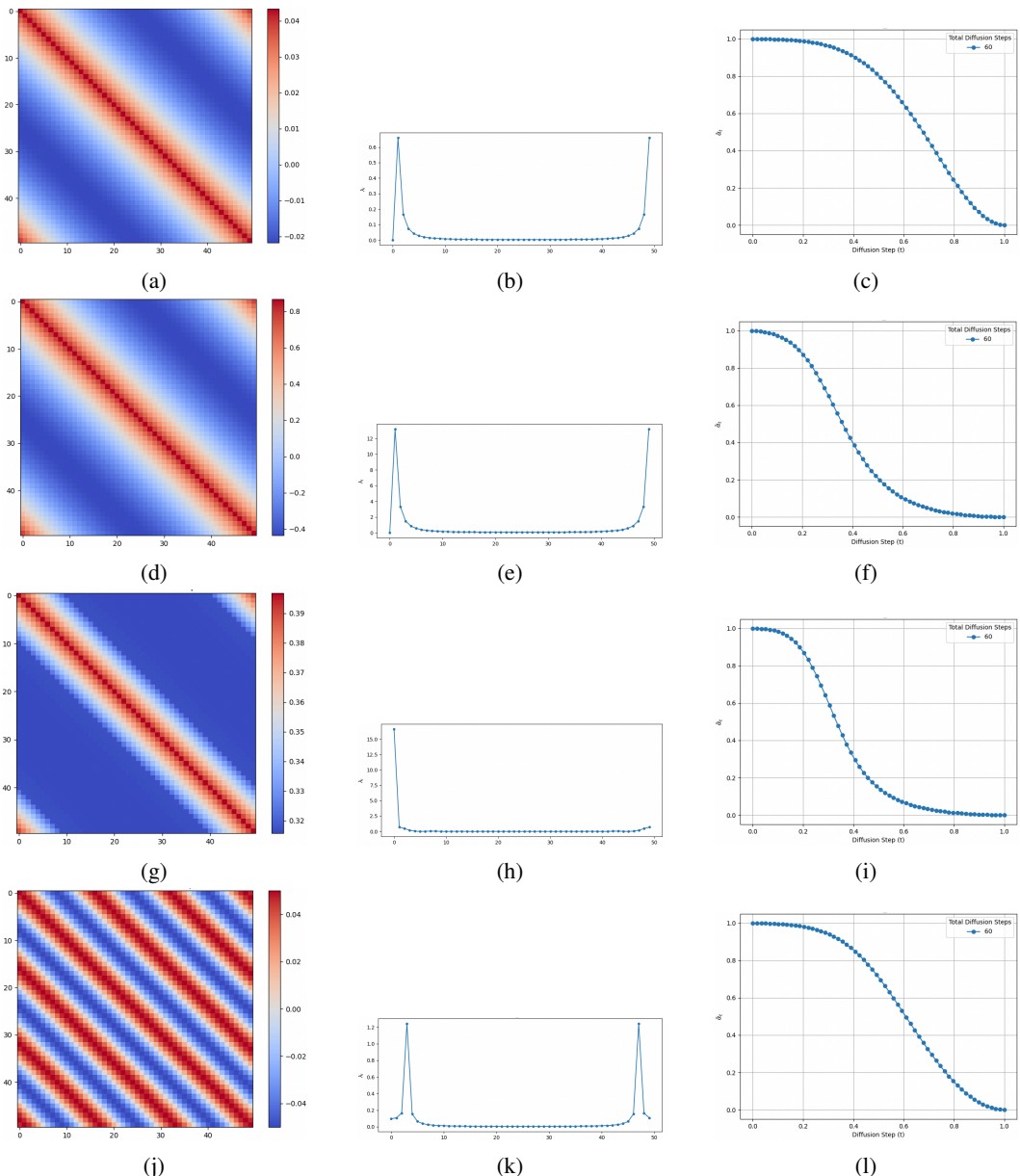

Figure 12: Various covariance matrices (first column) along with their eigenvalues (second column) and the corresponding optimized noise schedules (third column). The first row presents the matrix from 6.1 with $l = 0.05$, $\boldsymbol{\mu}_0 = 0.01 \cdot \mathbf{1}_d$. The second row shows the same matrix scaled by a factor of 20 while keeping the same $\boldsymbol{\mu}_0$. The third row displays a covariance matrix incorporating a Cosine function in the first row $a$ of the circulant covariance, with $\boldsymbol{\mu}_0 = 0.3 \cdot \mathbf{1}_d$. The fourth row features a circulant matrix derived from a sinusoidal signal with a frequency of $1000$ in the first row $a$ of the circulant covariance matrix, scaled by 0.01, and with $\boldsymbol{\mu}_0 = 0.1 \cdot \mathbf{1}_d$.

While we cannot cover all possible choices for the covariance matrix $\boldsymbol{\Sigma}_0$ and the vector $\boldsymbol{\mu}_0$, we aim to provide a broader perspective on the *KL-divergence* loss. Figure 13 illustrates a circulant covariance matrix whose first row is derived from a sinusoidal signal with a frequency of 1000 Hz, along with the corresponding spectral recommendation based on *KL-divergence*, where $\boldsymbol{\mu}_0 = 0.1 \cdot \mathbf{1}_d$.

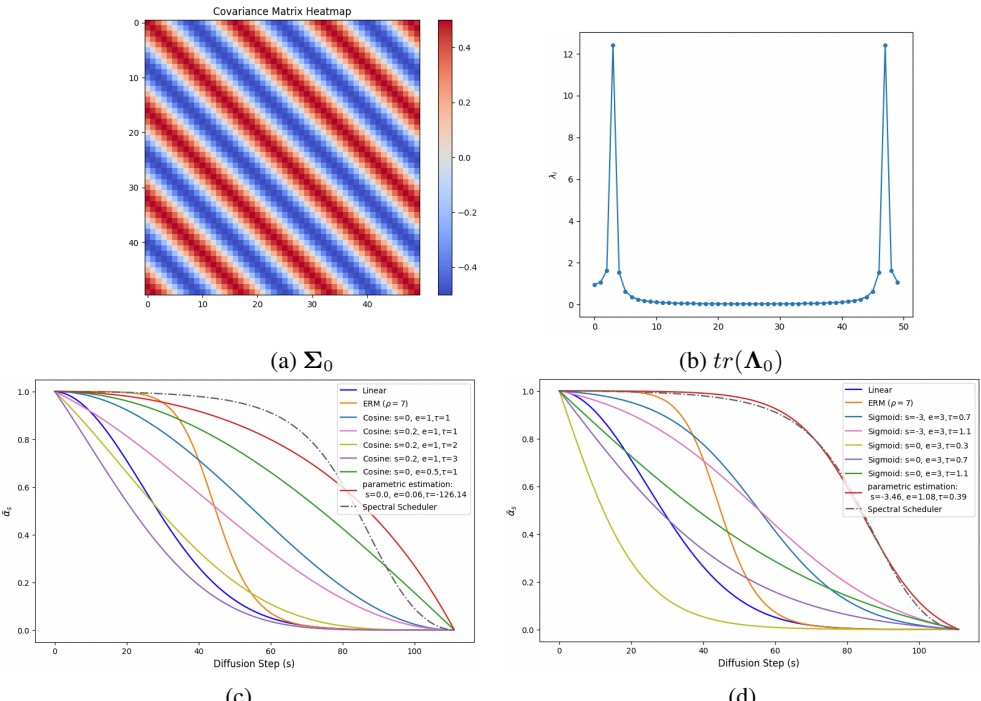

(a) $\boldsymbol{\Sigma}_0$  (b) $tr(\boldsymbol{\Lambda}_0)$

(c)  (d)

Figure 13: Figure 13a shows the circulant covariance matrix, $\boldsymbol{\Sigma}_0$, whose first row is derived from a sinusoidal signal with a frequency of 1000 Hz. Figure 13b displays the trace of the corresponding $\boldsymbol{\Lambda}_0$ matrix. Figures 13c and 13d compare the spectral recommendations for $d = 50$, 112 diffusion steps, using the *KL-divergence*, with various noise schedule heuristics including Cosine and Sigmoid, respectively. The parametric estimations for the Cosine and Sigmoid are highlighted in red.

The results above show that modifying the dataset properties, such as the covariance matrix and expectation, along with altering the loss function, leads to noise schedules with a similar overall structure but varying details. In Appendix K, we explore the connection between the dataset properties, the loss function, and the resulting noise schedules.

# I  Supplementary Experiments for Empirical Gaussian Distribution

We present additional details on the CIFAR-10, Gaussian MUSIC piano and SC09 datasets, along with the spectral noise schedules derived from them [22, 33].

## I.1  CIFAR-10 Dataset

The estimated covariance matrix for CIFAR-10 has a dimension of $d = 3072$, corresponding to the original image resolution of $32 \times 32$ with three color channels. Figure 14a presents the spectral schedule derived for this covariance matrix using a 112-step diffusion process. Notably, its structure resembles that of the EDM schedule. Figure 14b illustrates the Wasserstein-2 distance evaluated on the Gaussian distribution associated with CIFAR-10, demonstrating the optimality of the spectral schedule across all diffusion steps. This effect is especially pronounced at lower step counts, where discretization error is more significant. Interestingly, heuristic schedules whose structures closely align with the spectral schedule, such as EDM, also achieve comparable performance, further supporting the connection between schedule structure and diffusion quality.

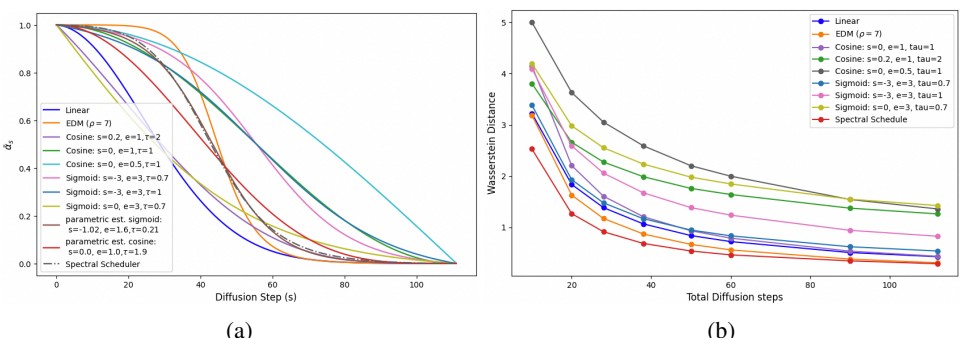

(a)                                                    (b)

Figure 14: Figure 14a compares the spectral schedule with several heuristic noise schedules for a 112-step diffusion process. The best-fitting parametric approximations for the Cosine and Sigmoid schedules are shown in red and brown, respectively. Figure 14b demonstrates the optimality of the spectral recommendation in the Gaussian case, revealing higher effectiveness in diffusion processes with fewer steps, where discretization error is more pronounced.

## I.2  MUSIC Dataset

To estimate the covariance matrix for the MUSIC dataset, we apply a sliding window of length $d = 400$ (0.025 seconds) to the original audio, excluding segments with $L_1$ energy below a threshold of $th = 0.05$ to reduce the influence of silent regions. The resulting covariance matrix is symmetric and exhibits an approximately Toeplitz structure.

Figure 15a presents the spectral schedule derived from the estimated covariance matrix using a 112-step diffusion process, while Figure 15b shows the Wasserstein-2 distance comparison for the MUSIC Piano Gaussian dataset. Similar conclusions regarding optimality and the alignment with heuristic schedules hold here as well, with the Cosine $(0, 0.5, 1)$ schedule being the closest in both structure and performance to the spectral recommendation.

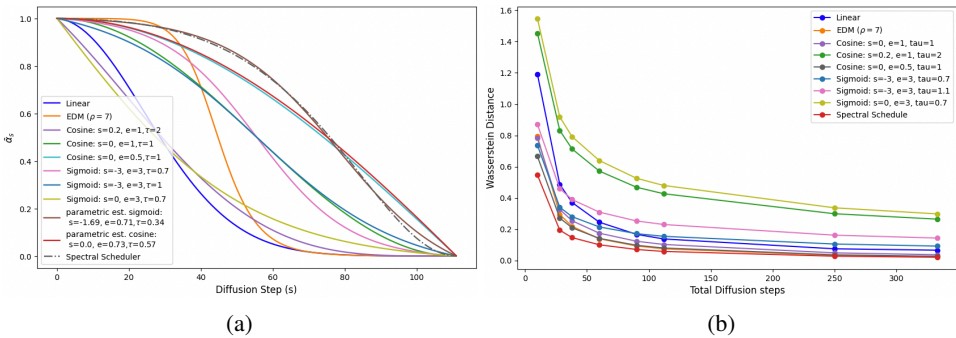

(a)                            (b)

Figure 15: Figure 15a compares the spectral schedule with several heuristic noise schedules for a 112-step diffusion process. The best-fitting parametric approximations for the Cosine and Sigmoid schedules are shown in red and brown, respectively. Figure 15b demonstrates the optimality of the spectral recommendation in the Gaussian case, revealing higher effectiveness in diffusion processes with fewer steps, where discretization error is more pronounced.

### I.3 SC09 Dataset

In this section, we apply our method to a different dataset, SC09. SC09 is a subset of the *Speech Commands Dataset* [33] and consists of spoken digits (0–9). Each audio sample has a duration of one second and is recorded at a sampling rate of 16 kHz.

Differing from Sec. 6.2, here we use segments of length $d = 16000$ samples (one second) and set $th = 0.05$ in one setting and $0.1$ in another. Figure 16 presents the spectral recommendations for $th = 0.05$ and $th = 0.1$ in the left and right columns, respectively.

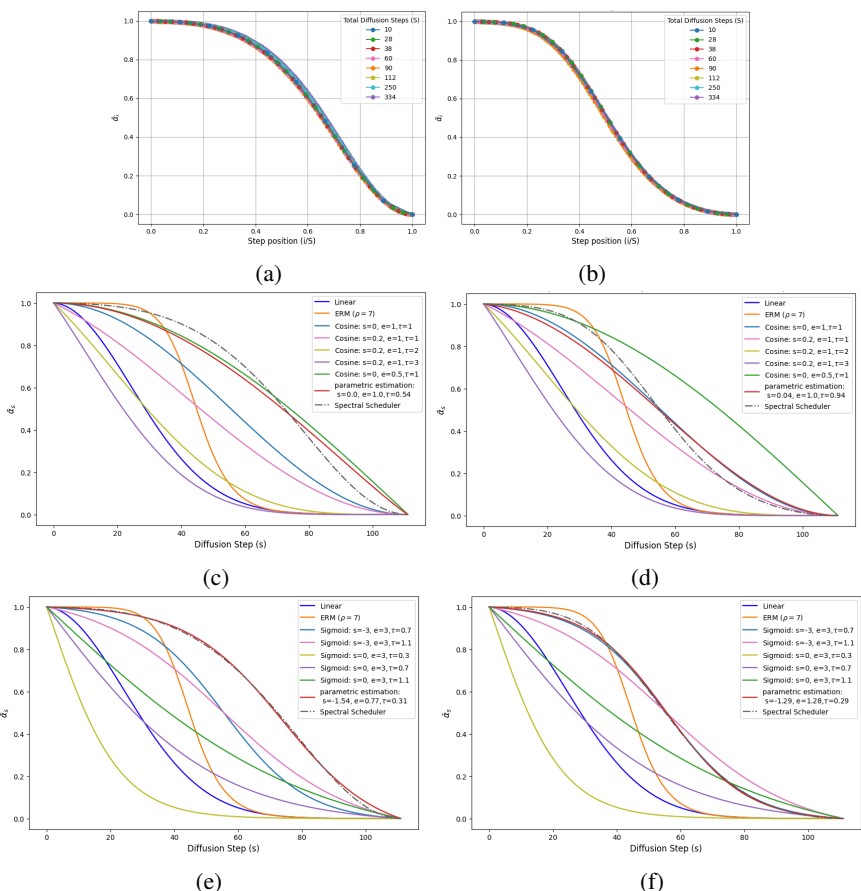

Figure 16: Spectral noise schedules for the $SC09$ dataset with $d = 16,000$ and thresholds $th = 0.05$ (left column) and $th = 0.1$ (right column). The first row shows spectral recommendations for various diffusion steps, while the second and third rows compare the spectral recommendation for 112 steps with heuristic noise schedules, including the Cosine and Sigmoid schedules. The parametric estimations for the Cosine and Sigmoid functions are shown in red, respectively.

Figures 16a and 16b demonstrate that the spectral recommendation for $th = 0.05$ exhibits a more concave behavior compared to $th = 0.1$. A more detailed discussion on the influence of the $th$ and $d$ parameters on the covariance matrix and the resulting noise schedule is provided in Appendices I.4 and K.

We also evaluate the Wasserstein-2 distance for the spectral recommendation on the SC09 Gaussian dataset, using a silence threshold of $th = 0.1$. As shown in Figure 17, similar conclusions hold: the spectral schedule remains optimal, with Cosine $(0, 1, 1)$ and EDM heuristics most closely matching its structure and performance.

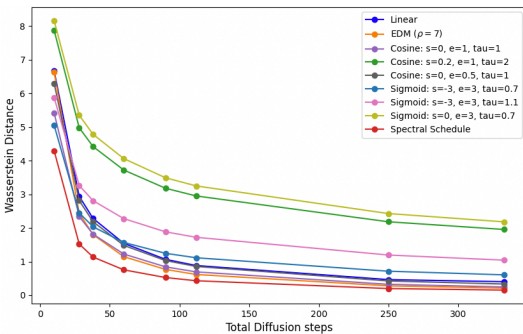

Figure 17: Wasserstein-2 distance comparison across different numbers of diffusion steps and various heuristic noise schedules for the SC09 Gaussian dataset. The results highlight the optimality of the spectral recommendation, showing higher effectiveness in diffusion processes with fewer steps, where discretization error is more pronounced.

### I.4 Analysis of Different Aspects

The estimation of the covariance matrix, which is essential for finding the spectral recommendation for a real dataset, relies on the choice of two key parameters: $th$ and $d$.

The parameter $d$ represents the dimension of the signals and controls the frequency resolution, which affects the eigenvalues. A smaller $d$ may result in a more generalized eigenvalue spectrum, reducing accuracy by averaging energy across neighboring eigenvalues. In contrast, a larger $d$ improves the precision in capturing frequency details but increases computational time for both estimation and optimization.

Figure 18 shows that as $d$ increases, the eigenvalue structure becomes more precise, with the maximum eigenvalues magnitude growing larger. Conversely, as $d$ decreases, the eigenvalue structure becomes more generalized, exhibiting a monotonic decrease, as discussed in Appendix K.3.

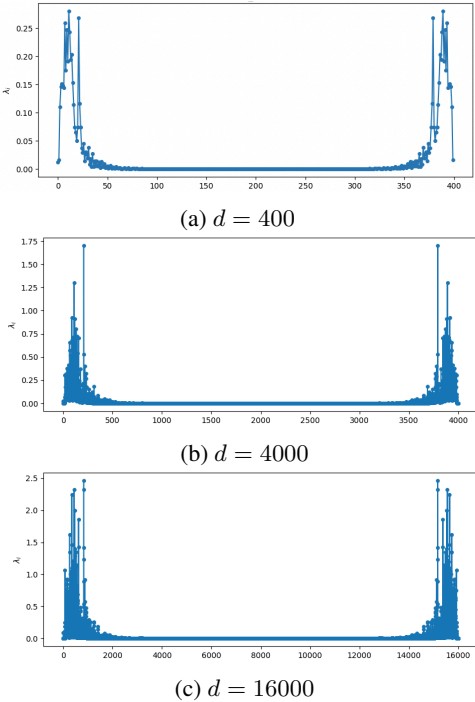

(a) $d = 400$

(b) $d = 4000$

(c) $d = 16000$

Figure 18: Eigenvalues of the MUSIC dataset using $d = [400, 4000, 16000]$

An additional consideration is the choice of the threshold value $th$. This threshold helps prevent the covariance matrix estimation from being overly influenced by silent regions in the signal, which are characterized by low $L_1$ energy. Adjusting $th$ affects both the covariance matrix values and the eigenvalues, i.e. $C \cdot Av = C \cdot \lambda v$, thereby influencing the resulting noise schedule, as shown in Appendix K.1.

## J  Supplementary Experiments for Empirical Distribution

In the following section, we present a more comprehensive analysis of the results for the general empirical distribution, where the Gaussianity assumption no longer holds and a neural denoiser is required.

We begin by providing technical details of the denoiser architectures used for each dataset. For CIFAR-10, we employed a pretrained denoiser from [13], trained with a continuous noise schedule. This approach facilitates a better comparison across noise schedules by reducing approximation error and ensuring alignment between training and testing conditions.

For the MUSIC dataset, we trained a model based on the architecture proposed in [16, 1], using a linear noise schedule with $T = 1000$ diffusion steps. Training was performed in an unconditional setting on raw waveforms with a batch size of $64$. For the SC09 dataset, we adopted the architecture from GitHub, and similarly trained it with a linear schedule of $T = 1000$ steps.

In the both cases, during synthesis, we mapped the tested noise schedule to the closest matching step in the linear schedule to enable use of the trained denoiser. While this approach is suboptimal compared to training with a continuous noise schedule, introducing additional approximation error, it reflects a practical scenario where a specific denoiser is given, and is therefore worthwhile to examine.

We assess the performance of the spectral recommendation on the MUSIC and SC09 datasets using second-order metric. In particular, we compute the *Wasserstein-2* distance between the covariance matrices estimated from $20,000$ generated samples at each diffusion step and those estimated from the real data. This allows us to evaluate how well each noise schedule preserves the core properties of the distribution during synthesis (Figure 20)

Additionally, for CIFAR-10, we use the standard FID metric to assess whether perceptual properties are preserved with our recommendation (see Figures 19). The FID score is calculated following the method outlined in [13], using $50,000$ samples for each computation, with seeds ranging from $0$ to $49,999$. As for FAD [15] calculation, it is less reliable for the MUSIC dataset due to its short frame lengths, and it provides limited insight for SC09, given its nature as a speech dataset.

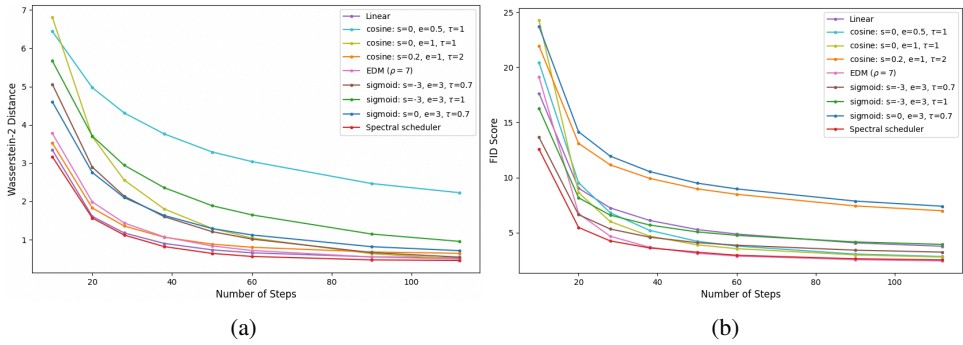

(a)                    (b)

Figure 19: *Wasserstein-2* distance (Figure 19a) and FID (Figure 19b) results on the CIFAR-10 dataset.

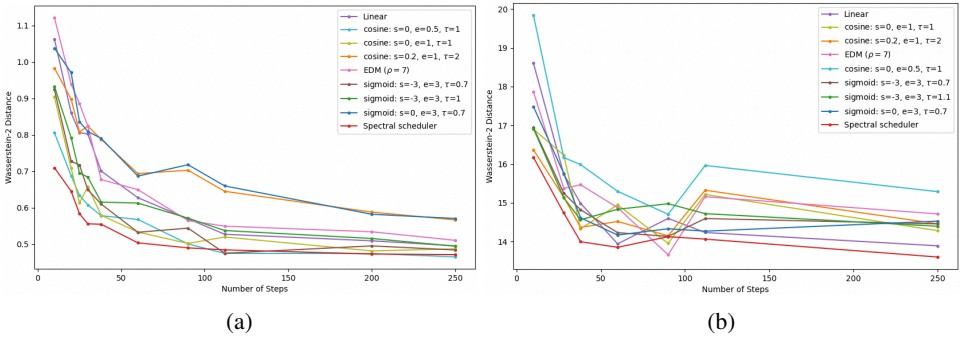

(a)                    (b)

Figure 20: *Wasserstein-2* distance on the MUSIC-piano dataset (Figure 20a) and SC09 dataset (Figure 20b), evaluated over diffusion steps: $\{10, 20, 28, 38, 60, 90, 112, 250\}$.

The results support the relevance of the spectral recommendation for a broader range of distributions. The improved stability observed on CIFAR-10 may be attributed to factors such as the model architecture, the training procedure, or differences between discrete and continuous noise schedules. Interestingly, heuristics that achieve similar performance, such as EDM for FID on the CIFAR-10 dataset and cosine $(0, 0.5, 1)$ schedules for the Wasserstein distance on the MUSIC dataset, tend to share structural similarities with the spectral recommendation. This suggests a connection between the structure of the noise schedule and overall performance, and points to the potential of the proposed method to inform the design of effective noise schedules.

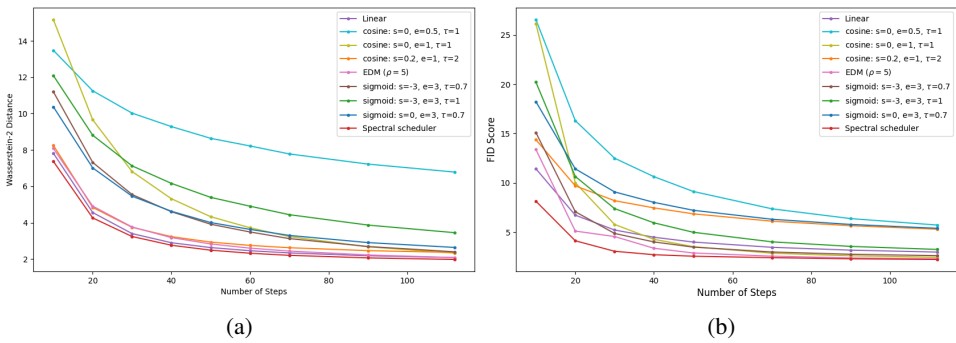

| (a) | (b) |

Figure 21: *Wasserstein-2* distance (Figure 21a) and FID results (Figure 21b) on the AFHQv2 dataset, evaluated over diffusion steps: $\{10, 20, 30, 40, 50, 70, 90, 112\}$.

We conducted an additional experiment on the higher-resolution AFHQv2 dataset [6], with a covariance matrix of size $12,288 \times 12,288$ (corresponding to $64 \times 64 \times 3$ images). Figure 21 presents Wasserstein distance and FID scores for the spectral recommendation and heuristic baselines, demonstrating the superior performance of the spectral schedule. In this case, EDM with $\rho = 5$ most closely approximates the spectral schedule in both structure and FID, reflecting dataset-specific adaptation.

### Experiment statistical significance

We compute the FID and Wasserstein-2 distance for CIFAR-10 using three independent runs, each consisting of 50,000 samples generated with different random seeds ([0–4999], [5000–9999], and [10000–14999]), to capture random variability. The results in Tables 1 and 2 report the Wasserstein-2 distance and FID (respectively), along with a $95\%$ confidence interval, calculated using the standard error and the t-distribution.

Table 1: Wasserstein scores (mean $\pm$ 95% CI) across selected diffusion steps for different noise schedules.

|  | 10 | 20 | 28 | 38 | 50 | 90 |
|---|---|---|---|---|---|---|
| Linear | $3.28 \pm 0.28$ | $1.61 \pm 0.01$ | $1.17 \pm 0.01$ | $0.90 \pm 0.02$ | $0.73 \pm 0.02$ | $0.54 \pm 0.04$ |
| EDM | $3.77 \pm 0.02$ | $1.99 \pm 0.01$ | $1.43 \pm 0.01$ | $1.07 \pm 0.01$ | $0.83 \pm 0.02$ | $0.53 \pm 0.04$ |
| Cosine (0, 1, 1) | $6.81 \pm 0.01$ | $3.69 \pm 0.01$ | $2.56 \pm 0.01$ | $1.80 \pm 0.01$ | $1.30 \pm 0.01$ | $0.63 \pm 0.01$ |
| Cosine (0.2, 1, 2) | $3.52 \pm 0.01$ | $1.83 \pm 0.01$ | $1.35 \pm 0.01$ | $1.06 \pm 0.01$ | $0.88 \pm 0.02$ | $0.67 \pm 0.03$ |
| Cosine (0, 0.5, 1) | $6.44 \pm 0.01$ | $4.97 \pm 0.02$ | $4.31 \pm 0.02$ | $3.76 \pm 0.01$ | $3.29 \pm 0.01$ | $2.46 \pm 0.01$ |
| Sigmoid (0, 3, 0.7) | $4.60 \pm 0.01$ | $2.75 \pm 0.01$ | $2.10 \pm 0.01$ | $1.63 \pm 0.01$ | $1.29 \pm 0.01$ | $0.81 \pm 0.02$ |
| Sigmoid ($-3$, 3, 1) | $5.67 \pm 0.01$ | $3.71 \pm 0.01$ | $2.94 \pm 0.01$ | $2.36 \pm 0.01$ | $1.89 \pm 0.01$ | $1.15 \pm 0.00$ |
| Sigmoid ($-3$, 3, 0.7) | $5.06 \pm 0.01$ | $2.89 \pm 0.01$ | $2.13 \pm 0.01$ | $1.60 \pm 0.01$ | $1.21 \pm 0.00$ | $0.66 \pm 0.02$ |
| Spectral | $\mathbf{3.21 \pm 0.24}$ | $\mathbf{1.59 \pm 0.05}$ | $\mathbf{1.12 \pm 0.01}$ | $\mathbf{0.81 \pm 0.02}$ | $\mathbf{0.63 \pm 0.02}$ | $\mathbf{0.46 \pm 0.04}$ |

Table 2: FID scores (mean $\pm$ 95% CI) across selected diffusion steps for different noise schedules.

|  | 10 | 20 | 28 | 38 | 50 | 90 |
|---|---|---|---|---|---|---|
| Linear | $17.59 \pm 0.26$ | $9.00 \pm 0.17$ | $7.19 \pm 0.25$ | $6.04 \pm 0.26$ | $5.22 \pm 0.25$ | $4.00 \pm 0.26$ |
| EDM | $19.04 \pm 0.25$ | $6.70 \pm 0.17$ | $\underline{4.64 \pm 0.19}$ | $\underline{3.63 \pm 0.22}$ | $\mathbf{3.09 \pm 0.21}$ | $\mathbf{2.52 \pm 0.19}$ |
| Cosine (0, 1, 1) | $24.31 \pm 0.14$ | $8.63 \pm 0.16$ | $6.00 \pm 0.16$ | $4.65 \pm 0.18$ | $3.86 \pm 0.20$ | $2.94 \pm 0.21$ |
| Cosine (0.2, 1, 2) | $21.92 \pm 0.29$ | $13.06 \pm 0.21$ | $11.10 \pm 0.20$ | $9.85 \pm 0.25$ | $8.90 \pm 0.25$ | $7.33 \pm 0.29$ |
| Cosine (0, 0.5, 1) | $20.49 \pm 0.27$ | $9.60 \pm 0.21$ | $6.85 \pm 0.07$ | $5.23 \pm 0.12$ | $4.22 \pm 0.14$ | $3.05 \pm 0.16$ |
| Sigmoid (0, 3, 0.7) | $23.66 \pm 0.17$ | $14.10 \pm 0.31$ | $11.90 \pm 0.23$ | $10.48 \pm 0.26$ | $9.44 \pm 0.27$ | $7.78 \pm 0.29$ |
| Sigmoid ($-3$, 3, 1) | $16.28 \pm 0.06$ | $8.16 \pm 0.23$ | $6.56 \pm 0.23$ | $5.66 \pm 0.21$ | $5.03 \pm 0.22$ | $4.12 \pm 0.24$ |
| Sigmoid ($-3$, 3, 0.7) | $\underline{13.67 \pm 0.06}$ | $6.60 \pm 0.19$ | $5.31 \pm 0.20$ | $4.55 \pm 0.24$ | $4.05 \pm 0.25$ | $3.36 \pm 0.22$ |
| Spectral | $\mathbf{12.50 \pm 0.22}$ | $\mathbf{5.44 \pm 0.20}$ | $\mathbf{4.21 \pm 0.21}$ | $\mathbf{3.56 \pm 0.24}$ | $\underline{3.20 \pm 0.23}$ | $\underline{2.60 \pm 0.21}$ |

The results presented in Tables 1 and 2 align with the experimental observations in Figure 19, supporting the spectral recommendation and its relation to alternative noise schedules. The optimality with respect to the Wasserstein loss remains robust, while for the FID metric, as the number of diffusion steps increases and discretization error becomes less significant, schedules such as EDM achieve comparable performance.

**Experimental result reproducibility**

To facilitate easier reproduction of our results, we provide examples of the derived spectral schedules obtained for the CIFAR-10 dataset by minimizing the Wasserstein-2 loss for [10, 20, 28] diffusion steps. The values are rounded to four decimal places.

**10 Diffusion Steps:** [0.9999, 0.9899, 0.9177, 0.6796, 0.3398, 0.1271, 0.0429, 0.0135, 0.0030, 0.00004]

**20 Diffusion Steps:** [0.9999, 0.9981, 0.9914, 0.9735, 0.9340, 0.8591, 0.7389, 0.5814, 0.4162, 0.2751, 0.1724, 0.1048, 0.0627, 0.0371, 0.0216, 0.0121, 0.0063, 0.0028, 0.0008, 0.00004]

**28 Diffusion Steps:** [0.9999, 0.9989, 0.9961, 0.9899, 0.9778, 0.9562, 0.9204, 0.8656, 0.7882, 0.6889, 0.5747, 0.4572, 0.3487, 0.2570, 0.1849, 0.1309, 0.0917, 0.0638, 0.0442, 0.0304, 0.0207, 0.0138, 0.0089, 0.0055, 0.0031, 0.0015, 0.0005, 0.00004]

# K    Further Discussion

## K.1    Relationship Between Noise Schedules and Eigenvalues

To explore the relationship between the optimal spectral noise schedule and the data characteristics, we solved the optimization problem for each eigenvalue individually, with the contributions from the other eigenvalues set to zero. Using the eigenvalues of the covariance matrix from 6.1, Figure 22a shows these eigenvalues, while 22b presents the optimal solutions for 50 diffusion steps, computed using the *Wasserstein-2* distance in the optimization problem. Each solution corresponds to a single eigenvalue (considering only positions 2 to 10 for clarity)[7]

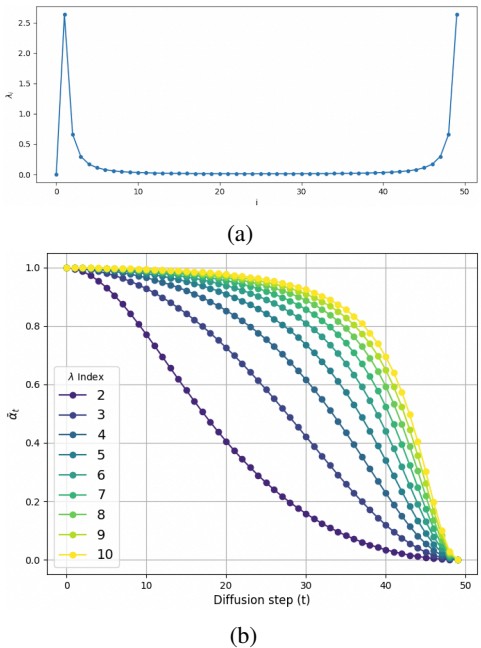

(a)

(b)

Figure 22: Figure 22a shows the eigenvalues for the covariance matrix from Section 6.1. Figure 22b presents the results of solving the optimization problem for each eigenvalue individually, with the contributions from the other eigenvalues set to zero.

It can be observed from Figure 22b that the solution becomes more concave as the magnitude of the eigenvalue decreases (yellow) and more convex as the magnitude increases (blue). Notably, this behavior is determined by the magnitude of the eigenvalues ($\{\lambda_i\}_{i=1}^d$) rather than their indices ($i$), as the objective functions are independent of the index itself.

When the eigenvalues, or equivalently the DFT coefficients, decrease monotonically, a direct relationship emerges between the eigenvalue magnitude and its corresponding frequency (for example, the $1/f$ behavior observed in speech [30]). In such cases, the first eigenvalues correspond to the low frequencies, having larger amplitudes, while the last correspond to high frequencies and smaller amplitudes. This pattern, along, with the previous observations [8], aligns with the well-known coarse-to-fine signal construction behavior of diffusion models.[9]

Interestingly, by examining the spectral recommendation from Figure 1a, it closely resembles the solutions obtained by emphasizing the highest eigenvalue (22b). This suggests that using the *Wasserstein-2 loss* tends to favor larger magnitude eigenvalues. This behavior is also reflected in the relative error, $(|\lambda_i - \lambda_{est}|)/(\lambda_i)$, shown in Figure 23, where larger eigenvalues exhibit smaller relative errors.

---

[7]The first eigenvalue is excluded as it disrupts the monotonicity.

[8]Low magnitude eigenvalues relate with concave schedule and high magnitude eigenvalues correspond to convex schedule.

[9]Higher-frequency components are emphasized by allocating more steps toward the end of the diffusion process, while lower-frequency components are emphasized earlier.

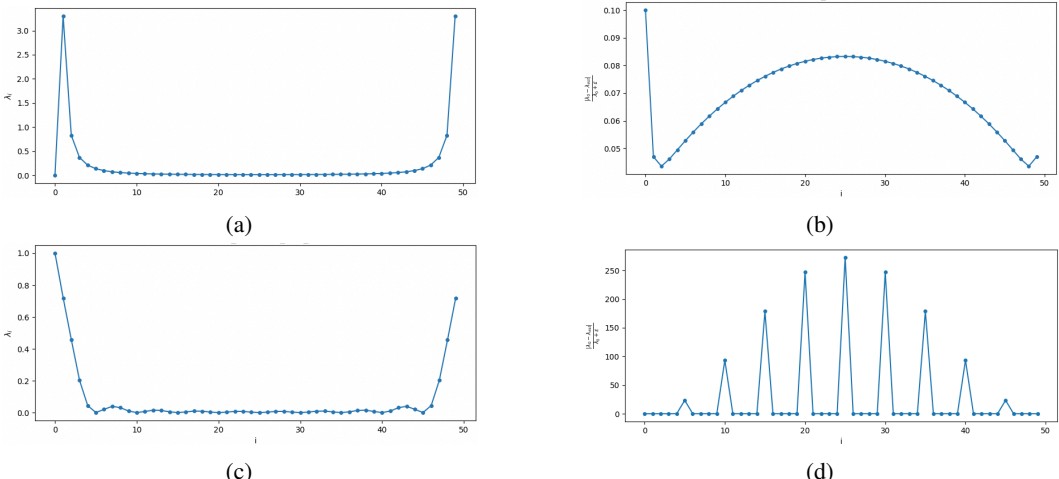

Figure 23: Figures 23a and 23c present the eigenvalues of matrices 12a and 12g, respectively. Figures 23b and 23d illustrate the relative error of the spectral recommendation, using 60 diffusion steps and the *Wasserstein-2* distance. Notably, larger eigenvalues exhibit smaller relative errors. In Figures 23b and 23c, the first elements were manually chosen, as they originally had extreme values (1000 and 10 respectively), and we aimed to keep the figure within a reasonable scale.

The influence of the eigenvalue magnitude, especially the dominant ones, on the resulting schedule is further illustrated through additional examples. Figure 12d displays the covariance matrix from Figure 12a, scaled by a factor of $C = 20$, which amplifies the dominant eigenvalues, as shown by the relation $C \cdot Av = C \cdot \lambda v$. Consequently, the spectral recommendation in Figure 12f appears more convex than in Figure 12c. A similar trend is observed in Figure 16, where the spectral recommendation for $th = 0.05$ exhibits a more concave shape compared to $th = 0.1$. This relationship open up a possibility of designing loss functions which focus on specific frequency ranges of interest. Further discussion is provided in K.3.

Note: We used the *Wasserstein-2* loss. However, alternative measures, such as *KL divergence*, could also yield similar results.

## K.2 Temporal Dynamics of the Diffusion Process

Our analysis has focused so far on the final distribution resulting from the diffusion process, as described by (10). However, the analytical formulation also enables the examination of phenomena occurring throughout the diffusion process. In particular, Equation (29) describes each intermediate state $\mathbf{v}_s$ as a function of the preceding noise schedule components and the initial noise $\mathbf{v}_0$. This enables the study of spectral properties and dynamics across all diffusion steps, rather than being limited to the final output.

Figures 24a and 24b show the relative error $|\lambda_i - \lambda_{\text{est}}|/\lambda_i + \epsilon$ for the 10 largest eigenvalues (sorted, largest on the right) over the final 20 steps of a 60-step diffusion process, using the *Cosine* schedule ($s = 0$, $e = 1$, $\tau = 1$) and the spectral schedule respectively. In both figures, eigenvalues are sorted in ascending order, allowing a comparison between high-frequency components (lower-magnitude eigenvalues) and low-frequency components (higher-magnitude eigenvalues).

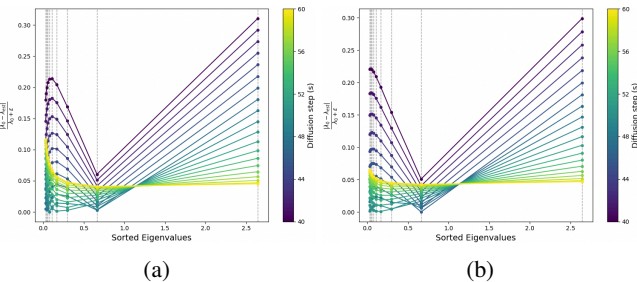

| (a) | (b) |

Figure 24: The relative errors of the the Cosine ($s = 0$, $e = 1$, $\tau = 1$) (24a) and the spectral (24b) schedules for the 10 largest eigenvalues (sorted, largest on the right) over the final 20 steps of a 60-step diffusion process.

In both figures, We observe that the error decreases consistently for large eigenvalues, but increases in the final steps for smaller ones. In addition, the final relative error (yellow curve) is notably lower for high-magnitude eigenvalues (i.e., low frequencies). This behavior aligns with the observed bias toward mid-to-high frequencies reported in prior works [39]. Notably, this effect is more pronounced under the Cosine schedule; while both schedules yield similar errors for large eigenvalues, the Cosine schedule exhibits much higher errors for small eigenvalues (i.e., high-frequency components). This suggests the effectiveness of the proposed optimization procedure in reducing such errors.

An additional aspect is the evolution of eigenvalues throughout the diffusion process. Figures 25a and 25b illustrate this for the *Cosine* ($s = 0$, $e = 1$, $\tau = 1$) and spectral schedules for a 60-step diffusion process. An interesting observation is that low-magnitude eigenvalues tend to converge to their final values faster than high-magnitude ones. Notably, the convergence rate does not necessarily reflect the accuracy of the final values, as discussed in Figure 24. Comparing convergence rates across different noise schedules or leveraging the intermediate expressions to design improved schedules are additional aspects that can be explored; however, we leave them for future work.

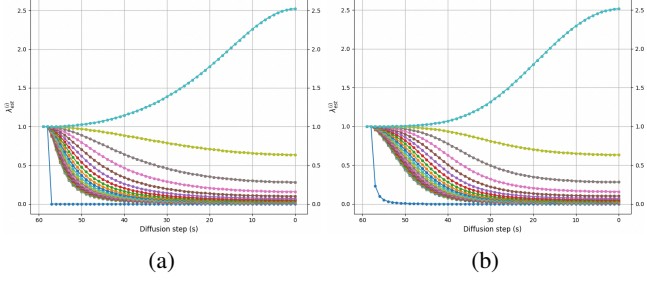

| (a) | (b) |

Figure 25: The dynamics of eigenvalues through a 60-step diffusion process for the Cosine ($s = 0$, $e = 1$, $\tau = 1$) (25a) and the spectral (25b)schedules.

Figure 26 illustrates the evolution of the *Wasserstein-2* error over the final 20 steps of a 60-step diffusion process. It can be observed that the *Wasserstein-2* error consistently decreases for both the cosine (0,1,1) (Figure 26a ) and the spectral noise schedules (Figure 26b ), with the spectral schedule converging to a lower final value

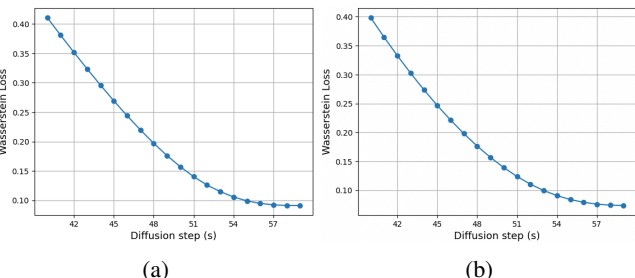

(a)                                   (b)

Figure 26: The dynamics of the *Wasserstein-2* distance over the final 20 steps of a 60-step diffusion process for the Cosine ($s = 0$, $e = 1$, $\tau = 1$) (26a) and the spectral (26b) schedules.

### K.3 Relationship Between Noise Schedules and the loss functions

Building on the spectral monotonic behavior, the loss function can be adjusted to weight different spectral regions in various ways, shaping the noise schedule based on specific objectives.

We propose a *weighted l1* loss for the first and the second moments of two Gaussian distributions $P(\hat{\mathbf{v}}_0; \bar{\boldsymbol{\alpha}})$ and $P(\mathbf{v}_0)$.

$$\mathcal{D}_{L_1}\big(P(\hat{\mathbf{v}}_0; \bar{\boldsymbol{\alpha}}), P(\mathbf{v}_0)\big) = \sum_{i=1}^{d} \frac{\lambda_i}{\sum_j \lambda_j} \left| [\mathbf{D}_1]_i^2 - \lambda_i \right| + \sum_{i=1}^{d} \frac{[\boldsymbol{\mu}_0^{\mathbf{u}}]_i^2}{\sum_j [\boldsymbol{\mu}_0^{\mathbf{u}}]_j^2} \left([\mathbf{D}_2]_i - 1\right)^2 \tag{64}$$

The first term applies a *weighted l1* loss to the eigenvalues, while the second term computes a *weighted l2* norm of the mean vectors.[10] This design ensures that eigenvalues with larger magnitudes and mean components with higher values have greater influence on the overall loss.

Figure 27 illustrates the spectral recommendation obtained by solving the optimization problem in 14 using the *Weighted l1* loss. The results are based on the Gaussian MUSIC-Piano dataset described in 6.2 where $d = 16,000$ and $th = 0.05$.

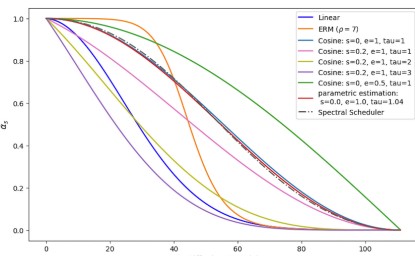

Figure 27: Comparison of the spectral schedule with various heuristic noise schedules for $S = 112$ diffusion steps. The figure presents the linear, EDM ($\rho = 7$), Cosine-based schedules, including *Cosine* ($s = 0, e = 1, \tau = 1$) from [23, 5]. The parametric estimation of the cosine function is shown in red.

Interestingly, using the *weighted l1* loss results in a spectral recommendation that aligns with established heuristic methods. Specifically, it corresponds to the manually designed *cosine (0,1,1)* schedule proposed in [23]. This observation could indicate a potential link between the design of widely used noise schedule heuristics and a bias against high-frequency regions, which has been observed in previous research [39].

Note: The relationship between the magnitude of the eigenvalues and their corresponding frequencies holds tight only when monotonic behavior is present. In real-world scenarios, as shown in Figure 18c, the eigenvalues' magnitudes generally decrease, but the function is not strictly monotonic. In such cases, an alternative approach is required, one that either analyzes broader spectral regions or considering both the values and indices of the eigenvalues.

---

[10]We aim to maintain the relationship between both components similar to the *Wasserstein-2* distance.

# L   Analysis of Mean Bias

In the following section, we analyze the mean bias expression. To maintain consistency and preserve the connection to the frequency domain, we conduct the analysis on the circulant matrices used in Section 6.1, which can be diagonalized using the discrete Fourier transform (DFT), denoted by $\mathcal{F}\{\cdot\}$. Naturally, the same approach can be extended to any covariance matrix, $\mathbf{\Sigma}_0$, using its eigenbasis $\mathbf{U}$.

The mean bias expression $(\mathbf{D}_2 - \mathbf{I})\boldsymbol{\mu}_0^{\mathcal{F}}$ arises from the difference between $\mathbb{E}\left[\mathbf{x}_0^{\mathcal{F}}\right]$ and $\mathbb{E}\left[\hat{\mathbf{x}}_0^{\mathcal{F}}\right]$. In particular, We focus on the absolute magnitude of the expression $|\mathbf{D}_2 - \mathbf{I}||\boldsymbol{\mu}_0^{\mathcal{F}}|$. The term $\mathbf{D}_2$, as defined in (10), depends on $\mathbf{\Lambda}_0$ and on $\bar{\alpha}$, the chosen noise schedule. Notably, for stationary signals, $\boldsymbol{\mu}_0$ is deterministic, resulting in the vector $\boldsymbol{\mu}_0^{\mathcal{F}}$ where all entries are zero except for the first element. specifically, for a mean-centered signal where $\boldsymbol{\mu}_0^{\mathcal{F}} = \mathbf{0}$, the DDIM process remains unbiased, regardless of $|\mathbf{D}_2 - \mathbf{I}|$ expression. In other cases, for a given $\boldsymbol{\mu}_0^{\mathcal{F}}$, the primary source of bias originates from the main diagonal of $|\mathbf{D}_2 - \mathbf{I}|$.

Figure 28 analyzes the mean bias for two choices of $\mathbf{\Lambda}_0$ with $d = 50$. It compares the values of $|\mathbf{D}_2 - \mathbf{I}|$ across several cosine noise schedule heuristics and illustrates how its behavior depends on the number of diffusion steps.

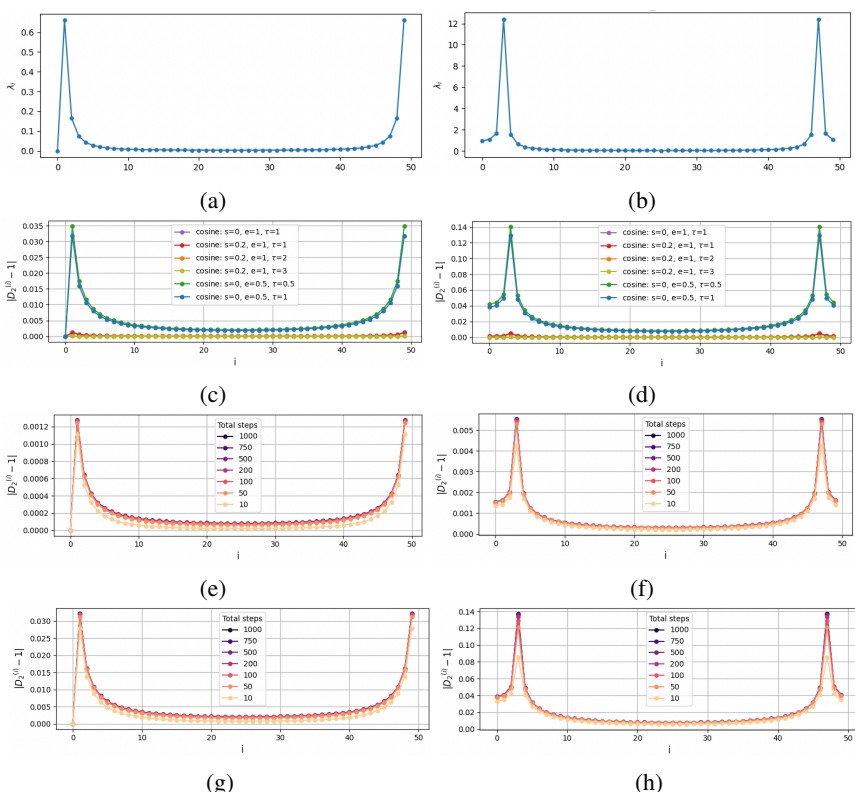

Figure 28: Figures 28a and 28b display the eigenvalues for two choices of $\mathbf{\Lambda}_0$ with $d = 50$. Figures 28c and 28d compare the mean bias values across different parametrization of the Cosine noise schedule using 112 diffusion steps. Figures 28e and 28f show the bias for varying numbers of diffusion steps with the Cosine (0,1,1) noise schedule, while Figures 28g and 28h illustrate the same for the Cosine (0,0.5,1) schedule.

Figures 28c and 28d reveal that for certain heuristics, such as Cosine (0,1,1), the bias is negligible, while for others, like Cosine (0,0.5,1), the bias increases. Additionally, the magnitude of the eigenvalues $\{\lambda_i\}_{i=1}^d$ plays a significant role in determining the bias; as the eigenvalues grow larger, the bias also tends to increase.

Figures 28e and 28f illustrate the bias across various numbers of diffusion steps $\{10, 50, 100, 200, 500, 750, 1000\}$ for the Cosine (0,1,1) noise schedule. Similarly, Figures 28g

and 28h show the bias for the Cosine (0,0.5,1). In both cases, increasing the number of diffusion steps leads to a gradual rise in the mean bias $|\mathbf{D}_2 - \mathbf{I}|$.

## M  DDPM vs DDIM

The explicit formulations of DDPM and DDIM in (12) and (10) also enable their comparison in terms of loss across varying diffusion depths and noise schedules. Figure 29 presents such a comparison using the *Wasserstein-2* distance on a logarithmic scale. The results show that DDIM sampling is faster and yields lower loss values than DDPM for each noise schedule, aligning with the empirical observations in [27]. In addition, it can be seen that the spectral schedule with DDIM consistently maintains optimality with respect to all other experiments.

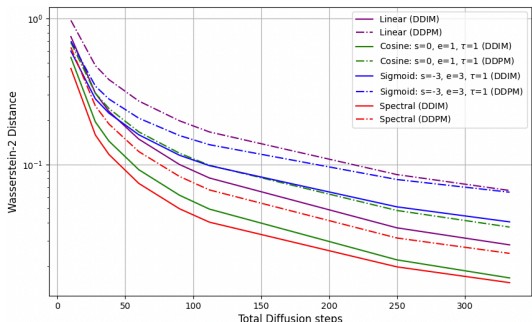

Figure 29: Comparison of the *Wasserstein-2* distance between DDIM and DDPM for different noise schedules, including the spectral recommendation, across various diffusion steps.

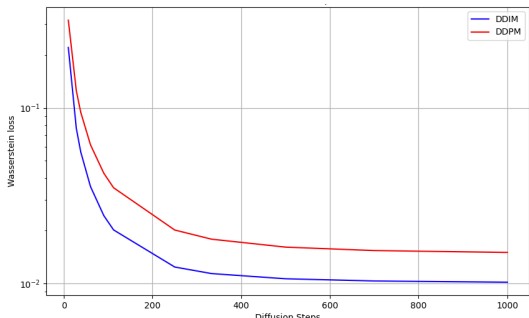

Figure 30: Comparison of the spectral noise schedules in DDPM and DDIM, derived from the covariance matrix outlined in Sec. 6.1, with parameters d=50, l=0.1, and total diffusion steps set to $\{10, 28, 38, 60, 90, 112, 250, 334, 500, 750, 1000\}$.

## N  Optimization Time Analysis

The computational cost of solving the optimization problem for time-dependent signals depends on several factors. As discussed in Appendix I.4, increasing the resolution of temporal signals improves both accuracy and adaptation to the data characteristics, but also increases computational cost (larger covariance matrices allow for higher frequency accuracy). Additionally, factors like the sampling rate and stationary regions influence the results.

We report the computation times for the covariance matrices of different datasets, with dimensions $400 \times 400$ for MUSIC, $3072 \times 3072$ for CIFAR10, and $12288 \times 12288$ for AFHQv2. For the MUSIC dataset, we applied stopping criteria of 1500 optimization steps or a function tolerance of $10^{-6}$. For CIFAR10 and AFHQv2, we used 2000 steps or the same tolerance to ensure convergence for a large number of diffusion steps. All computations were performed on a standard CPU. Notably, using slightly more relaxed stopping criteria could achieve the same optimal schedule in less time.

Table 3: Comparison of the optimization time (in seconds) for a $400 \times 400$ covariance matrix (e.g., MUSIC dataset), a $3072 \times 3072$ covariance matrix (e.g., CIFAR10 dataset) and a $12288 \times 12288$ covariance matrix (e.g., AFHQv2 dataset)

| Diffusion steps | $400 \times 400$ | $3072 \times 3072$ | $12288 \times 12288$ |
|---|---|---|---|
| 10 | 0.09 | 0.36 | 3.88 |
| 50 | 2.94 | 8.19 | 93.09 |
| 90 | 18.31 | 25.4 | 292.48 |
| 130 | 55.21 | 321.76 | 1555.87 |
| 170 | 69.5 | 558.76 | 2316.73 |
| 210 | 172.02 | 594.51 | 3884.07 |
| 250 | 261.69 | 949.00 | 7417.46 |

As shown in the Table 3, for a low number of steps, where discretization error is significant and the spectral scheduler greatly improves performance, the computation time ranges from less than a second to a few tens of seconds. Over approximately 130 diffusion steps, the computation time exceeds one minute for the CIFAR10; however, in this regime, discretization errors become negligible, the overall structure of the scheduler remains unchanged, and its density increases with minimal change.

To accelerate the optimization process, we leverage this structural similarity by adopting a gradual optimization strategy: we begin with a small number of diffusion steps (e.g., 10) which requires relatively low computational effort and use the resulting solution (interpolated to the appropriate length) to initialize optimization for a larger number of steps. This approach significantly reduces computation time by reusing information across step counts.

Table 4 compares optimization times on the AFHQv2 dataset using random initialization (left column) versus iterative initialization. For the iterative approach, the third column reports the time for each individual run, while the forth shows the total cumulative time to reach the solution for a given diffusion step count. As observed in Table 4, iterative initialization consistently reduces optimization time while achieving noise schedules and loss values comparable to random initialization. Further improvements may be possible by refining the strategy for transitioning between diffusion step counts during optimization.

An additional observation from Table 3 is the significant convergence time required for large covariance matrices, especially when a high number of diffusion steps is used. While covariance matrices in audio signals are typically considered in the stationary domain with moderate dimensions, image data often involve higher resolutions, resulting in larger matrices and increased computational cost. A possible approach to reduce computational cost is to assume a low-rank covariance matrix, selecting only the $d$ most significant directions and ignoring near-zero eigenvalues.

Specifically, we apply PCA to the original dataset and conduct optimization within a reduced-dimensional subspace. For the AFHQv2 dataset, the dimensionality is reduced from $64 \times 64 \times 3$ $(12, 288)$ to $32 \times 32 \times 3$ $(3, 072)$ using PCA, with results shown in the rightmost column of Table 4. As expected, optimization in this reduced subspace is faster than in the original space ($64 \times 64 \times 3$, left column). The subspace dimensionality acts as a hyperparameter; here, we set $d = 3072$, achieving a significant reduction while preserving the dominant spectral components.

Table 4: Comparison of optimization times (seconds) across proposed acceleration methods on AFHQv2 dataset

| Diffusion steps | Random init. $(12288 \times 12288)$ | Iterative init. $(12288 \times 12288)$ | Iterative init. (summed) $(12288 \times 12288)$ | PCA $(3072 \times 3072)$ |
|---|---|---|---|---|
| 10 | 3.88 | 3.92 | 3.92 | 0.49 |
| 50 | 93.09 | 39.60 | 43.52 | 12.71 |
| 90 | 292.48 | 21.61 | 65.13 | 32.80 |
| 130 | 1555.87 | 46.05 | 111.18 | 350.27 |
| 170 | 2316.73 | 20.59 | 131.78 | 548.57 |
| 210 | 3884.07 | 32.00 | 163.78 | 605.66 |
| 250 | 74117.47 | 68.23 | 232.02 | 919.99 |

