# OpenReview forum: "Spectral Analysis of Diffusion Models with Application to Schedule Design"
_NeurIPS.cc/2025/Conference — NeurIPS 2025 poster_

### Official Review · Reviewer_M8z5 · 2025-07-02

**Clarity:** 3
**Significance:** 3
**Originality:** 3
**Rating:** 5
**Confidence:** 4

**Summary:**

This work analyzes how to schedule the noise based on the data features from a spectral perspective. To achieve this goal, this work first presents a spectral analysis on the reverse process based on the Gaussian target distribution. Then, based on the spectral analysis, they propose an optimization problem to obtain the noise schedule specific to the given data feature. The simulation and real-world experiments (including VP and VE-based models) also support their theoretical results.

**Questions:**

Please see the Weakness part.
1.	It would be better to discuss the technique challenge when considering GMM target distribution (it has a closed-form score function, but does not have a closed-form denoiser).

**Ethical Concerns:**

["NO or VERY MINOR ethics concerns only"]

**Final Justification:**

As the authors addressed my concerns, I have raised my score to 5 and vote for accepting this paper.

**Limitations:**

yes

**Quality:**

3

**Strengths And Weaknesses:**

Strength:

1.	The spectral analysis for the noise schedule design is novel and interesting.

2.	The simulation and real-world experiments also support their theoretical results.

Weakness:

1.	The analysis of this work is limited to the Gaussian distribution, which does not match the real-world data distribution. The high-dimensional Gaussian assumption will also lead to a time-consuming optimization process for the given optimization problem.

For the above problems, some works [1][2] assume the target data distribution admits a low-dimensional linear subspace and the latent distribution is Gaussian. [2] further prove that under the linear subspace assumption, the optimization process of diffusion models is equivalent to PCA.

As mentioned in the limitation part of this work, PCA is a promising method, and it will be interesting to consider this modeling in future work.

[1] Chen, Minshuo, Kaixuan Huang, Tuo Zhao, and Mengdi Wang. "Score approximation, estimation and distribution recovery of diffusion models on low-dimensional data." In International Conference on Machine Learning, pp. 4672-4712. PMLR, 2023.

[2] Yang, Ruofeng, Bo Jiang, Cheng Chen, Baoxiang Wang, and Shuai Li. "Few-shot diffusion models escape the curse of dimensionality." Advances in Neural Information Processing Systems 37 (2024): 68528-68558.

---

> ### Author Rebuttal · Authors · 2025-07-30
>
> Thank you for your constructive feedback. We appreciate your insightful comments as well as the valuable references you provided. We address your points below.
>
> **Answer to weaknesses:**
> 1. We acknowledge that the Gaussian assumption does not fully capture the complexity of real-world data distributions. However, our aim was to present a broader spectral perspective on diffusion models and to develop a method that could potentially generalize to more realistic settings in future work (see answer to question 2). In addition, we include further results on the AFHQV2 dataset (64*64×3), which remain consistent with the theoretical conclusions and further support them (please refer to Weakness 2 in Reviewer YDdm).
> 2. As discussed in the limitations section, the computational time for solving the optimization problem scales with the number of diffusion steps and the data resolution. Importantly, this optimization is performed only once per dataset and target step count to determine the noise schedule. Nonetheless, we address these concerns explicitly here. Table 3 reports optimization times in seconds, including results from the paper and additional experiments on the AFHQv2 dataset at 64×64 resolution (see Reviewer YDdm).
>
> **Table 3 – Optimization times (in seconds) for different numbers of diffusion steps and dataset resolutions**
> \\[
> \\begin{array}{cccc}
> \\text{Diffusion steps} & \\text{MUSIC } & \\text{CIFAR10 }& \\text{AFHQv2 } \\\\
>  & (400\\!\\times\\!400) &(3072\\!\\times\\!3072) & (12288\\!\\times\\!12288) \\\\
> \\hline
> 10 & 0.09 & 0.36 & 3.88 \\\\
> 50 & 2.94 & 8.19 & 93.09 \\\\
> 90 & 18.31 & 25.4 & 292.48 \\\\
> 130 & 55.21 & 321.76 & 1555.87 \\\\
> 170 & 69.5 & 558.76 & 2316.73 \\\\
> 210 & 172.02 & 594.76 & 3884.07 \\\\
> 250 & 261.69 & 949.0 & 7417.46 \\\\
> \\end{array}
> \\]
>
> - Table 3 shows that optimization times increase with both resolution (e.g., covariance dimension) and the number of diffusion steps. To address this, we propose two methods to effectively accelerate the process.
>
> **Approach 1 – Iterative Initialization**
>
> The timings reported in Table 3 correspond to runs where the optimization was independently initialized from random points for each number of diffusion steps, which can result in high computational cost. However, as noted in Section 6.1, optimal solutions for different step counts tend to exhibit similar structures for a given dataset, even though they are not identical, since the noise schedule induces a different density for each step count (see Figure 1a).\
> To leverage this structural similarity, we adopt a gradual optimization strategy: we begin with a small number of diffusion steps (e.g., 10) which requires relatively low computational effort and use the resulting solution (interpolated to the appropriate dimension) to initialize optimization for a larger number of steps. This approach significantly reduces computation time by reusing information across step counts.\
> Table 4 compares optimization times on the AFHQv2 dataset using random initialization (left column) versus iterative initialization. For the iterative approach, one column reports the time for each individual run, while the other shows the total cumulative time to reach the solution for a given diffusion step count.
>
> *Key observations:*
> - Optimization times with iterative initialization are consistently faster than with random initialization; for example, on AFHQv2 with 250 diffusion steps, it is 31 times faster then using the random initialization.
> - The resulting noise schedules and corresponding loss values match those obtained with random initialization, regardless of the number of diffusion steps. (A figure illustrating this will be added in the revised paper.)
> - Alternative strategies for transitioning between diffusion step counts during optimization may offer additional speedups. Due to space constraints, we defer a detailed discussion of these configurations to the final version of the paper.
>
> **Approach 2 - Optimization over a Principal Subspace**
>
> We now turn to scenarios involving high-resolution datasets and thank the reviewer for highlighting references [9] and [10]. Paper [9] investigates the relationship between the intrinsic data distribution and the learned score function, along with examining its impact on sample complexity under a low-dimensional linear Gaussian subspace assumption. Similarly, Paper [10] connects few-shot diffusion models to a PCA-like objective. Building on these insights and as noted in our limitations, we propose to evaluate our optimization method on a PCA-derived subspace to explore potential computational acceleration.\
> Leveraging the fact that the optimization problem is formulated in the spectral domain, and assuming the dataset can be approximated as low-rank with many eigenvalues close to zero (see Figure 18), we propose to concentrate the optimization on the most significant components. Specifically, we apply PCA to the original dataset and perform the optimization in a reduced-dimensional subspace.  We demonstrate our method on the AFHQv2 dataset by reducing the original dimensionality from 64 × 64 × 3 (12288) to 32 × 32 × 3 (3072) using PCA. The results are shown in the rightmost column of Table 4.
>
> *Key observations:*
> - As expected, solving the optimization in the reduced dimension leads to shorter runtimes compared to operating in the original space (64 × 64 ×3 – see left column).
> - The resulting noise schedules and loss values closely match those obtained from the full dimensional baseline. (Following the derivation of the optimal schedule in the subspace, we assess its loss within the original full dimensional space.)
> - The subspace dimensionality serves as a hyperparameter. We chose $d = 3072$, which offers a substantial reduction while preserving the dominant spectral components. A more principled selection of $d$ is possible, which we leave to be addressed in the full paper.
>
> **Table 4 - Comparison of Optimization Time (is seconds) Across Proposed Methods**
>
> \\[
> \\begin{array}{ccccc}
> \\text{Diffusion steps}
> & \\text{Random Init. }
> & \\text{Iterative init. }
> & \\text{Iterative init. (summed) }
> & \\text{PCA } \\\\
> & (12288\\!\\times\\!12288)
> & (12288\\!\\times\\!12288)
> & (12288\\!\\times\\!12288)
> &(3072\\!\\times\\!3072) \\\\
> \\hline
> 10  & 3.88     & 3.92   & 3.92    & 0.49   \\\\
> 50  & 93.09    & 39.60  & 43.52   & 12.71  \\\\
> 90  & 292.48   & 21.61  & 65.13   & 32.80  \\\\
> 130 & 1555.87  & 46.05  & 111.18  & 350.27 \\\\
> 170 & 2316.73  & 20.59  & 131.78  & 548.57 \\\\
> 210 & 3884.07  & 32.00  & 163.78  & 605.66 \\\\
> 250 & 7417.47  & 68.23  & 232.02  & 919.99 \\\\
> \\end{array}
> \\]
>
>
> *General observations:*
> - As shown in Table 4, the iterative initialization method substantially accelerates the optimization process without necessitating the data dimensionality reduction required by the second approach, making it well-suited for high-resolution datasets and large numbers of diffusion steps. Combining both approaches is a promising option.
> - Another interesting direction comes from the common use of latent diffusion models for high-dimensional data, where the diffusion process is carried out in a lower-dimensional latent space. It would be interesting to explore how our method generalizes in such settings.
>
> **Answer to questions:**
> 1. Incorporating Gaussian Mixture Models (GMMs) into the analytical framework is an intriguing possibility, as explicit expressions for the score function are available in this setting. We anticipate that deriving optimal denoiser expressions for GMMs is feasible and leave this to future work. Notably, formulating an optimization problem similar to ours requires the sampling expressions to be jointly diagonalizable. Additional possible research directions include deriving analytical solutions for synthesis conditioned on degraded signals and designing loss functions which emphasize desired solution characteristics (See Appendix k3).
>
> [9] Chen, Minshuo, Kaixuan Huang, Tuo Zhao, and Mengdi Wang. "Score approximation, estimation and distribution recovery of diffusion models on low-dimensional data." In International Conference on Machine Learning, pp. 4672-4712. PMLR, 2023.
>
> [10] Yang, Ruofeng, Bo Jiang, Cheng Chen, Baoxiang Wang, and Shuai Li. "Few-shot diffusion models escape the curse of dimensionality." Advances in Neural Information Processing Systems 37 (2024): 68528-68558.

---

> > ### Comment · Area_Chair_jyj7 · 2025-08-05
> > **Action Required: Author–Reviewer Discussion Closing Soon**
> >
> > Dear Reviewer,
> >
> > A gentle reminder that the extended Author–Reviewer Discussion phase ends on **August 8 (AoE)**.
> >
> > Please read the authors’ rebuttal and participate in the discussion **ASAP**. Regardless of whether your concerns have been addressed, kindly communicate:
> >
> > - If your concerns have been resolved, please acknowledge this clearly.
> >
> > - If not, please communicate what remains unaddressed.
> >
> > The “Mandatory Acknowledgement” should only be submitted **after**:
> >
> > - Reading the rebuttal,
> >
> > - Engaging in author discussion,
> >
> > - Completing the "Final Justification" (and updating your rating).
> >
> > **As per policy, I may flag any missing or unresponsive reviews and deactivate them once additional reviewer feedback has been posted.**
> >
> > Thank you for your timely and thoughtful contributions.
> >
> >
> >
> >
> > Best regards,
> >
> > AC

---

> > ### Comment · Reviewer_M8z5 · 2025-08-06
> >
> > Thanks for the author's response on the large-scale real-world diffusion models, optimization time, and theory. As shown in the experiments, with the spectral schedule (optimized in only 10 seconds), we can achieve a better FID and W2 distance, which supports their theoretical results. I am also looking forward to the extension of the setting to be closer to real-world experiments. I will raise my score to 5.

---

> > > ### Author Response · Authors · 2025-08-07
> > >
> > > We would like to thank the reviewer for their constructive feedback, support of our work, and for raising the score. We share the interest in exploring this extension and will include a discussion of this intriguing direction within the paper.

---

> ### Comment · Area_Chair_jyj7 · 2025-08-04
> **Action Required: Author–Reviewer Discussion Closing Soon**
>
> Dear Reviewer,
>
>
>
> This is a gentle reminder that the **Author–Reviewer Discussion** phase ends within just three days (by **August 6**). Please take a moment to read the authors’ rebuttal thoroughly and engage in the discussion. Ideally, every reviewer will respond so the authors know their rebuttal has been seen and considered.
>
>
>
> Thank you for your prompt participation!
>
>
>
> Best regards,
>
>
>
> AC

---

### Official Review · Reviewer_M2PR · 2025-07-02

**Clarity:** 2
**Significance:** 2
**Originality:** 3
**Rating:** 4
**Confidence:** 3

**Summary:**

Diffusion model has a strong ability to model complex data distributions and generate realistic new samples. However, its generation process depends on a heuristically chosen noise schedule without a solid theoretical basis. In this paper, assuming the target distribution is Gaussian, the authors demonstrate the inference process as a closed-form spectral transfer function, showing how the generated signal develops in response to the initial noise. They then use this analysis to determine an optimal noise schedule that effectively matches the data's characteristics.

**Questions:**

1. The Gaussianity assumption about the target distribution is somewhat unrealistic because it is unlikely to occur in real datasets. This reduces the effectiveness of the proposed method when applied to actual data, as shown in Section 6.3. Figure 4 does not demonstrate the significant performance difference of the learned noise schedule compared to heuristic baselines.

2. Even in the synthetic case where the assumption holds, Figure 1.3 indicates that the performance gap (Wasserstein distance) of the learned noise schedule relative to others is minimal. This raises the question of whether a complex "trained" noise scheduler is truly necessary. Is the cost of tuning hyperparameters for heuristic noise schedules (e.g., cosine, sigmoid) higher than the cost of discovering the "trained" noise schedule? If not, what is the main reason for developing this "trained" noise schedule?

3. The statement in Lines 300-301 about the mean drift, “Additionally, as the depth of the diffusion process increases, the bias value tends to grow, regardless of the selected noise schedule.”, diminishes the motivation behind this paper.

4. Other minor points:
- In Eq 14, what are $\epsilon_0$, $\epsilon_S$? Additional explanations are needed.
- Visualizations of some figures (e.g., Figures 1b,c, and 4a, b, c) could be improved by combining the legends horizontally and adding color to make certain lines more distinguishable.

**Ethical Concerns:**

["NO or VERY MINOR ethics concerns only"]

**Final Justification:**

Before the rebuttal, my primary concerns were the unrealistic Gaussianity assumption and the minimal improvement achieved, potentially at a higher cost than other methods. After the rebuttal, the authors’ clarifications sufficiently addressed these issues. Therefore, I have decided to maintain my score.

**Limitations:**

Please see the above questions

**Quality:**

3

**Strengths And Weaknesses:**

Strengths:
- Based on the Gaussianity assumption, a novel theoretical spectral analysis of the discrete diffusion reverse process is presented, along with a closed-form expression for its spectral transfer function.
- Formulates an optimization problem to find an optimal noise schedule that matches data characteristics, regardless of the number of diffusion steps.
- A well-structured paper with high-quality writing.

Weaknesses:
- The Gaussianity assumption is too strong and unrealistic.
- The performance of the identified noise schedule does not appear to outperform other schedules.

---

> ### Author Rebuttal · Authors · 2025-07-29
>
> Thank you for your constructive feedback. We appreciate your comments and address them below.
>
> **Answer to weaknesses:**
> 1. We recognize that real-world datasets may not adhere to a Gaussian distribution. As this concern was also raised by Reviewer YDdm under Weakness 2, please refer to our detailed response in that section.
> 2. For a discussion of the optimal noise schedule’s performance, please see our responses to Questions 1 and 2.
>
> **Answer to questions:**
> 1. We address Questions 1 and 2 jointly, as they relate to closely connected aspects of the analysis. A key source of error in diffusion models is discretization error, which stems from the discrepancy between continuous sampling equations and the discrete sampling process implemented in practice. A key consideration for any given dataset is the choice of discretization strategy, particularly the selection of an appropriate noise schedule. As illustrated in Figure 3, the optimal noise schedule can vary significantly between datasets. To avoid the need for extensive exploration of numerous candidate schedules (which would require retraining the denoiser and synthesizing many samples for evaluation), we propose a method that efficiently recommends an optimal noise schedule. This recommendation is computed once per dataset and can be obtained in a reasonable amount of time. (see our response to Reviewer M8z5).\
> In Section 6.1 and Figure 1.3, we present an empirical evaluation, comparing the spectral schedule to existing heuristic schedules evaluated using Gaussian distributed data. Therefore, this experiment aligns completely with the theoretical framework. As shown in Figure 1.3, the spectral schedule is optimal for any given number of diffusion steps. Importantly, when fewer diffusion steps are used and discretization error becomes more significant, the advantage of this optimized schedule is even more evident. This leads to a clear improvement in balancing synthesis time with sample quality.\
> Determining the optimal noise schedule without our proposed method requires extensive experimentation and comparative analysis. Furthermore, even if the optimal schedule for a given dataset closely resembles an existing heuristic in terms of structure or performance, this does not reduce the significance of our approach since each heuristic can be regarded as a plausible candidate solution within the broader optimization framework.\
> In Section 6.3, we derive an optimal noise schedule under the Gaussian assumption and evaluate its performance on real-world datasets using FID and Wasserstein metrics. As shown in Figures 4a and 4b, the spectral noise schedule consistently remains optimal across both metrics and for any selected number of diffusion steps. Similar results were observed in our experiments with the AFHQv2 dataset (refer to our response to Reviewer YDdm under Weakness 2).\
> Figure 4 demonstrates that the spectral recommendation provides a clear advantage at low number of diffusion steps, where discretization error is significant. As the number of steps increases, performance differences across schedules diminish, yet the spectral schedule remains consistently optimal. Notably, for CIFAR-10, the structure of the optimal schedule resembles the EDM heuristic with ρ = 7. Consequently, the performance gap is more pronounced at low number of diffusion steps, and as the schedules become denser, the gap in FID results narrows, though a measurable difference from other baselines persists. As discussed earlier, this behavior does not detract from our method’s validity, since each scheduler can represent a plausible solution, and similarity in structure at high step counts naturally can leads to reduced differences. This is especially clear in the context where the authors of [2] suggest using 18 diffusion steps for EDM as an effective balance between sampling efficiency and synthesis quality. At this particular settings, our method significantly outperforms all other approaches, including EDM.\
> We note that the proposed approach is not a “complex trained noise scheduler” but rather the solution to a straightforward optimization problem performed once for a given dataset (see our responses to Reviewer YDdm under Weakness 2). In contrast, the traditional manual heuristic search for an optimal noise schedule is a time-consuming process that requires extensive tuning across numerous candidate schedules. For example, obtaining statistically robust FID measurements demands generating at least 50,000 samples, repeated three times, for each number of diffusion steps and each heuristic noise schedule under consideration. Even with such extensive experimentation, without analytical guidance, it is challenging to confirm whether the chosen schedule is truly optimal. Nonetheless, we are aware that the Gaussian assumption may not fully capture the characteristics of some real-world datasets. Naturally, extending the analytical framework to more general distributions (e.g. GMMs), would broaden the applicability of the recommendation. We view this as a valuable direction for future research.
>
> 3. In our work, we formulate the task of finding an optimal noise schedule as an optimization problem based on the Wasserstein loss, which decomposes into two components: the difference in mean and the difference in covariance characteristics. We observe a mean drift that tends to increase as the inference process progresses (lines 300–301). However, as noted in lines 296–300, “different choices of the noise schedule influence the bias value, with some choices effectively mitigating it.” This behavior is further illustrated in Appendix L, Figure 27, which shows how different noise schedules affect the magnitude of the mean drift to varying degrees. While a deeper diffusion process generally tends to increase drift, selecting an appropriate noise schedule can minimize this effect. Moreover, since the mean term explicitly appears in the loss function (${([D_{2}]_i-1)}^2$ in Equation (15)), the noise schedule optimization inherently accounts for this drift, thereby strengthening the motivation for our approach.
>
> 4. a) The parameters $\epsilon_0$ and $\epsilon_s$ are selected to start the synthesis process as close as possible to white noise and to end it near the clean data distribution, thereby enabling the capture of fine details. This choice also ensures compatibility between the training and synthesis phases when a denoiser is involved [8]. Specifically, we use $\epsilon_s = 0.04035 \times 10^{-3}$ and $\epsilon_0 = 10^{-4}$.  Furthermore, together with the applied inequality constraints, this setting was originally intended to guarantee that during optimization, the $\bar{\alpha}$ values remain within the [0, 1] range.\
> b) We appreciate the comments regarding the figures in the paper and will revise them accordingly.\
> \
> [8] Lin, Shanchuan and Liu, Bingchen and Li, Jiashi and Yang, Xiao, Common diffusion noise schedules and sample steps are flawed

---

> > ### Comment · Reviewer_M2PR · 2025-08-04
> > **Rebuttal response**
> >
> > Thank you for your detailed responses to my questions. My concerns have been adequately addressed. I will maintain my score with a leaning toward acceptance.

---

> > > ### Author Response · Authors · 2025-08-07
> > >
> > > We thank the reviewer for their constructive feedback. We are pleased that our responses have adequately addressed your concerns.

---

> ### Comment · Area_Chair_jyj7 · 2025-08-04
> **Action Required: Author–Reviewer Discussion Closing Soon**
>
> Dear Reviewer,
>
>
>
> This is a gentle reminder that the **Author–Reviewer Discussion** phase ends within just three days (by **August 6**). Please take a moment to read the authors’ rebuttal thoroughly and engage in the discussion. Ideally, every reviewer will respond so the authors know their rebuttal has been seen and considered.
>
>
>
> Thank you for your prompt participation!
>
>
>
> Best regards,
>
>
>
> AC

---

### Official Review · Reviewer_rtuD · 2025-07-03

**Clarity:** 3
**Significance:** 3
**Originality:** 3
**Rating:** 5
**Confidence:** 4

**Summary:**

An important aspect of diffusion model sampling is the choice of time/noise discretization, or "noise schedule". A bad noise schedule can yield poor sample quality, or at least make sampling require more steps for good sample quality than with a better schedule. Given that noise schedules tend to be chosen on an empirical basis, the authors of this work propose to use theoretical insight to design better noise schedules.

The key idea is that Gaussian/linear model sampling dynamics is solvable, even when time/noise is discretized. The authors use this fact to express the final sample in terms of the noise schedule and initial noise seed, and then to compute final sample quality (in terms of the Wasserstein-2 distance to the 'true' Gaussian distribution) in terms of the noise schedule. They can then compare their "spectral recommendation" to other schedules on real and toy data sets.

**Questions:**

1. Is it possible to define a measure of sample quality for the audio data set? It seems important to have a sense for how good the schedule is doing at generating high quality samples, not just at reproducing the data set's Gaussian approximation.
2. Have the authors identified a case where their recommendation is *particularly* effective? In the case of CIFAR-10, it seems like the EDM schedule does about as well, and does not require a Gaussian fit to use.
3. How feasible / computationally costly is it to do a Gaussian fit on various high-dimensional data sets? Can the authors discuss this point explicitly?

**Ethical Concerns:**

["NO or VERY MINOR ethics concerns only"]

**Final Justification:**

I think the idea for a 'principled' framework for noise schedule design is important and interesting, and that although some of the Gaussian model math has been done before, the paper still offers a worthy contribution. The authors have done additional experiments to bolster the case that their method works well. While it's true that their method sometimes does about as well as the off-the-shelf EDM method, this isn't always true, and they rightfully point out that it's helpful regardless to have some way to predict that one noise schedule or another will be good. They've done a good amount of work to address various reviewers' concerns. I think the paper should be accepted.

**Limitations:**

I think the extent to which their proposal yields meaningfully higher sample quality and/or efficiency should be discussed in more detail, and ideally the authors identify one or more cases where their approach yields a large improvement as opposed to a small one. The authors should also discuss the point that their method requires a Gaussian fit, which is difficult or impossible for large conditional diffusion models.

**Paper Formatting Concerns:**

No major concerns.

**Quality:**

3

**Strengths And Weaknesses:**

**Strengths.** I'm sympathetic to the idea of using theoretical insight to inform the design of things like noise schedules. The idea to exactly solve the (discrete-time) sampling dynamics associated with the Gaussian model, and use this result to determine the theoretical 'goodness' of noise schedules by computing Wasserstein-2 distances (and KLDs), is interesting. I have done similar math before, and do not doubt that the authors' math is correct.



**Weaknesses.** My main concern is that neither the theoretical analysis nor empirical analysis appear to be major contributions. The central theoretical result of the paper is the solution of the Gaussian model's sampling dynamics for an arbitrary $\alpha_t$, but this has been done many times now, as this paper explicitly mentions (see, e.g., line 92). Given this result, it's easy to compute other downstream things like Wasserstein-2 distances and KLDs. The main difference, as far as I can tell, is that this paper's result is in *discrete* rather than continuous time. Although this paper appears to view this as a major technical distinction between its calculations and prior work (see lines 215-216), I kind of disagree. Linear recurrence relations are essentially as easy to solve as linear ODEs. The calculation may be valuable, but it's extremely closely related to work that's been done before.

On the empirical side, the authors use a few data sets to illustrate that their method gives reasonable results. But the evaluation criteria are a little weird, since the authors are mainly concerned with the extent to which their method reproduces the *Gaussian approximation* of a given data set, rather than returns high quality samples. The most interesting plot is Fig. 4a, since it depicts FID scores (a measure of sample quality) for CIFAR-10. According to the plot, the EDM-recommended noise schedule does as good as their method. Since this is the only sample-quality-related plot I can see in the main text, I'm not convinced that the authors' proposed method is particularly good.

A subtle point which the authors don't mention is that even coming up with an 'optimal' noise schedule requires fitting a Gaussian to a data set, which is not always possible for extremely high-dimensional data sets, and is (at least to my knowledge) essentially impossible for text-to-image conditional models. If the EDM schedule performs about as well, but doesn't require this step, that's worth discussing explicitly.

I find all of the figures kind of hard to parse, since the text and labels are very small. It would really help the paper if the sizes of axis labels, numbers, and so on were greatly increased.

A few minor nitpicks. The authors mention the usefulness of a "spectral" perspective on diffusion models, which has parallels to signal processing. But I find this link really misleading. In signal processing settings, spectral analysis involves transforming the time domain to a frequency domain; here, since the data (e.g., an image) may not naturally have a time component, the eigendirections of the covariance matrix do not necessarily have any kind of frequency-like interpretation. Also, the aforementioned papers studying the Gaussian model also do this diagonalization trick, so the spectral idea is far from new.

The authors should probably cite some foundational papers in their first paragraph, where they write "Diffusion Models (DMs) have become ..." There are a number of typos, like misspelled words and missing periods (see, e.g., line 270). These should be fixed.

---

> ### Author Rebuttal · Authors · 2025-07-30
>
> Thank you for the detailed comments. After carefully reviewing the text, we would like to address the points you raised comprehensively.
>
> **Answer to weaknesses:**
> - Theoretical Observations: In lines 92 and 215–216, we refer to three works that rely on the assumption of a Gaussian distribution, each differing from our approach in a distinct manner. Work [3] proposes a method for optimizing noise schedules, presenting an analytical solution under more restrictive assumptions, specifically zero-mean Gaussians with diagonal covariance matrices (see lines 254–259). Furthermore, they introduce a discretization of the continuous SDE and formulate an optimization problem using linearization and KLUB loss. However, this approach is computationally demanding and limited to small number of diffusion steps. Moreover, the authors do not provide an explicit expression for sampling dynamics. Work [4] evaluates several sampling schemes by analyzing various sources of error. To derive continuous-time sampling expressions, the authors assume a centered data distribution and commutativity of the covariance matrices. Although related to our work, this study does not focus on optimizing the noise schedule or analyzing of the diffusion process. Work [5] explores the relationship between the learned score and the true data score. It derives a continuous-time sampling expression by analyzing components that are either orthogonal or aligned with the basis vectors. Notably, this work does not address the problem of optimizing the noise schedule. In both [4] and [5], the resulting time-domain expressions are continuous, disregarding discretization effects and missing an explicit connection to the full traversal of the noise schedule.
> In our work, we derive an explicit and compact expression in the spectral domain based on the discrete formulation. This allows for a simple optimization procedure to compute the noise schedule and also provides insights into the diffusion dynamics. In addition, We believe this perspective could serve as a useful basis for further exploration.
> - Empirical observations: Sections 6.1 and 6.2 validate the theory under the Gaussian assumption, where the experiment fully aligns with the theoretical framework; hence, the Wasserstein distance suffices for evaluation. The results demonstrate that our method outperforms the existing heuristics (see Appendix I). Section 6.3 examines the spectral recommendation on real data, where the Gaussian assumption does not hold. We evaluate performance using both the Wasserstein distanc which aligned with the optimization objective, and FID as a quality measure. Figures 4a and 4b show that for CIFAR-10, the spectral schedule  consistently outperforms heuristic baselines across all number of diffusion steps, highlighting the practical relevance of our theoretical insights.\
> *Interpretation of Results and Proximity to EDM:*\
> Figure 4a shows that for a small number of diffusion steps (e.g., up to 40), the FID gap between the optimal and heuristic schedules is significant. As the number of steps increases, the sampling intervals become smaller and performance differences gradually diminish. As discussed in lines 344–346, this behavior is expected: discretization error is more pronounced when fewer steps are used, making the optimization problem more significant in these regimes. Nonetheless, our solution remains optimal and effective across all number of diffsuion steps.\
> We believe the reviewer’s comment: “According to the plot, the EDM-recommended noise schedule does as good as their method... I’m not convinced that the authors’ proposed method is particularly good”, overlooks a central point. The observed similarity between the optimal solution and a specific heuristic does not diminish the value of our approach. Rather, it shows that certain heuristics may, for unique data characteristics, approximate the optimal schedule.  importantly, our goal is to avoid repeated, ad hoc comparisons across noise schedules. Hence, as aforesaid, our method is not heuristic-based but formulates a principled optimization problem whose solution may, in hindsight, resemble existing heuristics.\
> Figure 3 shows that for CIFAR-10, the optimal schedule shares structural similarity with EDM using $\rho = 7$, making EDM one of the closer-performing heuristics. However a notable performance gap remains, especially at low step counts, where scheduling is crucial. For instance, [2] recommends using 18 diffusion steps for CIFAR-10 and the EDM schedule as a practical trade-off between speed and quality; in this regime, our method consistently outperforms all baselines, including EDM. A similar trend observed in the MUSIC dataset with cosine schedules $(0, 0.5, 1)$. While the optimal schedule may resemble an existing heuristic structurally or in performance, this does not reduce the significance of our approach, as heuristics are plausible candidate solutions within the broader optimization framework.\
> Additionally, In response to your comment on the experimental scope, we have included additional results for the AFHQV2 dataset at 64×64×3 resolution (see our response to reviewer YDdm, Tables 1 and 2). These results demonstrate that the spectral recommendation remains optimal across all tested heuristics for varying step counts on higher-resolution real data. Furthermore, the optimal noise schedule in this case resembles the EDM schedule with $\rho = 5$.
>
> - We acknowledge the reviewer’s concern regarding the use of the term spectral. While “spectral analysis” is often associated with the frequency domain, it can broadly refer to the study of general eigenstructures (e.g., spectral clustering, PCA). In Section 6.1, our analysis focuses on circulant covariance matrices for one-dimensional signals, where the spectral perspective corresponds directly to the Fourier domain and frequency components. In Sections 6.2 and 6.3, the term is used more generally to denote the eigendirections of the covariance matrix. As noted in the related work section, other studies have also employed spectral analysis; however, their objectives and focus differ fundamentally from ours.
> - We appreciate your comment regarding the figures, missing citations, and typos, and we will revise the paper accordingly.
>
> **Answer to Questions:**
> 1. A commonly used metric for evaluating audio quality is the Fréchet Audio Distance (FAD) [6], which leverages features extracted from VGGISH [7]. As discussed in Appendix J, FAD is less reliable for the MUSIC dataset due to its short frame lengths, and less informative for SC09 given its speech-specific nature. A promising alternative would be to replace VGGISH with a model trained specifically for short audio or speech.
> 2. We would like to address this question in light of our response to Weakness 2. Figures 4a and 4b show that the spectral schedule remains optimal across all step counts. For a small number of diffusion steps, the optimal schedule yields a significant improvement over heuristic baselines, as discretization error is more pronounced in this regime. This advantage can help mitigate the tradeoff between sampling speed and synthesis quality.\
> For a larger number of diffusion steps, the gap between the optimal solution and similarly structured heuristics becomes smaller. However, as noted, this does not detract from the value of our approach, which remains effective in this regime. A key strength of our method is its ability to analytically determine an optimal schedule without relying on manual tuning. \
> Regarding the reviewer’s comment that “in the case of CIFAR-10, it seems like the EDM schedule does about as well, and does not require a Gaussian fit to use”, we emphasize that such a conclusion is retrospective and data-dependent. For example, on MUSIC and SC09, cosine schedules with tuned parameters better approximate the optimal solution, while for AFHQV2, a different $\rho$ value for EDM yields improved alignment. Without an analytical framework, identifying the best-performing schedule is speculative and requires a trial-and-error process.
> 3. Please refer to our response to Reviewer M8z5 (Weakness 2), where we provide a detailed comparison of different data resolutions, diffusion step counts, and optimization runtimes. We also propose two approaches for reducing optimization time and demonstrate their effectiveness on the AFHQV2 dataset.
>
> **Limitations:**
>
> As noted, our work assumes a Gaussian-distributed dataset, which may not fully hold in settings such as large conditional diffusion models. In such cases, extending the analytical framework to more expressive models, such as GMMs or probabilistic models conditioned on degraded signals, could be a promising direction for future research. Furthermore, adapting our method to the latent space of latent diffusion models or to the intrinsic manifold structure of datasets may provide additional avenues for exploration (see discussion by Reviewer M8z5).
>
> [3] Amirmojtaba Sabour, Sanja Fidler, and Karsten Kreis. Align your steps: optimizing sampling schedules in diffusion models\
> [4] Emile Pierret and Bruno Galerne. Diffusion models for gaussian distributions: Exact solutions and wasserstein errors.\
> [5] Binxu Wang and John J Vastola. The unreasonable effectiveness of gaussian score approximation for diffusion models and its applications\
> [6] Kevin Kilgour, Mauricio Zuluaga, Dominik Roblek, and Matthew Sharifi. Fr’echet audio distance: A metric for evaluating music enhancement algorithms. \
> [7] Hershey, Shawn and Chaudhuri. CNN architectures for large-scale audio classification

---

> > ### Comment · Reviewer_rtuD · 2025-08-07
> >
> > I thank the authors for their time and detailed responses to my concerns. I think the overall idea is nice, and that some execution-related issues (e.g., the hard-to-read figures) are easily fixable.
> >
> > Thinking on it more, I agree that the general approach to schedule optimization is interesting and that it's helpful to have a 'principled' framework for doing it, even if in some cases the schedule does about as well as other known choices. The authors make a good point that their schedule does a bit better than EDM on CIFAR-10 for < 40 steps. This was hard to notice in the original plot, and I think either focusing more on this region, or else including an additional panel that is a zoomed-in version, would be really helpful to readers. I also think it would be helpful to readers to add a few samples (from the same noise seeds) to be able to see the difference in quality between their schedule's output, and the EDM output, at a glance. Maybe for 40 steps, 30 steps, and 20 steps, or something like that, just as a helpful illustration.
> >
> > I think the additional experiments (e.g., in response to M8z5) help strengthen their case.
> >
> > Overall, I think the paper is a good contribution, and given the authors' responses and additional experiments, I lean towards accepting it. I will raise my score to a 5.

---

> ### Comment · Area_Chair_jyj7 · 2025-08-04
> **Action Required: Author–Reviewer Discussion Closing Soon**
>
> Dear Reviewer,
>
>
>
> This is a gentle reminder that the **Author–Reviewer Discussion** phase ends within just three days (by **August 6**). Please take a moment to read the authors’ rebuttal thoroughly and engage in the discussion. Ideally, every reviewer will respond so the authors know their rebuttal has been seen and considered.
>
>
>
> Thank you for your prompt participation!
>
>
>
> Best regards,
>
>
>
> AC

---

> > ### Comment · Area_Chair_jyj7 · 2025-08-05
> > **Action Required: Author–Reviewer Discussion Closing Soon**
> >
> > Dear Reviewer,
> >
> > A gentle reminder that the extended Author–Reviewer Discussion phase ends on **August 8 (AoE)**.
> >
> > Please read the authors’ rebuttal and participate in the discussion **ASAP**. Regardless of whether your concerns have been addressed, kindly communicate:
> >
> > - If your concerns have been resolved, please acknowledge this clearly.
> >
> > - If not, please communicate what remains unaddressed.
> >
> > The “Mandatory Acknowledgement” should only be submitted **after**:
> >
> > - Reading the rebuttal,
> >
> > - Engaging in author discussion,
> >
> > - Completing the "Final Justification" (and updating your rating).
> >
> > **As per policy, I may flag any missing or unresponsive reviews and deactivate them once additional reviewer feedback has been posted.**
> >
> > Thank you for your timely and thoughtful contributions.
> >
> >
> >
> >
> > Best regards,
> >
> > AC

---

> > > ### Comment · Area_Chair_jyj7 · 2025-08-06
> > > **Action Required: Author–Reviewer Discussion Closing Soon**
> > >
> > > Dear Reviewer,
> > >
> > > This is, **again**, a reminder to actively participate in the author–reviewer discussion (please also refer to the previous two reminders, **Action Required: Author–Reviewer Discussion Closing Soon**).
> > >
> > > **Per policy, failure to respond to the authors’ rebuttal may result in your review being flagged as insufficient, which could lead to desk rejection of your own submission and be recorded for future reference.**
> > >
> > > Please read the authors’ response and reply as soon as possible. Regardless of whether your concerns have been fully addressed:
> > >
> > > - If your concerns are resolved, kindly confirm this clearly.
> > >
> > > - If not, please explain what issues remain.
> > >
> > > Thank you for your cooperation.
> > >
> > > Best,
> > >
> > > AC

---

> ### Author Response · Authors · 2025-08-07
>
> We thank the reviewer for their thoughtful feedback and support of our work. As recommended, we will revise the figures for improved clarity, incorporate clearer numerical comparisons, and add synthesis examples at different numbers of diffusion steps to better illustrate qualitative differences.

---

### Official Review · Reviewer_YDdm · 2025-07-03

**Clarity:** 2
**Significance:** 2
**Originality:** 3
**Rating:** 4
**Confidence:** 2

**Summary:**

The paper presents a novel spectral analysis of the discrete diffusion reverse process. This leads to a closed-form expression for the spectral transfer function, which characterizes how the final generated signal results from a linear transformation of the initial Gaussian noise (Theorems 3.5 and 3.6).

Building on this analysis, the paper introduces an optimization problem to compute a data-dependent noise schedule that aligns with the spectral properties of the target data. This approach allows for the derivation of an optimal noise schedule without being constrained by heuristic bounds or fixed step counts.

The experimental results—spanning synthetic setups and real-world data under Gaussian assumptions—demonstrate that the proposed spectral schedule outperforms standard heuristics (e.g., linear or cosine). Specifically, results on CIFAR-10, MUSIC, and SC09 datasets show that the optimal schedules adapt to the unique spectral characteristics of each dataset, highlighting the limitations of one-size-fits-all heuristics.

**Questions:**

-	In Section 6.3 (lines 338–340), the authors use a pretrained denoiser trained with a continuous noise schedule for the CIFAR-10 dataset. Why not use a pretrained model with a standard discrete schedule, such as this one: https://huggingface.co/google/ddpm-cifar10-32

-	The current approach involves an iterative optimization process that scales with the number of diffusion steps. Is it possible to approximate or bypass this optimization to accelerate computation?

-	How would the results differ if the denoiser were trained from scratch using the optimized spectral schedule, rather than applying it only at inference time?

**Ethical Concerns:**

["NO or VERY MINOR ethics concerns only"]

**Final Justification:**

I thank the authors for further clarifications on the two remaining concerns. I satisfy with the rebuttal and still lean on accepting this paper.

Thanks

**Limitations:**

Yes

**Paper Formatting Concerns:**

No concerns

**Quality:**

2

**Strengths And Weaknesses:**

Note: I have not verified the theoretical proofs in full detail, and my comments are based on my best understanding. I am open to adjusting my evaluation in light of other reviews.

Strengths

•	The idea of a data-dependent, adaptive noise schedule is intuitive and well motivated.

•	The method is inference-time only—it does not require retraining the denoiser.

•	The method is designed for discrete processes (DDIM/DDPM) rather than continuous-time formulations (SDE/ODE), which aligns better with practical implementations.

Weaknesses

•	As acknowledged in the paper’s limitations section, computing the optimal spectral schedule can be slow—scaling with the number of diffusion steps and dataset size—taking several minutes for large setups.

•	The assumption that the target data is drawn from a Gaussian distribution may be too strong for real-world applications. While the experiments include pretrained diffusion models, CIFAR-10 is a relatively small and low-resolution dataset and may not strongly violate this assumption.

---

> ### Author Rebuttal · Authors · 2025-07-29
>
> Thank you for your detailed and constructive feedback. You provided insightful comments and we would like to respond to them.
>
> **Answer to weaknesses:**
> 1. As the efficiency of the optimization procedure was also noted by Reviewer M8z5, please refer to our response under Weakness 2, where we outline several alternative approaches for solving the optimization problem. These methods scale well with both the number of diffusion steps and the dataset size, while achieving an identical noise schedule and hence performance, in significantly less time (on the order of tens of seconds).
>
> 2. In this work, we focus our theoretical analysis on Gaussian signals, which allows us to derive clear and explicit formulations that facilitate the study of both the final output and the intermediate stages of the diffusion process. Additionally, these formulations shed light on various phenomena and their relationship to spectral properties (see Section 6.1 and Appendix K).  We acknowledge that real-world datasets may not follow a Gaussian distribution. However, in Section 6.3, we demonstrate how the recommendations derived from our analysis can remain relevant in practical settings, with solutions achieving optimal results in terms of Wasserstein distance and FID (quality). In addition, Possible extensions of the analytical solution are discussed in our response to Reviewer M8z5 under Question 1.\
> In response to the comment regarding the relatively low resolution of CIFAR-10, we conducted an additional experiment on the higher-resolution AFHQv2 dataset [1], which has a covariance matrix of size 12,288 × 12,288 (corresponding to 64×64×3 images). Applying our method, we derived the optimal noise schedule and evaluated its performance on this real-world dataset using the pretrained denoiser from [2]. Tables 1 and 2 show a comparison of Wasserstein distance and FID scores between our optimal noise schedule and various heuristic baselines on AFHQv2.
>
> **Table 1 – Wasserstein-2 distance comparison on the AFHQV2 dataset between the spectral noise schedule and other heuristics**
>
>
> \\[
> \\begin{array}{lcccccccc}
>  & 10 & 20 & 30 & 40 & 50 & 70 & 90 & 112 \\\\
> \\hline
> \\text{Edm }(\\rho=5) & 8.08 & 4.91 & 3.76 & 3.18 & 2.83 & 2.43 & 2.22 & 2.08 \\\\
> \\text{Edm }(\\rho=7) & 8.21 & 5.01 & 3.19 & 3.34 & 2.93 & 2.50 & 2.32 & 2.18 \\\\
> \\text{Linear} & 7.82 & 4.55 & 3.40 & 2.89 & 2.63 & 2.33 & 2.18 & 2.07 \\\\
> \\text{Cosine }(0,1,1) & 15.16 & 9.66 & 6.81 & 5.31 & 4.33 & 3.23 & 2.67 & 2.32 \\\\
> \\text{Cosine }(0,0.5,1) & 13.47 & 11.25 & 10.02 & 9.28 & 8.64 & 7.78 & 7.22 & 6.78 \\\\
> \\text{Sigmoid }(0,3,0.7) & 10.37 & 7.01 & 5.45 & 4.62 & 4.00 & 3.30 & 2.90 & 2.64 \\\\
> \\text{Sigmoid }(-3,3,0.7) & 11.19 & 7.31 & 5.54 & 4.60 & 3.91 & 3.12 & 2.69 & 2.39 \\\\
> \\text{Spectral} & 7.37 & 4.26 & 3.24 & 2.76 & 2.48 & 2.20 & 2.06 & 1.97 \\\\
> \\end{array}
> \\]
>
> **Table 2 – FID score comparison for the AFHQV2 dataset between the spectral noise schedule and other heuristics**
>
> \\[
> \\begin{array}{lcccccccc}
>  & 10 & 20 & 30 & 40 & 50 & 70 & 90 & 112 \\\\
> \\hline
> \\text{Edm }(\\rho=5) & 10.07 & 4.72 & 3.58 & 2.97 & 2.67 & 2.48 & 2.32 & 2.24 \\\\
> \\text{Edm }(\\rho=7) & 13.37 & 5.11 & 4.56 & 3.37 & 2.89 & 2.57 & 2.38 & 2.30 \\\\
> \\text{Linear} & 11.42 & 6.73 & 5.22 & 4.49 & 4.00 & 3.47 & 3.18 & 2.98 \\\\
> \\text{Cosine }(0,1,1) & 26.11 & 10.01 & 5.79 & 4.27 & 3.53 & 2.87 & 2.60 & 2.46 \\\\
> \\text{Cosine }(0,0.5,1) & 26.54 & 16.31 & 12.50 & 10.63 & 9.13 & 7.37 & 6.38 & 5.72 \\\\
> \\text{Sigmoid }(0,3,0.7) & 18.22 & 11.42 & 9.08 & 8.02 & 7.22 & 6.31 & 5.78 & 5.39 \\\\
> \\text{Sigmoid }(-3,3,0.7) & 15.09 & 7.08 & 4.87 & 4.01 & 3.48 & 2.99 & 2.76 & 2.62 \\\\
> \\text{Spectral} & 8.14 & 4.13 & 3.06 & 2.72 & 2.56 & 2.41 & 2.30 & 2.23 \\\\
> \\end{array}
> \\]
>
>
>
> Tables 1 and 2 demonstrate that the spectral recommendation outperforms all other heuristics across both metrics. As noted in the paper, the performance gap is especially pronounced at low number of diffusion steps, where discretization error tends to be higher. Nevertheless, the spectral schedule remains optimal across all number of steps, making it well-suited for real datasets. A heuristic that is closest to the spectral schedule in both structure and FID performance is EDM with ρ = 5 (a figure illustrating this will be included in the paper, as it cannot be shown here). This differs from the results observed on CIFAR-10, MUSIC and SC09, highlighting the unique spectral characteristics of each dataset. \
> In addition to our experiment on AFHQv2, a higher-resolution dataset, our response to Reviewer M8z5 shows that our method can effectively leverage spectral properties in high-resolution settings. Moreover, we see extending the theoretical framework to more general distributions, such as GMMs, as a intiruiting direction for future research.
>
> **Answer to questions:**
> 1. Section 6.3 evaluates the spectral recommendation against various existing noise schedules by generating thousands of samples for each schedule across different numbers of diffusion steps. We employ a pretrained denoiser trained on a continuous noise schedule for two main reasons. First, it ensures a consistent and fair evaluation by reducing variability that may result from training different denoisers, thereby allowing us to focus directly on assessing the quality of the noise schedules themselves. Second, it allows sampling from multiple noise schedules without the need for retraining, thereby improving efficiency. While relying on a single denoiser trained with a continuous schedule, rather than multiple denoisers trained on specific discrete schedules, may introduce some effects (see our response to Question 3), these are expected to be minor and consistent across all schedules, particularly compared to the variability introduced by training separate models.
> 2. In our response to Reviewer M8z5 (under Weakness 2), we discuss the method’s efficiency and present two approaches to enhance the efficiency of the optimization process: one targeting scenarios with a large number of diffusion steps, and another focusing on high-resolution datasets. Additionally, we would like to note the optimization is performed once per dataset and specified number of diffusion steps, following a straightforward procedure.
> 3. We anticipate that training the denoiser specifically for the spectral recommendation would improve the final results, as the denoiser would learn to map noisy inputs to clean data based on the precise noise characteristics of that schedule. This tailored training enhances accuracy and stability, allowing the model to fully leverage its capacity for the given scheduler. As noted in our response to Question 1, we expect these improvements to be consistent across different noise schedules, thereby preserving their relative performance. Due to time and resource constraints during the response period, we were unable to complete training the denoiser on CIFAR-10 or AFHQv2 using the spectral schedule. However, we recommend training the denoiser accordingly once the spectral schedule is established to derive better results. We will address this point in the paper.
>
> [1] Yunjey Choi and Youngjung Uh and Jaejun Yoo and Jung-Woo Ha. StarGAN v2: Diverse Image Synthesis for Multiple Domains
>
> [2] Tero Karras, Miika Aittala, Timo Aila, and Samuli Laine. Elucidating the design space
> of diffusion-based generative models. Advances in neural information processing systems,
> 35:26565–26577, 2022

---

> > ### Comment · Reviewer_YDdm · 2025-08-04
> > **Response to the rebuttal**
> >
> > I thank the authors for your effort on addressing my concerns on the rebuttal, where most of them have been addressed. The two remaining concerns are just about the Gaussian assumption (also raised by other reviewers) and the scalability of the method as stated by authors that  achieving the noise scheduler might need order of tens of seconds and might not suitable for training from scratch pipeline.
> >
> > In conclusion, I still lean on accepting this paper, but with low confident and suggest the AC to see comments from other reviewers.

---

> ### Comment · Area_Chair_jyj7 · 2025-08-04
> **Action Required: Author–Reviewer Discussion Closing Soon**
>
> Dear Reviewer,
>
>
>
> This is a gentle reminder that the **Author–Reviewer Discussion** phase ends within just three days (by **August 6**). Please take a moment to read the authors’ rebuttal thoroughly and engage in the discussion. Ideally, every reviewer will respond so the authors know their rebuttal has been seen and considered.
>
>
>
> Thank you for your prompt participation!
>
>
>
> Best regards,
>
>
>
> AC

---

> ### Author Response · Authors · 2025-08-05
> **Response to reviewer YDdm**
>
> We thank the reviewer for their response and would like to further clarify the remaining concerns.
>
> Regarding the scalability of our method, as discussed in our response to Question 2, the optimization is performed only once per given dataset and takes only tens of seconds in practice, which we view as a reasonable one-time cost given its repeated use during sampling. Moreover, the techniques we introduced during the rebuttal phase (such as iterative initialization; see our reply to Reviewer M8z5) have substantially reduced the computation time. Hence, we consider the proposed approach to offer an efficient and scalable solution.
>
> In addition, as mentioned in our response to Question 3, training a model from scratch using the spectral noise schedule is expected to improve performance relative to the continuous schedule, rather than lead to degradation. As explained in our response to Question 1, we adopted the continuous schedule in our experiments to enable a fair comparison between our proposed spectral method and existing heuristic baselines.
>
> Regarding the Gaussian assumption, as we noted, this choice provides analytical tractability and allows us to express a closed-form optimization problem in the spectral domain. We demonstrated that the resulting schedules serve as effective recommendations for real-world datasets such as CIFAR-10, MUSIC, SC09, and AFHQv2, which was included during the rebuttal in response to your concerns as a higher-resolution example. We also discussed this aspect in detail in our responses to Reviewer M8z5, including potential directions for future work beyond the Gaussian setting.

---

> > ### Comment · Reviewer_YDdm · 2025-08-06
> > **Thanks for the response**
> >
> > I thank the authors for further clarifications on the two remaining concerns. I satisfy with the rebuttal and still lean on accepting this paper.
> >
> > Thanks

---

> > > ### Author Response · Authors · 2025-08-07
> > >
> > > We thank the reviewer for their positive evaluation and are pleased that our clarifications satisfactorily responded to their concerns.

---

### Comment · Area_Chair_jyj7 · 2025-08-01
**Kindly Engage with Author Responses**

Dear Reviewers,





The authors have submitted their responses to your reviews. At your earliest convenience, please take a moment to engage with their replies. Your continued discussion and clarifications will be invaluable in ensuring a fair and constructive review process for all parties.

Thank you again for your thoughtful contributions and dedication.





Warm regards,



Your AC

---

### Note · Authors · 2025-08-14

Dear AC and Reviewers, \
Thank you for your detailed feedback and the comprehensive review process. \
Our rebuttal addressed the following key points:
- While our analysis assumes Gaussianity, the derived spectral schedules outperform baselines in FID and Wasserstein distance across diverse real-world datasets and modalities (CIFAR-10, MUSIC, AFHQv2, and SC09). By adapting the derived schedules to the spectral characteristics of each dataset, they consistently surpass heuristic baselines across all diffusion step counts, particularly in regimes where discretization error is more pronounced.
- Existing heuristics can serve as plausible candidate solutions within the broader optimization framework, and the spectral schedule may, in retrospect, resemble them. Structural similarities to heuristics do not diminish the significance of our approach, which is based on a principled optimization problem and avoids repeated ad hoc comparisons across noise schedules.
- We demonstrated the scalability of our method on the higher resolution AFHQv2 dataset and analyzed the resulting schedule structure relative to existing heuristics.
- We introduced two approaches to enhance optimization efficiency: one for scenarios with a large number of diffusion steps and another for high resolution datasets. These methods reduce optimization time to a few tens of seconds and were demonstrated on the AFHQv2 dataset.
- We showed that the explicit spectral formulation provides insight into spectral properties and related phenomena, including mean drift and the coarse to fine signal construction behavior.
- We have revised the paper to address comments on figures, missing citations, and typos, incorporate additional experiments conducted during the rebuttal, and discuss potential avenues for future work.

---

### Decision · Program_Chairs · 2025-09-17

**Decision:**

Accept (poster)

**Comment:**

Although stemming from insights based on the Gaussianity assumption in the theoretical analysis, the derived spectral schedules provide a more principled scheduler design through solving an optimization problem, compared to heuristic designs in the literature, and outperform baselines in FID across various datasets. These insights may be useful for broader diffusion model scheduler design. Including additional experiments on larger-scale datasets in a revision would further demonstrate the effectiveness of the proposed method, which I highly recommend. Overall, I recommend **acceptance**.